# High-throughput mathematical analysis identifies Turing networks for patterning with equally diffusing signals

Luciano Marcon[1], Xavier Diego[2,3], James Sharpe[2,3,4], Patrick Müller[1]*

[1]Friedrich Miescher Laboratory of the Max Planck Society, Tübingen, Germany; [2]EMBL-CRG Systems Biology Research Unit, Centre for Genomic Regulation, The Barcelona Institute of Science and Technology, Barcelona, Spain; [3]Universitat Pompeu Fabra, Barcelona, Spain; [4]Institucio Catalana de Recerca i Estudis Avançats, Barcelona, Spain

**Abstract** The Turing reaction-diffusion model explains how identical cells can self-organize to form spatial patterns. It has been suggested that extracellular signaling molecules with different diffusion coefficients underlie this model, but the contribution of cell-autonomous signaling components is largely unknown. We developed an automated mathematical analysis to derive a catalog of realistic Turing networks. This analysis reveals that in the presence of cell-autonomous factors, networks can form a pattern with equally diffusing signals and even for any combination of diffusion coefficients. We provide a software (available at http://www.RDNets.com) to explore these networks and to constrain topologies with qualitative and quantitative experimental data. We use the software to examine the self-organizing networks that control embryonic axis specification and digit patterning. Finally, we demonstrate how existing synthetic circuits can be extended with additional feedbacks to form Turing reaction-diffusion systems. Our study offers a new theoretical framework to understand multicellular pattern formation and enables the wide-spread use of mathematical biology to engineer synthetic patterning systems.

**\*For correspondence:** pmueller@tuebingen.mpg.de

**Competing interests:** The authors declare that no competing interests exist.

## Introduction

How cells self-organize to form ordered structures is a central question in developmental biology (*Hiscock and Megason, 2015*), and identifying self-organizing mechanisms promises to provide new tools for synthetic biology and regenerative medicine (*Chen and Weiss, 2005*; *Guye and Weiss, 2008*; *Isalan et al., 2008*; *Bansagi et al., 2011*; *Chau et al., 2012*; *Mishra et al., 2014*; *Schaerli et al., 2014*; *Wroblewska et al., 2015*). More than six decades ago, Alan Turing proposed a theoretical model in which interactions between diffusible substances can break the initial symmetry of cell fields to form periodic patterns (*Turing, 1952*). Subsequent work from Gierer and Meinhardt postulated that such self-organizing processes require differential diffusivity between a short-range self-enhancing activator and a feedback-induced long-range inhibitor (*Gierer and Meinhardt, 1972*). Numerous studies have proposed models based on these concepts to explain pattern formation during development, including skin appendage specification (*Sick et al., 2006*; *Harris et al., 2005*), lung branching (*Menshykau et al., 2012*; *Hagiwara et al., 2015*), tooth development (*Salazar-Ciudad and Jernvall, 2010*), rugae formation (*Economou et al., 2012*), and digit patterning (*Sheth et al., 2012*; *Raspopovic et al., 2014*). However, the evidence in support of specific activator-inhibitor pairs has been limited, and few studies have provided experimental support for the differential diffusivity of activators and inhibitors (*Kondo and Miura, 2010*; *Marcon and Sharpe, 2012*; *Müller et al., 2012*).

**eLife digest** Developing embryos initially consist of identical cells that specialize over time to create the different parts of the adult animal. More than sixty years ago, Alan Turing proposed that this spontaneous breaking of uniformity could be controlled by two molecules that interact with each other and move by diffusion at different rates between cells. In such "reaction-diffusion" systems, the interactions between the molecules cause repeating peaks in their concentrations in different locations, which could influence how different parts of the embryo develop. However, how these hypothetical molecules relate to the genes that control embryonic development has remained largely unknown.

Marcon et al. have now developed a computational method to identify the conditions that enable periodic patterns to form spontaneously in realistic reaction-diffusion systems with mobile signaling molecules and immobile factors such as membrane-localized receptors. By computationally screening millions of biologically relevant networks, Marcon et al. found that a key requirement of classical Turing models – that the mobile signaling molecules must diffuse at different rates – does not need to be met for patterns to form. Instead, some networks can form patterns with signals that diffuse at equal rates, while others can form patterns with any combination of diffusion rates.

The computational method developed by Marcon et al. can be used to interpret the mechanisms that allow patterns to form in biological systems, such as those that control embryonic development. It can also be used to develop synthetic networks that regulate genes for the formation of tissues in particular spatial patterns.

Pattern formation processes are regulated by the interactions between secreted signaling molecules and their receptors that activate complex cell-autonomous signaling events. However, since most reaction-diffusion models have been reduced to abstract networks of two diffusible reactants, the influence of immobile cell-autonomous factors on reaction-diffusion patterning is largely unknown. Previous theoretical studies on selected network topologies have challenged the differential diffusivity requirements and indicated that in the presence of an immobile substance, patterns can form for a wider range of reaction and diffusion parameters (*Othmer and Scriven, 1969*; *Pearson and Horsthemke, 1989*; *Pearson, 1992*; *Pearson and Bruno, 1992*; *Rauch and Millonas, 2004*; *Levine and Rappel, 2005*; *Miura et al., 2009*; *Raspopovic et al., 2014*; *Korvasova et al., 2015*). These and other studies (*Meinhardt, 2004*; *Werner et al., 2015*) suggest that extending models beyond abstract two-node systems can reveal different pattern formation requirements and may uncover new biologically relevant network designs. However, due to the complex mathematical analysis required to identify and understand such systems, extending reaction-diffusion models to more realistic signaling networks has been challenging, and the main assumption in the field has remained that complex models should reduce to simple systems that require an effective differential diffusivity.

Here, we developed the freely available and user-friendly software RDNets (available at http://www.RDNets.com) to perform a high-throughput mathematical analysis of complex reaction-diffusion networks with non-diffusible components. In comparison to previous numerical studies, this method guarantees completeness, reproducibility, and detailed mechanistic insights into the principles underlying pattern formation. We used RDNets to build a comprehensive catalog of minimal three-node and four-node reaction-diffusion networks that include interactions between diffusible signals and cell-autonomous factors. Our results show that reaction-diffusion systems have three types of requirements for the diffusible signals depending on the network topology: Type I networks require differential diffusivity, Type II networks allow equal diffusivities, and Type III networks allow for unconstrained diffusivity. Overall, 70% of the networks identified by our analysis are of Type II and Type III and thus do not require differential diffusivity to form a spatial pattern. This reveals that realistic reaction-diffusion systems are based on mechanisms that are fundamentally different from the concepts of short-range activation and long-range inhibition based on differential diffusivity (*Gierer and Meinhardt, 1972*) that have been predominant in previous models of pattern formation. Our software can be used to explore these new networks

and is a unique tool to understand *in vivo* reaction-diffusion systems and to engineer synthetic circuits with spatial patterning capabilities.

## Results

Understanding how complex gene regulatory networks control cellular behavior is a challenging problem in biology; even small networks can contain regulatory feedbacks that make systems behaviors difficult to predict (*Le Novère, 2015*). Mathematical biology has helped to identify network motifs that underlie basic behaviors such as oscillations, bi-stability or noise reduction (*Kepler and Elston, 2001*; *Shen-Orr et al., 2002*; *Mangan and Alon, 2003*), but this approach has been difficult to scale up to more complex networks and behaviors. Previous studies have overcome this obstacle by using numerical simulations to screen for topologies that implement a certain behavior (*Salazar-Ciudad et al., 2000*; *Ma et al., 2009*; *Cotterell and Sharpe, 2010*). However, such simulations demand large computational power, their coverage is incomplete, and they do not have the explanatory power of analytical approaches. The ideal tool to analyze the behavior of gene networks should retain the explanatory power of mathematical approaches and yet be able to comprehensively screen for network topologies and the underlying mechanistic principles.

We have developed the web-based software RDNets (http://www.RDNets.com) to derive a comprehensive catalog of minimal three-node and four-node reaction-diffusion networks and their pattern-forming conditions. Our analysis reveals that networks have different diffusivity requirements depending on the topology. RDNets can constrain candidate topologies with qualitative and quantitative experimental data, making it a convenient tool for users that aim to study developmental patterning networks or to design synthetic reaction-diffusion circuits.

### Automated mathematical analysis of reaction-diffusion networks

We developed an automated linear stability analysis (*Murray, 2003*) to derive the pattern forming conditions of networks with *N* nodes (*Figure 1a*, Materials and methods). Linear stability analysis determines whether a system can form a pattern by testing i) if the concentrations of the reactants are stable at steady state, and ii) if diffusion-driven instabilities arise with small perturbations. Because of its mathematical complexity, this type of analysis has been the exclusive domain of mathematicians and systems biologists (*Koch and Meinhardt, 1994*; *Satnoianu et al., 2000*; *Murray, 2003*; *Miura and Maini, 2004*), and its application beyond two-reactant models has required dedicated theoretical studies for selected networks (*Othmer and Scriven, 1971*; *White and Gilligan, 1998*; *Klika et al., 2012*; *Korvasova et al., 2015*). To generalize the analysis to networks with more than two nodes, we utilized a modern computer algebra system and developed the software pipeline RDNets that automates the algebraic calculations. Within this framework, secreted molecules like ligands and extracellular inhibitors are represented by diffusible nodes, and cell-autonomous components such as receptors and kinases are represented by non-diffusible nodes. Our software analyzes networks with *k* interactions between the nodes; these interactions are represented by first order kinetics rates, where a positive rate corresponds to an activation and a negative rate to an inhibition.

The software pipeline comprises six steps to identify patterning networks:

1. Construction of a list of possible networks of size *k*.
2. Selection of strongly connected networks without isolated nodes or nodes that solely act as read-outs.
3. Deletion of symmetric networks, such that isomorphic networks are considered only once.
4. Selection of networks that are stable in the absence of diffusion (i.e. homogeneous steady state *stability*).
5. Selection of networks that are unstable in the presence of diffusion (i.e. *instability* to spatial perturbations).
6. Analysis of the possible reaction-diffusion topologies associated with the networks and derivation of the resulting in-phase and out-of-phase patterns.

Steps 4 and 5 represent the core part of the automated linear stability analysis and involve the majority of analytical computations. In Step 6, our software screens the possible reaction-diffusion topologies associated with a network. A reaction-diffusion network of size *k* defines only a set of *k*

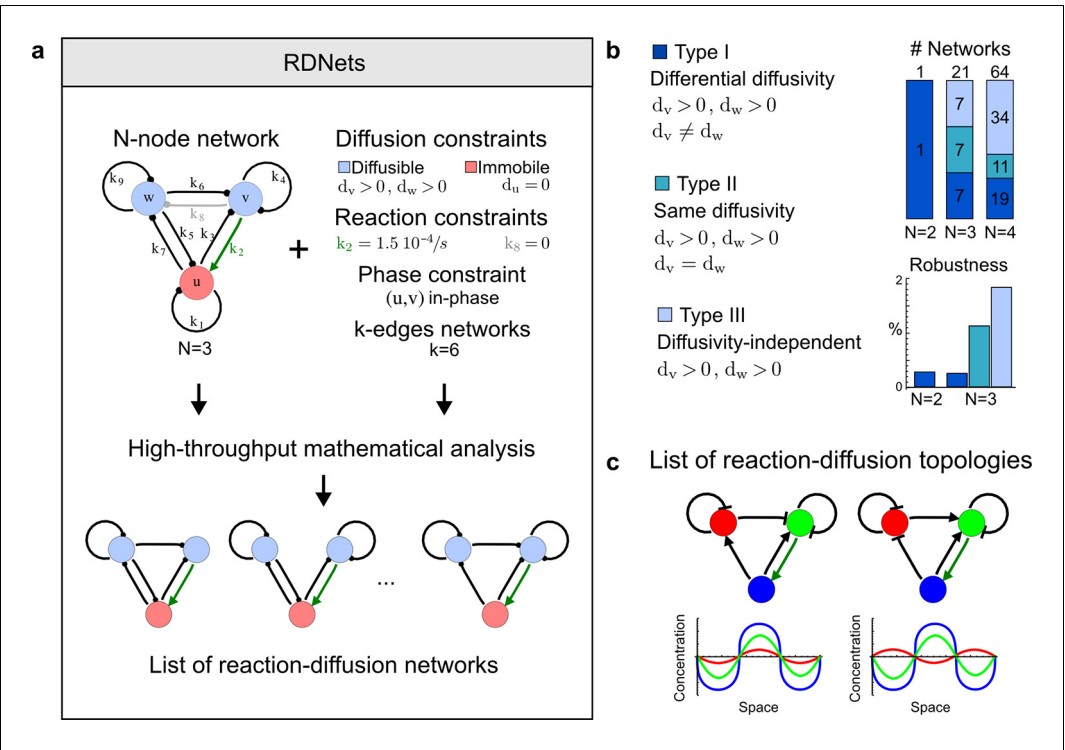

**Figure 1.** High-throughput screen for reaction-diffusion patterning networks using RDNets. (**a**) Schematic representation of the software RDNets to identify pattern-forming networks. RDNets exploits a computer algebra system for high-throughput mathematical analysis of reaction-diffusion networks with *N* nodes and *k* edges. Diffusion and reaction constraints, including the number of diffusible (blue) and non-diffusible (red) nodes and quantitative parameters (here: $k_2$, $k_8$), can be specified as inputs. Additionally, the phase of the resulting periodic pattern can be selected. A list of reaction-diffusion networks is given as output. (**b**) Bar charts summarizing the number of networks for the 2-, 3-, and 4-node signaling network cases. Resulting networks can be of three types: Type I requires differential diffusivity, Type II allows for equal diffusivity, and Type III is diffusivity-independent. Type II and Type III networks are more robust to parameter changes than Type I networks. (**c**) Simulations of the possible topologies associated with a given network show that the minimal three-node systems can form in-phase and out-of-phase periodic patterns depending on the network topology. See Appendix 6 for a full list of parameters.

The following figure supplements are available for figure 1:

**Figure supplement 1.** Catalog of all 3-node networks with two diffusible nodes (blue), one non-diffusible node (red) and six interactions.

**Figure supplement 2.** Comprehensive catalog of 4-node Type I reaction-diffusion networks with two diffusible (blue) and two non-diffusible (red) nodes representing the interaction between two signaling pathways.

**Figure supplement 3.** Comprehensive catalog of 4-node Type II reaction-diffusion networks with two diffusible (blue) and two non-diffusible (red) nodes representing the interaction between two signaling pathways.

**Figure supplement 4.** Comprehensive catalog of 4-node Type III reaction-diffusion networks with two diffusible (blue) and two non-diffusible (red) nodes representing the interaction between two signaling pathways.

regulatory links between nodes but does not make any assumption on whether these are activating or inhibiting interactions. In the following, we refer to the possible combination of activating and inhibiting interactions as 'network topologies'.

# High-throughput mathematical screen for minimal three-node and four-node reaction-diffusion networks

We used our software RDNets to systematically explore the effect of cell-autonomous factors in reaction-diffusion models for the generation of self-organizing patterns. We studied two types of networks: a) 3-node networks with two diffusible nodes and one non-diffusible node representing the interaction between two secreted molecules and one signaling pathway, and b) 4-node networks with two diffusible nodes and two non-diffusible nodes representing the interaction between multiple ligands and signaling pathways. *Table 1* shows the number of networks identified at each step of our automated mathematical analysis (see *Figure 1—figure supplements 1–4* for the complete catalog of the identified reaction-diffusion networks). Our analysis revealed that in the presence of cell-autonomous factors there are three types of networks with different constraints on the diffusible signals:

$$\text{Type I (requires differential diffusivity)}: \quad \exists\,(d_i, d_j) \subset D, d_j \neq d_j \wedge \forall\, d_i \in D, d_i > 0$$
$$\text{Type II (allows for equal diffusivity)}: \quad \forall\,(d_i, d_j) \subset D, d_j = d_j \wedge \forall\, d_i \in D, d_i > 0$$
$$\text{Type III (unconstrained diffusivity)}: \quad \forall\, d_i \in D, d_i > 0$$

where $D$ is the list of diffusion coefficients that are non-zero.

We found that 70% of the identified networks with non-diffusible nodes are of Type II and Type III (*Figure 1b*), showing that in the presence of cell-autonomous factors the differential diffusivity requirement is unexpectedly rare. Type III networks have never been characterized before and surprisingly have patterning conditions that are independent of specific diffusion rates. We found that Type III networks are not only numerous but also extremely robust to changes in parameter values compared to Type I and Type II networks (*Figure 1b*, Materials and methods). Using numerical simulations, we systematically confirmed our mathematical analysis and determined that a network can form all possible combinations of in-phase or out-of-phase periodic patterns depending on the network topology (*Figure 1c*, Appendix 1). Together, our results show that realistic reaction-diffusion networks are intrinsically robust, do not require differential diffusivity, and have patterning capabilities identical to classical two-node reaction-diffusion models. Importantly, the novel class of Type III networks that we discovered suggests a new mechanism of pattern formation that is independent of short-range activation and long-range inhibition based on differential diffusivity.

## The network topology defines Type I, Type II and Type III networks

To obtain insight into the organizing principles underlying the three types of networks identified by our high-throughput analysis, we developed a novel graph-theoretical formalism to express the pattern forming conditions in terms of network feedbacks rather than reaction parameters (see Materials and methods and Appendix 2). This analysis determines which feedback cycles contribute to the stability and the instability conditions (*Figure 2a,b*) and defines the topological features that underlie Type I, Type II, and Type III networks. In agreement with previous studies (*Murray, 2003*), our analysis confirmed that two-node networks can only simultaneously satisfy the stability and instability conditions when the diffusion ratio $d$ between the inhibitor and the activator is greater than one (*Figure 2*, left column). This observation has been linked with the widespread belief that reaction-diffusion systems require differential diffusivity to implement short-range auto-activation and long-

**Table 1.** From an initial number of possible networks (Step 1), RDNets progressively identifies reaction-diffusion networks that can form a pattern (Step 6).

| Steps | 3 nodes | | 4 nodes | |
|---|---|---|---|---|
| | # networks | # topologies | # networks | # topologies |
| 1. Minimal systems | 84 | 5376 | 11440 | 1464320 |
| 2. Strongly connected | 48 | 3072 | 2284 | 292352 |
| 3. Non-symmetrical | 25 | 1600 | 597 | 76416 |
| 4. Stable | 24 | 556 | 324 | 8640 |
| 5-6. Reaction-diffusion | 21 | 84 | 64 | 512 |

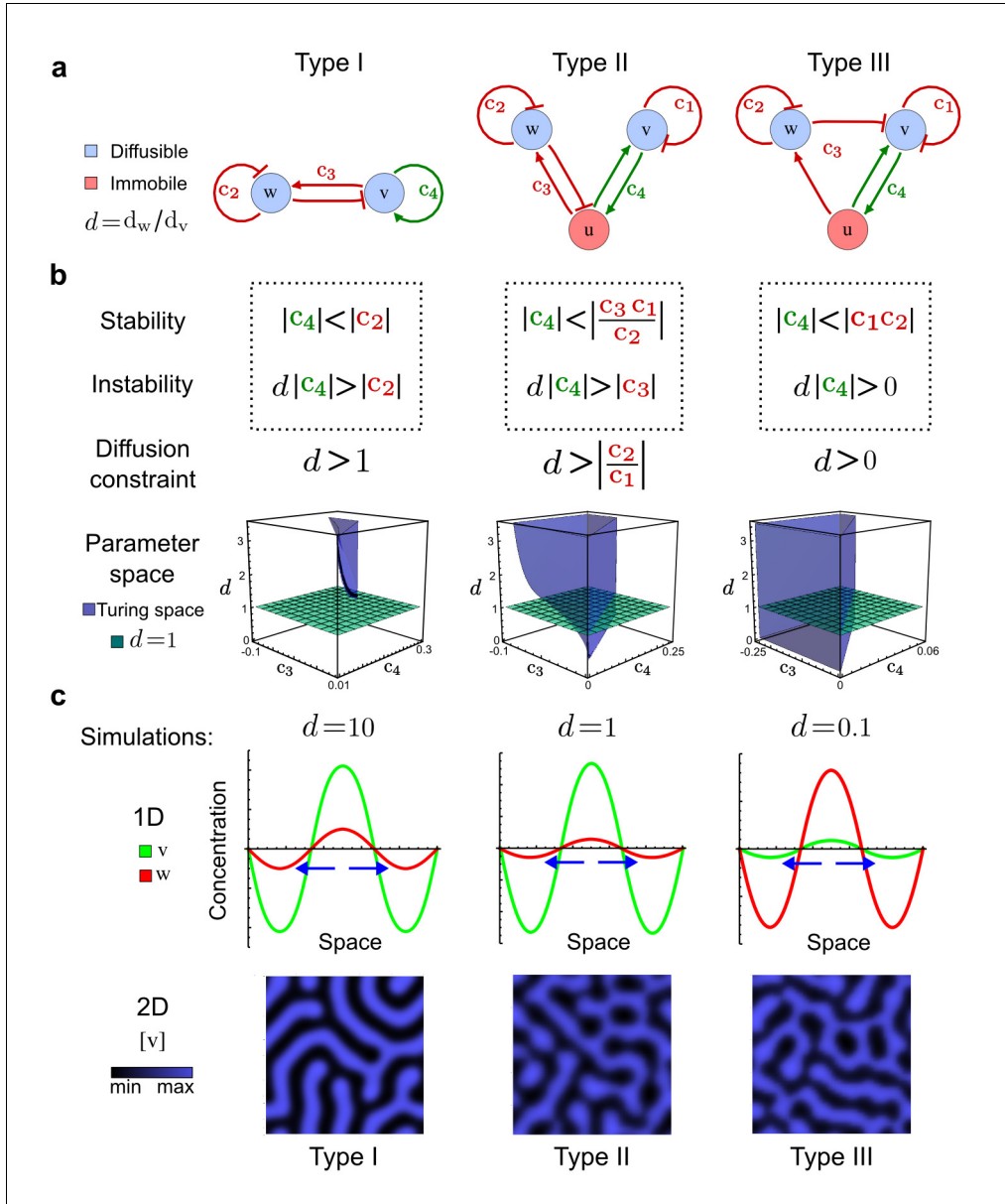

**Figure 2.** Analysis of the organizing principles underlying reaction-diffusion networks. (a) Schematic diagram of a 2-node network of Type I, a 3-node network of Type II, and a 3-node network of Type III. $c_1$ to $c_4$ indicate feedback cycles, red indicates overall inhibition and green overall activation, and $d=d_w/d_v$ represents the diffusion ratio. The two-node network (left column) is a classical activator-inhibitor system, the other two networks are more realistic 3-node networks wired through a cell-autonomous factor $u$. (b) Linear stability analysis of the topologies shown in (a) reveals that pattern-forming conditions require a trade-off between stability and instability feedback cycles, which gives rise to the diffusion constraint. The blue volume highlights the parameter set that allows for pattern formation (Turing space); the three parameters $c_3$, $c_4$, and $d$ vary independently along the axes. Intersecting the Turing space with a plane of equal diffusion coefficients $d=1$ shows that, in contrast to Type II and Type III networks, patterning in Type I networks is not possible with equal diffusivities. (c) 1D simulations show that the apparent longer inhibitor range (blue arrows) observed in the Type I network is also maintained in the Type II network even with $d=1$ and therefore does not result from differential diffusivity. The Type III network with $d=0.1$ surprisingly shows an apparent longer range for the activator v. 1D and 2D simulations show that Type II and Type III topologies form patterns similar to those generated by classical 2-node models. See Appendix 6 for a full list of parameters.

range inhibition. Our analysis instead suggests that the differential diffusivity requirement arises from the opposite nature of the stability and instability conditions, which require that the destabilizing feedback must be both higher and lower than the stabilizing feedback. Since the diffusion term only appears in the destabilizing condition, it assumes the role of a unique pivot that can satisfy both conditions simultaneously when $d > 1$. Our results indicate that the presence of non-diffusible nodes allows feedbacks that do not appear in the instability conditions to act as an additional pivot to satisfy both conditions simultaneously by increasing stability. This is the case for most Type II networks (*Figure 2*, middle column) that contain additional negative feedbacks that allow for equal diffusivities (*Klika et al., 2012*; *Korvasova et al., 2015*). Importantly, our analysis also reveals that non-diffusible nodes can implement positive feedbacks that can drive the network unstable independently of stabilizing feedbacks and for any diffusion ratio $d$. This is the case for Type III networks (*Figure 2*, right column), where the stability and instability conditions are uncoupled and can be simultaneously satisfied for large parameter sets. This is possible because immobile factors can act as 'capacitors' that retain and amplify perturbations independently of the reactants' diffusion coefficients (see Appendix 3 for details). Such systems represent a fundamentally new pattern formation mechanism that has not been described previously.

Together, our results show that models based on 'short-range auto-activation and long-range inhibition' implemented by differential diffusivity are only a special case of a general trade-off between stabilizing and destabilizing feedbacks required for pattern formation. The virtually indistinguishable simulations of Type I networks with differential diffusivity and Type II networks with equal diffusivities reveal that the final aspect of the periodic patterns does not reflect a difference in the range of activators and inhibitors but only a difference in their amplitude (*Figure 2c*, see Appendix 3 for details). Indeed, in other Type II and Type III networks the relationship between the amplitude of activators and inhibitors can even be inverted, such that the perceived range of the activator appears larger than the perceived range of the inhibitor. Therefore, in contrast to previous studies (*Kondo and Miura, 2010*), we propose that long-range lateral inhibition is not required to limit the expansion of the activator (Appendix 3).

## Qualitative and quantitative constraints for candidate networks

To demonstrate the functionality and applicability of RDNets, we analyzed two known self-organizing developmental patterning networks, the Nodal/Lefty reaction-diffusion system and the BMP/Sox9/Wnt network. In the following, we show how quantitative and qualitative experimental data from these developmental systems can be used to constrain the high-throughput analysis and to characterize the possible underlying patterning topologies.

It has been proposed that Nodal and Lefty implement an activator-inhibitor system that patterns the germ layers and the left-right axis in vertebrates (*Chen and Schier, 2001*; *Shiratori and Hamada, 2006*; *Shen, 2007*; *Meinhardt, 2009*; *Schier, 2009*; *Kondo and Miura, 2010*; *Rogers and Schier, 2011*; *Korvasova et al., 2015*) (*Figure 3a*). In agreement with this hypothesis, the self-enhancing activator Nodal has been shown to diffuse 7.5 times slower than the feedback-induced inhibitor Lefty in living zebrafish embryos (*Müller et al., 2012*). The Nodal/Lefty system has been modeled as a two-component activator-inhibitor system (*Nakamura et al., 2006*; *Müller et al., 2012*), but the influence of cell-autonomous factors including receptors and the well-characterized intracellular signal transduction cascade via phosphorylated Smad2/3 (*Schier, 2009*) has not been studied. We used our software to screen for networks that extend the two-node Nodal/Lefty system with a non-diffusible node corresponding to active Nodal signaling (*Figure 3b*). The screen was constrained with known qualitative regulatory interactions: a positive feedback loop between Nodal and its signaling, and a promotion of Lefty by Nodal signaling (*Figure 3b*). Moreover, we constrained the two negative self-regulations on Nodal and Lefty, which represent their clearance from the diffusible pool, with the previously measured clearance rate constants (*Müller et al., 2012*). Finally, we selected only reaction-diffusion networks that produced in-phase patterns of Nodal and Lefty, which recapitulate their overlapping expression domains (*Schier, 2009*). With these constraints, our mathematical analysis identified just two possible minimal networks: In one network Lefty inhibits Nodal signaling indirectly at the receptor level, and in the other network Lefty inhibits Nodal directly (*Figure 3b*). These predictions are in agreement with the two possible mechanisms by which Lefty has been proposed to inhibit Nodal activity: by binding to the Nodal receptor or by directly sequestering Nodal (*Chen and Shen, 2004*). However, the role and significance of these two alternative

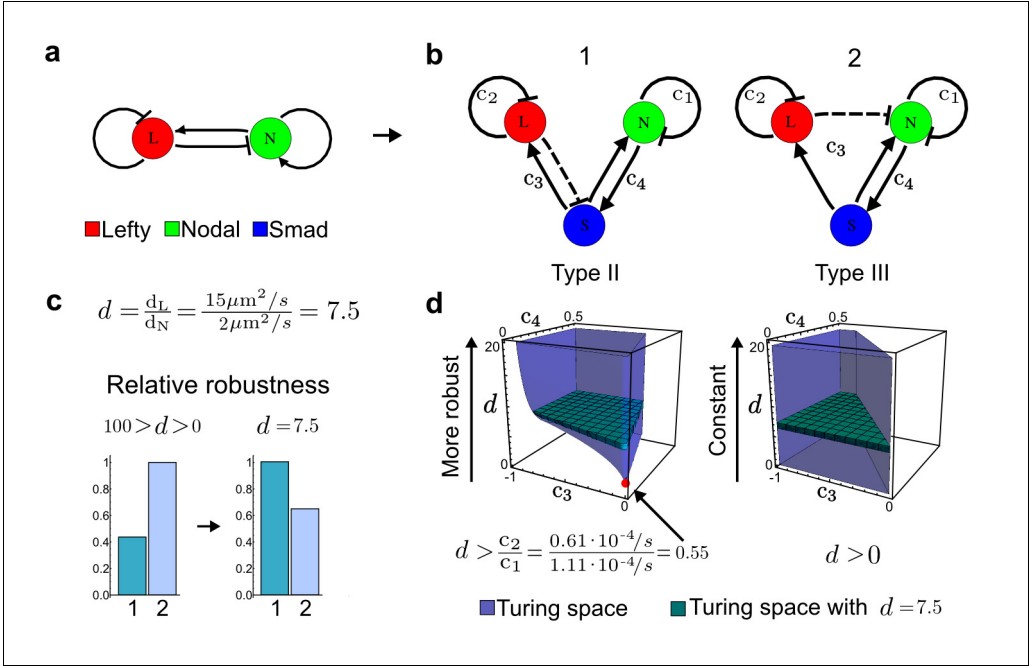

**Figure 3.** Modeling of the Nodal/Lefty reaction-diffusion system with realistic signaling networks. (**a**) Schematic diagram of the Nodal/Lefty activator-inhibitor system. Nodal (green) is the self-enhancing activator that promotes the feedback inhibitor Lefty (red). (**b**) Extension of the Nodal/Lefty system with an immobile cell/receptor-complex node (blue) to distinguish between two possible feedback modes. In both networks, the self-enhancing activation and the Nodal-induced Lefty expression occurs through a non-diffusible cell/receptor-complex represented by the activated signal transducer pSmad2/3 (*S*, blue). In the Type II network, Lefty inhibits Nodal through the receptor node *S*, whereas in the Type III network, Lefty inhibits Nodal directly (dashed lines). (**c,d**) The Type III network is more robust to parameter changes over a broader range of diffusivities (bar chart on the left and bigger Turing space [blue volume]) compared to the Type II network. However, constraining the two topologies with previously measured diffusion coefficients (*d*=7.5) demonstrates that the Type II network is more robust for biologically relevant parameters (bar chart on the right and bigger area of the green plane corresponding to *d*=7.5 within the Turing space [blue volume]). Experimental data for the previously measured clearance rate constants ($c_1$, $c_2$) of Nodal and Lefty predicts that the minimum allowed diffusion ratio for the Type II network is *d*=0.55 (red dot).

The following figure supplement is available for figure 3:

**Figure supplement 1.** A possible evolutionary scenario for evolving the differential diffusivity of Nodal and Lefty.

mechanisms for Nodal/Lefty-mediated patterning has remained unclear (*Cheng et al., 2004*; *Middleton et al., 2013*). Our mathematical analysis predicts that the first mechanism (Lefty blocks the receptor complex) determines a Type II network, whereas the second mechanism (Lefty blocks Nodal directly) determines a Type III network. Importantly, both models suggest that the Nodal/Lefty system may form patterns without differential diffusivity of activator and inhibitor. Using the clearance rate constants of Nodal and Lefty as quantitative constraints, our mathematical analysis predicts a possible minimum diffusion ratio $d = 0.55$ for the Type II network, whereas the Type III network allows for any combination of diffusion coefficients (*Figure 3d*). The robustness analysis of the networks shows that for unconstrained valued of *d*, the Type III network is more robust to parameter changes (*Figure 3c*). However, when we fix the diffusion ratio to the experimentally quantified value (*Müller et al., 2012*) ($d = 7.5$), the Type II network becomes more robust than the Type III network (*Figure 3d*). This shows that, while Nodal and Lefty do not necessarily need to have different diffusivities to form a pattern, the combination of differential diffusivity and clearance rate constants increases the robustness of the Type II system.

As a second example, we used RDNets to analyze the BMP/Sox9/Wnt (BSW) self-organizing network that underlies digit patterning (*Sheth et al., 2012*; *Raspopovic et al., 2014*). The expression patterns and the signaling activity of the network components have been well-characterized showing

that Sox9 forms periodic expression peaks that are out-of-phase of *BMP* expression and Wnt activity (*Figure 4a*). A three-node reaction-diffusion network with two diffusible nodes for the secreted signals BMP and Wnt and a non-diffusible node for the transcription factor Sox9 has previously been derived based on the known regulatory interactions (*Figure 4b*). It was shown that this network recapitulates the out-of-phase pattern between BMP/Wnt and Sox9 and forms a pattern with extremely low differential diffusivity requirements ($d = 1.25$). Our comprehensive mathematical analysis reveals that this three-node system is just another topology of the reaction-diffusion network that we analyzed for the extended Nodal/Lefty system; it is therefore a Type II network that can potentially form a pattern even when BMP and Wnt have equal diffusion coefficients. In previous studies, this observation was missed because the clearance rates of BMP and Wnt had been assumed to be identical (*Raspopovic et al., 2014*). However, as we showed in the previous example for Nodal and Lefty, if BMP is cleared faster than Wnt, the diffusion ratio can be equal to or lower than one, $d \leq 1$. The three-node BSW model recapitulates the out-of-phase pattern between BMP/Wnt and Sox9, but

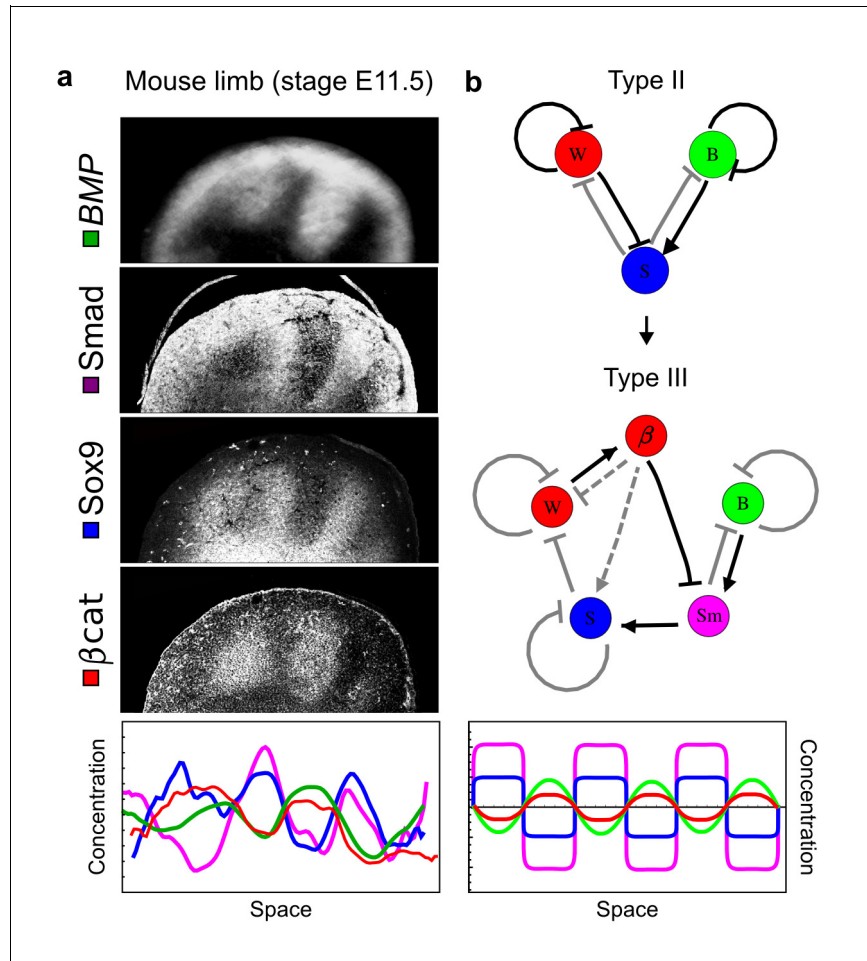

**Figure 4.** Modeling of mouse digit patterning with realistic signaling networks. (a) Experimental patterns of *BMP* (green), pSmad1/5/8 (purple), Sox9 (blue), and β-catenin (red) in a mouse limb at stage E11.5 (data reproduced from *Raspopovic et al., 2014*). (b) Extension of a previously proposed simple three-node network for digit patterning involving BMP, Sox9, and Wnt to a more realistic five-node network incorporating known interactions (black) between Wnt (W, red), BMP (B, green), Smad1/5/8 (Sm, pink), Sox9 (S, blue), and β-catenin (β, red); interactions predicted by RDNets are shown in gray, and dashed lines correspond to alternative interactions that implement networks with similar robustness. The simulations of the new five-node network recapitulate the unintuitive out-of-phase pattern between *BMP* expression (green) and its own signaling through pSmad1/5/8 (purple). The mathematical analysis predicts that these patterns can be formed when β-catenin inhibits Sox9 indirectly through pSmad1/5/8. See Appendix 6 for a full list of parameters.

due to its high abstraction level it does not explain the opposite *BMP* expression and BMP activity patterns observed in the experimental data (*Figure 4a*). We therefore used RDNets to screen for more complex models with five nodes that represent all components of the network: two diffusible nodes for BMP (B) and Wnt (W) and three non-diffusible nodes, one for the canonical BMP pathway through pSmad1/5/8 (Sm), one for the intracellular Wnt signaling cascade (β-catenin, β), and one for Sox9 (S). We selected only networks that formed in-phase and out-of-phase patterns reflecting the experimental data (*Figure 4a*). Previous studies (*Raspopovic et al., 2014*) showed that Sox9 is promoted by BMP signaling through pSmad1/5/8 and is inhibited by Wnt through β-catenin. Similar to the Nodal/Lefty example, we constrained the mathematical screen by incorporating these known regulatory interactions. Unexpectedly, the screen revealed that if β-catenin would directly inhibit Sox9, no network could recapitulate the out-of-phase patterns between *BMP* expression and BMP signaling. By performing a more general screen that left this interaction unconstrained, we found that the opposite *BMP* expression and signaling patterns can be obtained when β-catenin indirectly inhibits Sox9 through pSmad/1/5/8. RDNets also predicts that the most robust networks include the following additional interactions: i) a negative feedback from Sox9 to Wnt, ii) a negative feedback from pSmad1/5/8 to BMP, and iii) either a positive feedback from β-catenin to Sox9 or a negative feedback from β-catenin on Wnt (*Figure 4b*, gray arrows). Interestingly, the majority of networks identified by our screen was of Type III, suggesting that the proportion of Type III networks increases when more non-diffusible nodes are added.

## Designing robust synthetic reaction-diffusion circuits

Although reaction-diffusion mechanisms have a simple network design, they exhibit unique self-organizing capabilities making them appealing for synthetic engineering (*Diambra et al., 2015*). So far, the synthetic implementation of reaction-diffusion systems has been impeded by the small pattern-forming parameter space of simple two-node models, their requirement for differential diffusivity (*Carvalho et al., 2014*), and a general gap between abstract models and real sender-receiver reaction-diffusion circuits (*Marcon and Sharpe, 2012*; *Barcena Menendez et al., 2015*).

RDNets provides a comprehensive catalog of reaction-diffusion networks that do not require differential diffusivity of the signaling molecules, which enables bioengineers to explore new mechanisms to form periodic spatial patterns in a robust manner. We demonstrate the utility of RDNets by proposing an extension to an existing synthetic circuit for cell-cell communication in yeast (*Chen and Weiss, 2005*). The original synthetic circuit introduced a diffusible plant hormone, cytokinin isopentenyladenine (IP), and its receptor AtCRE1 into yeast (*Figure 5a*). This circuit was used to implement a sender-receiver and a quorum sensing mechanism based on a positive feedback loop between IP-signaling and IP (*Figure 5a*). We used RDNets to identify possible signaling networks that can extend this positive feedback with additional interactions to form a reaction-diffusion pattern. Since at least two diffusible nodes are required to form a pattern (*Murray, 2003*), we screened minimal 4-node networks that include the engineered positive feedback and candidate interactions with another diffusible node. In order to look for realistic and easily implementable signaling circuits, we explored only networks with interactions between diffusible nodes through non-diffusible factors representing intracellular signaling cascades. We also imposed self-regulations on diffusible nodes to be exclusively inhibitory, representing decay. With these constraints, our high-throughput analysis identified 16 minimal reaction-diffusion networks (5 Type I, 3 Type II, 8 Type III), of which the Type II and Type III networks were most robust to parameter changes (*Figure 5—figure supplement 1*). In the following, we demonstrate how the conditions derived by RDNets can be used to engineer the most simple and robust Type II network (*Figure 5a* - right). In addition to the positive feedback loop, this network contains three additional negative feedbacks: two are self-regulations that correspond to decay, and one is a negative feedback between the newly introduced diffusible node and the non-diffusible node representing the receptor. This network suggests that a simple extension to the circuit developed in *Chen and Weiss (2005)* could be obtained by a) destabilizing the signaling hormone and the receptor to increase their turn-over ($c_1$, $c_2$), and b) introducing another hormone that signals through the same receptor and implements a negative feedback loop to its own expression or activity ($c_3$, *Figure 5a* - right).

In addition to revealing possible topologies, our automated analysis provides the mathematical formulae of pattern forming conditions. This feature together with the specification of quantitative constraints can be used to calculate pattern forming parameter ranges. To determine the strength

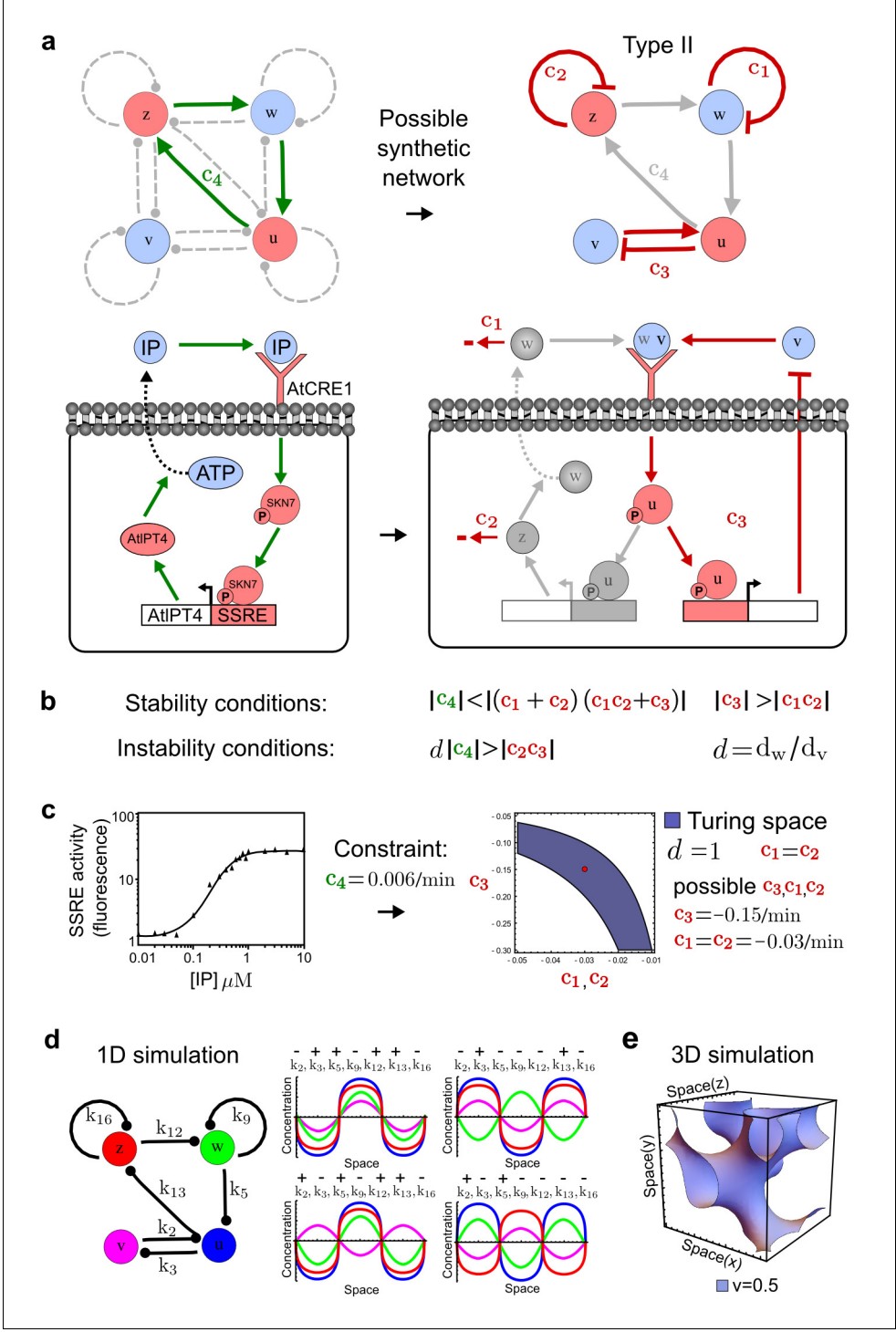

**Figure 5.** Combining signaling modules to form new synthetic reaction-diffusion networks. (**a**) Left: Schematic diagram of a four-node network to engineer a patterning system from an existing signaling module (**Chen and Weiss, 2005**) that implements a positive feedback ($c_4$, green). In the previously engineered synthetic network, the positive feedback highlighted by $c_4$ was implemented by the hormone Cytokinin isopentenyladenine (IP) that activates the receptor AtCRE1 to induce the SSRE-promoter-driven expression of AtIPT4, which catalyzes IP production. Right: A possible Type II reaction-diffusion network predicted by RDNets, in which the positive feedback module composed of $w$, $u$ and $z$ (representing IP, receptors/transducers, and AtIPT4 shown in (**a**)) is extended by a node $v$ that activates $u$, which in turn inhibits $v$ (cycle $c_3$). The cycles $c_1$ and $c_2$ correspond to signal decay. (**b**) Stability and instability conditions of the predicted network. (**c**) Constraining RDNets with previous

*Figure 5 continued on next page*

*Figure 5 continued*

measurements of the positive feedback cycle $c_4$ obtained by fitting experimental data (graph on the left, *Chen and Weiss, 2005*) identifies exact parameter ranges for the new interactions in the synthetic reaction-diffusion network (graph and formulae on the right). (d) 1D simulations show that different topologies of this synthetic network can be engineered to produce all possible in- and out-of-phase periodic patterns depending on the sign of the reaction rates shown above the graphs. (e) A 3D simulation of the synthetic patterning system forms tubular structures that could be exploited for tissue engineering.

The following figure supplement is available for figure 5:

**Figure supplement 1.** Catalog of possible synthetic networks that extend an existing feedback loop (red arrows).

of the newly introduced negative feedbacks required for pattern formation, we constrained the positive cycle strength with the first-order kinetic rate quantified in *Chen and Weiss (2005)* by fitting measurements of the signaling activity of IP (*Figure 5c*). Moreover, we assumed that the diffusion and decay rates are similar. With these constraints, RDNets determined that the newly introduced negative feedback has to be stronger than the positive feedback and decay rates (*Figure 5c* - right). This could be implemented using the more responsive IP-signaling promoter (TR-SSRE) developed in *Chen and Weiss (2005)* to drive the expression of the inhibitor. This specific synthetic network represents only one possibility. We find other Type III networks to be even more robust to parameter changes, but they appear to require the design of more complex synthetic circuits (Appendix 4). Once a synthetic network is designed, RDNets can also be used to automatically derive kinetic models that can simulate the reaction-diffusion network (*Figure 5d,e*, Appendix 5). Numerical simulations can be used to investigate the qualitative aspect of the pattern and its spatial periodicity. In the long term, all these features open new avenues for designing synthetic reaction-diffusion circuits that could be coupled with gene expression to enable complex applications, such as fabrication of spatially patterned three-dimensional biomaterials and tissue engineering in mammalian cells (*Chen and Weiss, 2005*; *Carvalho et al., 2014*).

## Discussion

We developed the web-based software RDNets, which exploits a modern computer algebra system to identify new reaction-diffusion networks that reflect realistic signaling systems with diffusible and cell-autonomous factors. Our approach is a new example of high-throughput mathematical analysis, which has several benefits over previous numerical approaches (*Ma et al., 2009*). First, RDNets can be run from most web browsers and does not demand large computational power. Second, our mathematical analysis yields closed-form solutions and is complete, in contrast to numerical simulations that can necessarily only sample from a small region of the entire parameter space. Third, RDNets derives the conditions for pattern formation and therefore provides mechanistic explanatory power to the users. In addition, it helps to identify reaction-diffusion topologies that are in agreement with qualitative and quantitative experimental constraints, which makes it an unprecedented tool for users that aim to study developmental patterning networks or to design reaction-diffusion synthetic circuits.

Motivated by theoretical studies that showed that non-diffusible factors can influence pattern forming conditions, we used our software to systematically explore the effect of non-diffusible reactants in reaction-diffusion models. Our analytical approach is both comprehensive and informative and reveals that depending on the network topology, reaction-diffusion systems can belong to three classes: Type I systems that require differential diffusivity, Type II systems that can form patterns with equal diffusivity, and Type III systems that form patterns independent of specific diffusion rates. In particular, the novel class of Type III networks has not been described before and challenges models of short-range activation and long-range inhibition based on differential diffusivity that have dominated the field of developmental and theoretical biology for decades (see Appendix 3 for details).

We used RDNets to obtain new mechanistic insight into two developmental patterning systems. By using quantitative data to constrain possible patterning networks, we found a Type II and a Type

III topology that extend the Nodal/Lefty activator-inhibitor system with realistic cell-autonomous signaling. In such extended networks, Nodal and Lefty do not necessarily need to have different diffusivities to form a pattern. However, our results suggest that the differential diffusivity can contribute to a more robust patterning system in Type II networks with indirect Nodal signaling inhibition. We propose that the high general robustness of the Type III network might have played a role for the evolution of the Nodal/Lefty reaction-diffusion system in the first place, and that the indirect Nodal signaling inhibition of Type II networks might have been fine-tuned during evolution (*Figure 3—figure supplement 1*). Similarly, we extended the three-node BSW digit patterning model with additional previously characterized cell-autonomous factors and constrained a five-node model with qualitative data. Our analysis identified realistic network topologies that accurately reflect the previously puzzling opposite pattern of BMP ligands and BMP signaling and predicts novel interactions between network components.

Finally, we used RDNets to design a novel synthetic patterning circuit based on a previously engineered positive feedback module. Identifying a comprehensive catalog of gene networks that can perform a certain behavior has been shown to be a successful strategy to uncover the design space of stripe-forming networks (*Cotterell and Sharpe, 2010*), which can be directly useful to synthetic biology. In particular, this approach permitted a whole family of network mechanisms to be synthetically constructed in bacteria – all capable of forming a gene expression stripe in a bacterial lawn (*Schaerli et al., 2014*). Similarly, our software provides a comprehensive catalog of reaction-diffusion networks and enables bioengineers to explore new mechanisms to form periodic spatial patterns in a robust manner. These networks explicitly include non-diffusible factors that mediate signaling and are easy to relate with sender-receiver synthetic toolkits (*Barcena Menendez et al., 2015*). In addition, we found that the majority of realistic reaction-diffusion networks eliminate the differential diffusivity requirement that is difficult to implement synthetically (*Carvalho et al., 2014*; *Barcena Menendez et al., 2015*). The possibility to use qualitative and quantitative constraints to screen for reaction-diffusion networks makes RDNets a unique tool to customize patterning systems from initial starting networks. Moreover, the pattern-forming conditions derived by the software can be used to estimate parameter ranges and network robustness. Particularly promising is our finding that each network is associated with a set of topologies that exhaustively determine all the in-phase and out-of-phase relations between periodic patterns (*Figure 5d*). It is therefore possible to design network topologies that promote the co-localized expression of any desired combination of factors. This will enable novel applications in tissue engineering, where the co-localized expression of differentiating factors can be used to induce specific tissues (*Kaplan et al., 2005*). Coupled with the three-dimensional pattern-forming capabilities of reaction-diffusion mechanisms (*Figure 5e*), this could be used to devise new strategies for engineering scaffolds or tissues with complex architecture.

In summary, our analysis defines new concepts of reaction-diffusion-mediated patterning that are directly relevant for developmental and synthetic biology. We demonstrate three applications of our software RDNets to understand developmental mechanisms and to design synthetic patterning systems, but this approach can be extended to numerous other patterning processes (*Economou et al., 2012*; *Menshykau et al., 2012*; *Hagiwara et al., 2015*). We therefore expect that RDNets will contribute to the wide-spread use of mathematical biology and that a similar approach could be applied to other dynamical processes such as oscillations and traveling waves (*Bement et al., 2015*).

# Materials and methods

## Details of the automated mathematical analysis

We analyzed reaction-diffusion networks represented by a reaction matrix $J$ and a diffusion matrix $D$ of size $NxN$, where $N$ is equal to the number of nodes. The matrix $J$ corresponds to the Jacobian of the reaction-diffusion system and contains partial derivatives that describe the relative influence of one node on another. Elements of the reaction matrix represent the first order kinetics rates of the regulatory interactions in the network, where a positive rate corresponds to an activation and a negative rate to an inhibition. The matrix $D$ contains the diffusion rates of the reactants along its principal diagonal and is zero otherwise.

Our analysis aims to identify minimal reaction-diffusion networks, defined as the networks with the minimal number of edges $k$ that can form a reaction-diffusion pattern. In the case of 2-node networks, it has been described that minimal reaction-diffusion networks must have $2x2=4$ edges (**Murray, 2003**), and therefore only a completely connected network is allowed. This completely connected 2-node network allows for only two reaction-diffusion topologies: the 'activator-inhibitor system' that forms in-phase periodic patterns, and the 'substrate-depleted model' that forms out-of-phase periodic patterns. Our automated approach takes the following inputs through a graphical user interface: the number of network nodes $N$, constraints on $J$ and $D$ including reaction or diffusion rates set to zero, and the number of regulatory interactions $k$. This last parameter defines the number of edges that each network should have with an upper bound of $NxN$ edges representing a completely connected network (**Figure 1a**). This parameter also defines the number of possible networks that are analyzed by the software, which is calculated according to

$$\binom{NxN}{k} = \frac{NxN!}{k!(NxN-k)!}$$

This number represents the possible subsets of size $k$ that can be taken from $J$ and corresponds to the number of possible networks of size $k$.

An important part of the automated high-throughput mathematical analysis is the derivation of the characteristic polynomial, a mathematical expression that determines the stability of the reaction-diffusion system, which is calculated from the determinant of a matrix that combines $J$ and $D$, the 'wave number' $q$, and the eigenvalue $\lambda$. For 3-node networks, the characteristic polynomial has the form

$$\lambda^3 + \lambda^2 a_1 + \lambda a_2 + a_3 = 0$$

where $\lambda$ is the eigenvalue associated with the reaction-diffusion system, and the coefficients $a_1$, $a_2$ and $a_3$ are polynomials formulated in terms of the elements of $J$, $D$ and $q$ (see Appendix 1). The eigenvalue $\lambda$ determines the stability of the network: negative real solutions of $\lambda$ represent a system that is stable around its steady state, while a positive real solution of $\lambda$ represents an unstable system. The variable $q$ that appears in the coefficients $a_1$, $a_2$ and $a_3$ is the wave number that is introduced by the linear stability analysis and is multiplied for $D$. For values $q>0$, this parameter defines the periodicity of the reaction-diffusion pattern. Step 4 of our pipeline entails finding the ranges of the reaction parameters in $J$ and diffusion parameters in $D$ for each network, for which the solutions $\lambda$ are all negative when $q=0$. Similarly, Step 5 requires finding parameter intervals, for which at least one solution $\lambda$ has a positive real part when $q>0$. For characteristic polynomials of degree higher than 2, this is usually done by using the Routh-Hurwitz stability criterion (**Murray, 2003**), a mathematical theorem that finds the necessary and sufficient condition for all negative roots in terms of the polynomial coefficients $a_1$, $a_2...a_n$. However, as the number of network nodes $N$ increases, finding these parameter intervals becomes challenging and tedious because the coefficients $a_1$, $a_2...a_n$ are also complex polynomials of high degree in $q$. We used a computer algebra system to automatically derive and analyze the Routh-Hurwitz criterion in terms of the coefficients $a_1$, $a_2...a_n$. Finally, Step 6 requires to evaluate which of the $2^k$ possible topologies that exist for a given network are compatible with the pattern-forming conditions derived in Step 5 (see Appendix 1 for details).

The complete analysis of minimal networks is limited by the existence of analytical solutions. According to the Abel-Ruffini theorem, there is no general algebraic solution for systems with more than four nodes. However, in practice many five-node networks can be solved if the constraints specified in the input of RDNets lead to a simplification of the coefficients of the characteristic polynomial, as is the case for the five-node digit patterning network (**Figure 4**). Analytical approaches become also challenging when further diffusible nodes are added and when minimal models are extended with additional interactions.

## Robustness calculation

We analyzed the robustness of the networks by calculating the volume of the parameter space that respects the pattern-forming condition in relation to the unit length multidimensional space of all the possible parameter values. This robustness measure corresponds to the probability of randomly picking pattern-forming parameters. The pattern-forming parameter volume is calculated with a

multiple integral of the pattern-forming conditions over all the parameters of the reaction-diffusion networks, in the form

$$\iiint_{l_1} \cdots \iiint_{dl_1} f(k_1, \ldots k_{NxN}, d_1 \ldots d_N) \, dk_1 \ldots dk_{NxN} dd_1 \ldots dd_N$$

where $k_1 \ldots k_{NxN}$ are the reaction parameters and $d_1 \ldots d_n$ the diffusion parameters, $f(k_1 \ldots k_{NxN}, d_1 \ldots d_n)$ are the pattern-forming conditions of the networks, and $l_1$, $l_2$ and $dl_1$, $dl_2$ are the limits of reaction and diffusion variables that are set respectively to (-0.5, 0.5) and (0, 1) representing a multidimensional parameter space of unit side length.

### Graph-theoretical formalism

To investigate the topological basis of Type I, Type II, and Type III networks, we developed a new theoretical framework based on graph theory that can be used to rewrite the pattern-forming conditions in terms of network feedbacks rather than their reaction rates. Further details of this theory are provided in Appendix 2.

### Graphical user interface of RDNets and specification of qualitative and quantitative constraints

Our web-based software RDNets was written in Mathematica (Wolfram Research Inc., Champaign, Illinois) and is available at http://www.RDNets.com. RDNets requires only the installation of the freely available Wolfram CDF player plugin (http://www.wolfram.com/cdf-player/). A simple graphical interface can be used to specify inputs and constraints and to run the high-throughput mathematical analysis. Constraints can be specified by clicking on the nodes or edges of the networks, or by providing specific values for the corresponding parameters (see User Guide available at http://www.RDNets.com). These constraints are automatically translated into mathematical formulae that are coupled with the symbolic linear stability analysis performed by the computer algebra system. The graphical user interface can be used to explore and simulate the list of reaction-diffusion topologies given as output of the linear stability analysis. Additional constraints can be progressively added to the analysis to refresh the output and to narrow down the list of candidate topologies (see User Guide available at http://www.RDNets.com).

## Acknowledgements

This work was supported by EMBO and Marie Curie Long-Term Fellowships to LM; a Sinergia grant (Swiss National Science Foundation) to XD; a Plan Nacional grant from MINECO and support from ICREA and Severo Ochoa to JS; an ERC Starting Grant, an HFSP Career Development Award and support from the Max Planck Society to PM.

## Additional information

### Funding

| Funder | Grant reference number | Author |
| --- | --- | --- |
| European Molecular Biology Organization | EMBO Long-Term Fellowship ALTF 433-2014 | Luciano Marcon |
| Schweizerischer Nationalfonds zur Förderung der Wissenschaftlichen Forschung | Sinergia grant CRSII3_141918 | Xavier Diego |
| Ministerio de Economía y Competitividad | Plan Nacional grant BFU2015-68725-P | James Sharpe |
| Institució Catalana de Recerca i Estudis Avançats | | James Sharpe |
| Severo Ochoa | SEV-2012-0208 | James Sharpe |
| European Research Council | ERC Starting Grant 637840 (QUANTPATTERN) | Patrick Müller |

| Human Frontier Science Program | Career Development Award CDA00031/2013-C | Patrick Müller |
|---|---|---|
| Max-Planck-Gesellschaft (Max Planck Society) | | Patrick Müller |

The funders had no role in study design, data collection and interpretation, or the decision to submit the work for publication.

## Author contributions

LM, Conception and design, Acquisition of data, Analysis and interpretation of data, Drafting or revising the article; XD, Analysis and interpretation of data, Drafting or revising the article; JS, Drafting or revising the article; PM, Conception and design, Analysis and interpretation of data, Drafting or revising the article

## Author ORCIDs

Luciano Marcon, http://orcid.org/0000-0003-0957-9170
Patrick Müller, http://orcid.org/0000-0002-0702-6209

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

# Appendix 1: Details of the automated linear stability analysis

We consider a reaction-diffusion system of the form

$$\frac{\partial \mathbf{c}}{\partial t} = \mathbf{f}(\mathbf{c}) + \mathbf{D}\nabla^2 \mathbf{c} \tag{1}$$

$$\bar{\mathbf{n}} \cdot \nabla \mathbf{c} = 0 \, \text{on} \, \partial\mathbf{\Omega} \tag{2}$$

where $\mathbf{c}$ is a vector of $N \geq 2$ reactant concentrations, $\mathbf{f}$ represents the reaction kinetics, and $\mathbf{D}$ is a diagonal matrix of diffusion coefficients greater than or equal to zero. **Equation (2)** represents zero-flux boundary conditions, where $\partial\mathbf{\Omega}$ is the closed boundary of the spatial domain $\mathbf{\Omega}$ and $\bar{\mathbf{n}}$ is the outward normal to $\partial\mathbf{\Omega}$. We restrict our analysis to zero flux boundary conditions because we are interested in deriving the analytical conditions required to form a self-organizing spatial pattern in the absence of pre-existing asymmetries or external inputs.

To derive the pattern formation conditions of a reaction-diffusion system of size $N$, we use a computer algebra system to automatically build a list of $N$ reactants ($l$), a reaction Jacobian matrix of size $N$x$N$ ($\mathbf{J}$), which represents the partial derivatives evaluated at steady state, and a diagonal diffusion matrix $\mathbf{D}$ of size $N$x$N$ in the form

$$l = \begin{pmatrix} r_1 \\ r_2 \\ \vdots \\ r_N \end{pmatrix} \quad \mathbf{J} = \begin{pmatrix} k_1 & k_2 & \cdots & k_N \\ k_{N_{+1}} & k_{N_{+2}} & \cdots & k_{2N} \\ \vdots & \vdots & \ddots & \vdots \\ k_{N_{-1}xN+1} & k_{N_{-1}xN+2} & \cdots & k_{NxN} \end{pmatrix} \quad \mathbf{D} = \begin{pmatrix} d_{r_1} & \cdots & 0 \\ 0 & d_{r_2} & \cdots & 0 \\ \vdots & \vdots & \ddots & \vdots \\ 0 & 0 & \cdots & d_{r_N} \end{pmatrix}$$

where up to the six-node-case we name the reactants $r_1, r_2, r_3, r_4, r_5,$ and $r_6$ as $\mathrm{u}, \mathrm{v}, \mathrm{w}, \mathrm{z}, \mathrm{h},$ and $r$.

Following classical linear stability analysis (**Murray, 2003**), we derive the stability matrix $\mathbf{F}^{\mathrm{RD}}$ as

$$\mathbf{F}^{\mathrm{RD}} = \mathbf{J} \cdot C - \mathbf{D} \cdot q^2 \tag{3}$$

where $q$ is the wave-number and $C$ is a newly introduced $N$x$N$ connectivity matrix whose elements can only be 0 or 1 and which is used to systematically select subsets of the Jacobian matrix $\mathbf{J}$.

Our software RDNets also takes as input the parameter $k$, which represents the number of edges that will be considered between network nodes. In other words, the number $k$ defines $N - k$ elements of $\mathbf{J}$ that will be set to zero. Finally, the software can also take as input a series of constraints on the elements of $\mathbf{J}$ and $\mathbf{D}$ from qualitative and quantitative experimental data.

## Step 1. Possible networks of size $k$

We first generate a list of possible networks with $k$ edges. This is done by systematically selecting all the possible combinations of $N - k$ elements of $\mathbf{J}$ that can be set to zero. The number of combinations is calculated as the binomial coefficient

$$\binom{N}{N-k} = \frac{N!}{(N-k)!(N-(N-k))!} = \frac{N!}{(k!(N-k)!)} = \binom{N}{k}$$

For each of these combinations we derive a matrix $C$ with $N - k$ elements set to 0 and the remaining elements set to 1. **Appendix 1—figure 1** shows an example of a $C$ matrix and its corresponding network in the case of $N = 3$ and $k = 6$.

**Appendix 1—figure 1.** Example of a matrix $C$ and its corresponding network for $N = 3$ and $k = 6$.

Importantly, we define the minimal size $k$ of a reaction-diffusion network of $N$ nodes as the minimal number of edges $k$ for which at least one of the possible networks can satisfy the requirements for Turing instabilities. In the case of $N = 2$, it is well known that all the elements of **J**, that is $2x2 = 4$, have to be different than zero, and therefore the minimal network size is $k = 4$ (**Murray, 2003**). In other words, all the regulatory edges of 2-node networks have to be present to be able to form a pattern, and therefore there is just one possible network. By analyzing 2-node, 3-node, 4-node, and 5-node networks, we empirically identified the minimal network sizes shown in **Appendix 1—Table 1**.

**Appendix 1—Table 1.** Minimal network sizes for $N = 2, 3, 4, 5$.

| $N$ | 2 | 3 | 4 | 5 |
|---|---|---|---|---|
| minimal $k$ | 4 | 6 | 7 | 8 |

# Step 2. Selecting only strongly connected networks

From the matrices $C$, we construct a list of adjacency directed graphs that represent each network. These graphs can be constructed by deriving the adjacency matrices from $C$ as the transpose $C^T$. Finally, we use an in-built function of the computer algebra system Mathematica to select matrices $C$ that correspond to strongly connected graphs. This filter guarantees that networks with isolated nodes or nodes that solely act as read-outs are discarded (**Appendix 1—figure 2**).

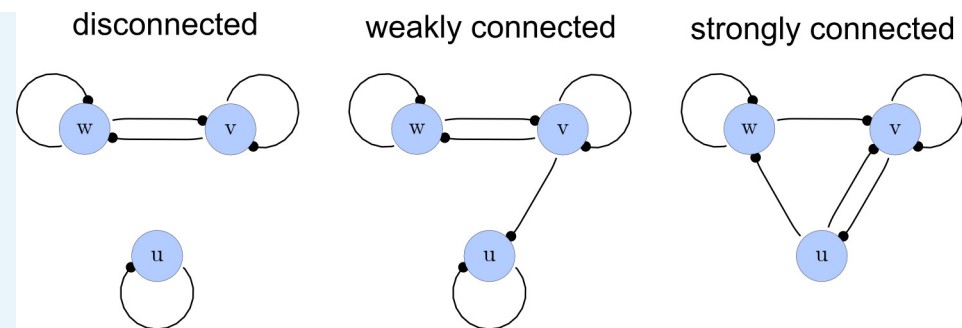

**Appendix 1—figure 2.** Network connections. Left: A disconnected network. Middle: A weakly connected network with a read-out node u. Right: A strongly connected network.

## Step 3. Deletion of symmetric networks

The third step of our automated analysis removes all symmetric networks. Symmetric networks are defined as networks whose graphs are isomorphic. To find isomorphic networks, we extended the in-built isomorphism test function of Mathematica to take into account all reaction or diffusion constraints that may introduce additional differences. An example of a symmetric network and their isomorphisms is given in *Appendix 1—figure 3*.

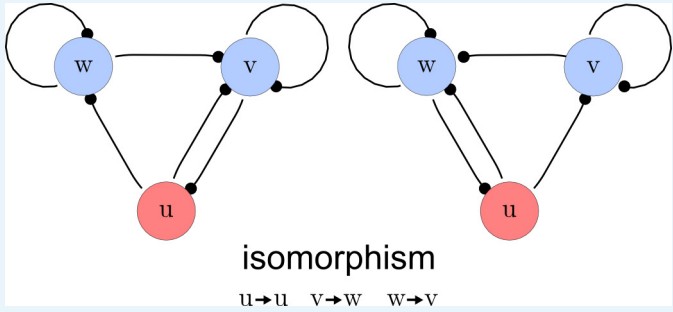

**Appendix 1—figure 3.** Deleting symmetric networks. Two symmetric networks with a corresponding isomorphism that maps equivalent nodes are shown.

## Step 4. Selecting stable networks

This step is a central part of the linear stability analysis and required us to develop a program that derives the characteristic polynomial of a reaction-diffusion system in a symbolic manner. This was done by calculating the determinant of a matrix $A$ defined as

$$A = \lambda I - \mathbf{F}^{\mathrm{RD}} \tag{4}$$

$$\mathrm{Det}(A) = \lambda^N + \lambda^{N-1}a_1 + \lambda^{N-2}a_2 + \cdots + a_N = 0 \tag{5}$$

where $\lambda$ is the eigenvalue and $I$ the identity matrix. The coefficients of the characteristic polynomial $(a_1, \cdots, a_N)$ are also polynomials in terms of $\mathbf{J}, \mathbf{D}$ and $q$. A diffusion-driven instability requires that the solutions of the characteristic polynomial are all negative when $q = 0$ and that there is at least one real positive solution when $q > 0$:

$$\forall \lambda_i, \Re(\lambda_i) < 0 \quad \text{for} \quad q = 0 \tag{6}$$

$$\exists \lambda_i, \Re(\lambda_i) > 0 \quad \text{for} \quad q > 0 \tag{7}$$

As the number of nodes $N$ increases, finding the analytical solution of **Equation (5)** becomes unfeasible. As an alternative, the Routh-Hurwitz stability criterion can be used to derive the necessary and sufficient conditions to satisfy **Equation (6)** by deriving Routh-Hurwitz terms that are obtained by combining the coefficients $(a_1, \cdots, a_N)$. Our software RDNets automatically derives the Routh-Hurwitz terms $\Delta_i$ for the general case of degree $N$. For example, in the case of $N = 4$ the Routh-Hurwitz terms are

$$
\begin{aligned}
\Delta_1 &= a_1 \\
\Delta_2 &= a_1 a_2 - a_3 \\
\Delta_3 &= -a_4 a_1^2 + a_2 a_3 a_1 - a_3^2 \\
\Delta_4 &= a_4 \left( -a_4 a_1^2 + a_2 a_3 a_1 - a_3^2 \right)
\end{aligned}
\tag{8}
$$

The Routh-Hurwitz stability criterion states that all eigenvalues have a negative real part if and only if $\Delta_i > 0$ for i=1 to $N$. As mentioned above, the coefficients $a_1$ to $a_4$ are polynomials in terms of **J**,**D** and $q$, which quickly become complex as $N$ increases. With $N = 4$, the general **J**,**D** matrices, the characteristic polynomial and the corresponding coefficients are

$$
\mathbf{J} = \begin{pmatrix} k_1 & k_2 & k_5 & k_{10} \\ k_3 & k_4 & k_6 & k_{11} \\ k_7 & k_8 & k_9 & k_{12} \\ k_{13} & k_{14} & k_{15} & k_{16} \end{pmatrix} \quad
\mathbf{D} = \begin{pmatrix} d_u & 0 & 0 & 0 \\ 0 & d_v & 0 & 0 \\ 0 & 0 & d_w & 0 \\ 0 & 0 & 0 & d_z \end{pmatrix}
$$

$$
\begin{pmatrix}
q^2 d_u - k_1 + \lambda & -k_2 & -k_5 & -k_{10} \\
-k_3 & q^2 d_v - k_4 + \lambda & -k_6 & -k_{11} \\
-k_7 & -k_8 & q^2 d_w - k_9 + \lambda & -k_{12} \\
-k_{13} & -k_{14} & -k_{15} & q^2 d_z - k_{16} + \lambda
\end{pmatrix}
$$

$$\lambda^4 + \lambda^3 a_1 + \lambda^2 a_2 + \lambda a_3 + a_4 = 0 \tag{9}$$

$$
a_4 = \begin{aligned}
& q^8 \;\; (d_u d_v d_w d_z) & + \\
& q^6 \;\; (k_4(-d_u)d_w d_z - d_v(d_w(k_{16}d_u + k_1 d_z) + k_9 d_u d_z)) & + \\
& q^4 \;\; (k_9 k_{16} - k_{12}k_{15})d_u d_v + \cdots + (k_4 k_9 - k_6 k_8)d_u d_z) & + \\
& q^2 \;\; (d_z((k_1 k_6 - k_3 k_5)k_8 + k_4(k_5 k_7 - k_1 k_9)) + \cdots + k_2(k_3 k_{16} - k_{11}k_{13})) & + \\
& \quad\; -k_5 k_8 k_{11} k_{13} + k_2 k_9 k_{11} k_{13} + k_3 k_9 k_{10} k_{14} + \cdots + k_3(k_5 k_8 - k_2 k_9)k_{16} &
\end{aligned}
$$

$$
a_3 = \begin{aligned}
& q^6 \;\; (d_u(d_v(d_w + d_z) + d_w d_z) + d_v d_w d_z) & + \\
& q^4 \;\; (-d_v((k_9 + k_{16})d_u + (k_1 + k_9)d_z + \cdots) + \cdots - d_w((k_1 + k_4)d_z + \cdots)) & + \\
& q^2 \;\; (-k_6 k_8 d_u + k_5 k_7(-d_v) + k_1 k_9 d_v + k_4 k_9 d_u + \cdots - d_v k_{12}k_{15}) & + \\
& \quad\; (k_3 k_5 k_8 + k_1 k_6 k_8 + k_9 k_{10} k_{13} - k_5 k_{12} k_{13} + \cdots + k_1 k_{12} k_{15} - k_8 k_{11} k_{15}) &
\end{aligned}
$$

$$
a_2 = \begin{aligned}
& q^4 \;\; (d_u(d_v + d_w + d_z) + d_v(d_w + d_z) + d_w d_z) & + \\
& q^2 \;\; (-(k_1 + k_4 + k_{16})d_w + \cdots + (k_4 + k_9 + k_{16})(-d_u)) & + \\
& \quad\; (-k_2 k_3 - k_5 k_7 - k_6 k_8 + k_4 k_9 - k_{10} k_{13} + \cdots + (k_4 + k_9)k_{16}) &
\end{aligned}
$$

$$a_1 = q^2 \; (d_u + d_v + d_w + d_z) \; + \\ -k_1 - k_4 - k_9 - k_{16}$$

It is evident that even in the case with $N = 4$ the coefficients have complex formulae. Moreover, when substituted in *Equation (8)*, they give rise to further complex Routh-Hurwitz terms of higher order in $q$. The stability conditions are derived when $q = 0$, which simplifies the coefficients. However, even in this case the manual derivation of the conditions that make all Routh-Hurwitz terms positive is challenging. We therefore automated these tedious algebraic calculations using the computer algebra system in Mathematica.

## Step 5. Selecting unstable networks with diffusion

In the previous section we showed that the conditions for the stability of a reaction network can be derived by imposing that all the Routh-Hurwitz determinants are positive at $q = 0$. Conversely, the requirements for the existence of diffusion-driven instabilities are obtained by imposing that at least one Routh-Hurwitz determinant turns negative when diffusion is considered: $\exists i, |\Delta_i(q)| < 0$ for some $q > 0$.

In addition, the Routh-Hurwitz theorem allows – by checking which of the determinants turn negative – to derive the conditions that lead to stationary Turing patterns or oscillatory patterns. This can be done by considering the necessary and sufficient conditions for the absence of diffusion-driven instabilities presented in previous studies (*Kellog, 1972*; *Cross, 1978*; *Othmer, 1982*). A violation of any of these conditions is necessary and sufficient for the existence of a single real eigenvalue crossing the right half plane and thus for the formation of a stable Turing pattern. This occurs if and only if $\Delta_N(q) < 0$ for $q > 0$, which can be rewritten as the simpler equivalent condition $a_N(q) < 0$ and $a_k(q) > 0$ for $q > 0$ and for all $k$ different than $N$, by taking into account the identities between the $a_k$ coefficient and $k_{th}$ elementary symmetric functions of the eigenvalues (*Horn and Johnson, 1990*). Since in this work we are interested in the analysis of stable Turing patterns, we can use these conditions to select for networks that produce a stable pattern and filter out those that produce oscillatory patterns.

As shown in the previous section, these conditions generally lead to complex formulae. Non-diffusible factors are associated to vanishing entries in the diagonal of $\mathbf{D}$, which reduces the complexity of the equations. Nevertheless, the negativity conditions of Routh-Hurwitz determinants (see for example $\Delta_3$ in *Equation [8])* remain tedious to derive. However, the algebraic calculations are automated with the computer algebra system in RDNets.

The last step of the analysis consists of finding when the stability conditions and the instability conditions for the existence of Turing patterns are fulfilled simultaneously. Importantly, the computer algebra system allows to handle the combinatorial explosion of algebraic cases systematically in moderate computational time.

## Step 6. Analysis of possible network topologies

The networks derived in Step 5 represent connectivities between nodes (matrix $C$) associated with coefficients of the characteristic polynomial that can satisfy the pattern-forming conditions. These conditions are written in terms of $J$ and $D$ but do note make any explicit assumption on whether elements of the reaction matrix $J$ must have a positive (representing activation) or a negative (representing inhibition) value. We define a network topology as a set of signs associated with reaction rates of $J$ whose correspondent elements in $C$ are set to one. Given a network of size $k$, there are $2^k$ possible topologies that encode all the possible combinations of positive and negative signs for a given matrix $C$. Reaction-diffusion topologies are topologies that can satisfy the pattern forming conditions. Our high-

throughput analysis of minimal 3-node and 4-node signaling networks reveals that every unconstrained network is associated with a set of topologies that exhaustively determine all possible in-phase and out-of-phase periodic patterns between network nodes (*Appendix 1—figure 4*).

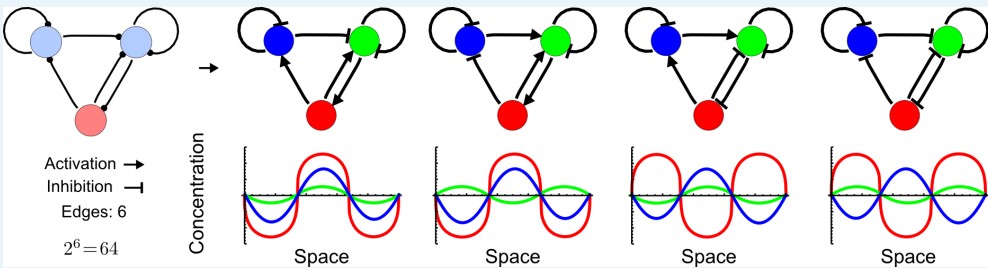

**Appendix 1—figure 4.** Network topologies determine all possible in- and out-of-phase periodic patterns. The shown 3-node network with two diffusible (blue) nodes and one non-diffusible (red) node has 6 edges and therefore $2^6 = 64$ topologies. Four of the 64 possible topologies are reaction-diffusion systems and represent all possible in-phase and out-of-phase pattern between the nodes.

## Optional step: Pattern phase prediction

RDNets can analyze the list of networks topologies to select for a specific phase of the periodic patterns. This feature requires to perform both analytical and numerical computations that derive the relative sign of the eigenvectors associated with a network topology. First, a random set of parameters that satisfies the pattern forming conditions is obtained for each network topology using an in-built function of the computer algebra system. The parameters are substituted into the stability matrix (**Equation [3]**) to leave $q$ as the only free parameter, and the solutions of the characteristic polynomial $\lambda$(**Equation [5]**) are calculated. The solution $\lambda_r$ without a complex part is selected, and its dispersion relation is analyzed to identify $q_{max}$ for which the eigenvalue $\lambda$ is maximum using a gradient ascend numerical method.

Finally, $q_{max}$ is substituted into $q$ in **Equation (3)**, and the eigenvectors of the matrix are calculated (a list of eigenvectors for each solution $\lambda$). Each component of the eigenvector associated with $\lambda_r$ represents a reactant, and the relative sign between them represents the phase of the patterns. Reactants with eigenvector components of the same sign will be in-phase, while reactants with eigenvector components of opposite sign will be out-of-phase.

# Appendix 2: Graph-theoretical formalism to analyze network topologies

Our high-throughput mathematical analysis reveals that different network topologies determine pattern-forming requirements with three different types of diffusion constraints: Type I requires differential diffusivity, Type II allows for equal diffusivity, and Type III represents conditions independent of specific diffusion rates. To investigate the topological basis of these constraints, we developed a new framework based on graph theory to analyze the pattern-forming conditions in terms of network feedbacks rather than elements of the reaction matrix $J$. This can be done by rewriting the coefficients of the characteristic polynomial in terms of cycles of the graph associated with the reaction matrix. These coefficients can be used to derive the Routh-Hurwitz determinants and thus the stability and instability conditions in terms of cycles.

## Analyzing pattern-forming conditions in terms of network feedbacks

As mentioned in Appendix 1, a diffusion-driven instability occurs when the characteristic polynomial in *Equation (5)* has all negative real solutions with $q = 0$ and further has at least one positive real solution when $q > 0$. Given the Routh-Hurwitz conditions, the necessary and sufficient conditions to respect these requirements can be formulated in terms of the coefficients $(a_1, \cdots, a_N)$ of the characteristic polynomial. In the following, we derive a new graph-theoretical formalism to express the coefficients $a_k$ in terms of cycles of the graph associated with the reaction matrix $\mathbf{J}$.

Let $\gamma_k = \{i_1, ..., i_k\}$ be a sequence of $k$ distinct integers such as $1 \leqslant i_1 < i_2 .... < i_k \leqslant N$, and let $S_k^N$ be the set of all the different $\gamma_k$ sequences of $k$ elements in $\{1, ..., N\}$. $\mathbf{F}^{RD}(\gamma_k)$ denotes the $k$-by-$k$ principal submatrix of $\mathbf{F}^{RD}$ given by the coefficients with row and column indices equal to $\gamma_k = \{i_1, ..., i_k\}$. There are $N!/(N-k)!k!$ different $k$-by-$k$ principal submatrices $\mathbf{F}^{RD}(\gamma_k)$, which are in a one-to-one correspondence with all the different sequences $\gamma_k$ in $S_k^N$. The sum of the determinants of all the different $k$-by-$k$ principal submatrices is denoted by

$$E_k(\mathbf{F}^{\mathrm{RD}}) = \sum_{\gamma_k \subseteq S_k^N} \det[\mathbf{F}^{\mathrm{RD}}(\gamma_k)] \tag{10}$$

The following identity, which can be verified using the Laplace expansion of the determinant (*Horn and Johnson, 1990*), expresses the coefficients of the characteristic polynomial in *Equation (5)* in terms of $E_k$

$$a_k = (-1)^k E_k(\mathbf{F}^{\mathrm{RD}}) \qquad k = 1, ..., N \tag{11}$$

The new formulation of the coefficients $a_k$ of the characteristic polynomial can be expanded to partially uncouple the contribution of the diffusion and reaction terms:

$$a_k = (-1)^k \sum_{\gamma_k \subseteq S_k^N} \det[\mathbf{J}(\gamma_k) - \mathbf{D}(\gamma_k)q^2] \tag{12}$$

Uncoupling of the diffusion and reaction contributions is achieved by expanding the minors of order $k$ in *Equation (12)* and grouping the resulting terms according to the number of entries from the diffusion matrix. In this way, each minor $\det[\mathbf{F}^{\mathrm{RD}}(\gamma_k)]$ is expressed as a summary of products of all possible minors $\det[\mathbf{J}(\gamma_m)]$ of order $m \leqslant k$ given by the sequences $\gamma = \{i_1, ..., i_m\}$

multiplied by the $k - m$ coefficients $(d_{j_1} \cdot d_{j_2} \dots \cdot d_{j_{k-m}})$ given by the complementary set $\bar{\gamma} = \{j_1, \dots, j_{k-m}\}$ in $\mathbf{D}$. The subsets $\bar{\gamma}_m$ and $\gamma_m$ are complementary sets in $\gamma_k$ in the sense that $\gamma_m \cap \bar{\gamma}_m = \emptyset$ and $\gamma_m \cup \bar{\gamma}_m = \gamma_k$. Then, after some tedious algebra, the following expression is reached:

$$a_k = \sum_{\gamma_k} \left\{ (-1)^k \det[\mathbf{J}(\gamma_k)] + \sum_{m=1}^{k-1} q^{2(k-m)} \sum_{\gamma_m \subset \gamma_k} (-1)^m \det[\mathbf{J}(\gamma_m)] \det[\mathbf{D}(\bar{\gamma}_m)] \right.$$
$$\left. + q^{2k} \det[\mathbf{D}(\gamma_k)] \right\}$$

(13)

The first and third summands in **Equation (13)** stem exclusively from reaction and diffusion terms. The second is formed by minors of the reaction matrix weighted by the complementary coefficients of the diffusion matrix. Next, we show how determinants can be formulated in terms of cycles of the reaction graph associated with a matrix. This allow us to reformulate **Equation (13)** in terms of feedbacks of the reaction-diffusion network.

The definition of the reaction and interaction graphs closely follows the definition of the Coates graph of a general square matrix (**Brualdi and Cvetkovic, 2008**). The reaction graph $G_R[\mathbf{J}]$ is a labeled, weighted, directed graph associated to the linearization of the reaction-diffusion system. In a system with $N$ interacting species, $G_R[\mathbf{J}]$ is a graph with $N$ nodes that has a directed edge from node $j$ to node $i$ if $\mathbf{J}_{ij} \neq 0$. The weight assigned to the edge is the coefficient $\mathbf{J}_{ij}$. Note that according to this definition, the entries $\mathbf{J}_{ii}$ in the diagonal of the reaction matrix have associated an edge with $i$ as the initial and terminal node. These type of edges are called loops and account for decay terms and autocatalysis in the reaction. The interaction graph $G_I[\mathbf{F}^{RD}]$ is the equivalent of the reaction graph including the diffusion term in $\mathbf{F}^{RD}$. If the diffusion matrix is diagonal, both graphs are topologically identical and the only difference lies in the weight of the loops. The following $4 \times 4$ matrix $\mathbf{A}$ is used as an example to illustrate these definitions:

$$\mathbf{A} = \begin{bmatrix} l_1 & -b & +c & +d \\ 0 & l_2 & 0 & 0 \\ -e & 0 & l_3 & 0 \\ 0 & 0 & +f & l_4 \end{bmatrix}$$

(14)

According to the definitions given previously, $G_R[\mathbf{A}]$ is the 4-node graph shown in **Appendix 2—figure 1**.

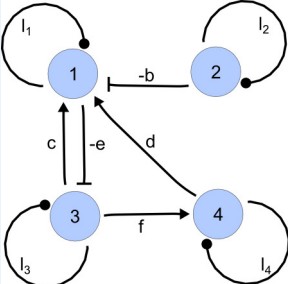

**Appendix 2—figure 1.** Interaction graph associated with matrix $\mathbf{A}$.

The definitions from graph theory introduced next will be necessary to develop the framework for the analysis of the stability of a reaction-diffusion system. The indegree and the outdegree of a node are the number of edges that have this node as the initial or terminal node, respectively. A loop, defined as an edge that originates and ends at the same node, contributes 1 to both the indegree and the outdegree of that node. As an example, in the

graph of *Appendix 2—figure 1*, node $3$ has indegree equal to $2$, for it has an incoming edge from node 1 and the loop. The outdegree of this node is $3$, because there are two edges going to nodes $1$ and $4$ plus the loop contribution (*Appendix 2—figure 2*).

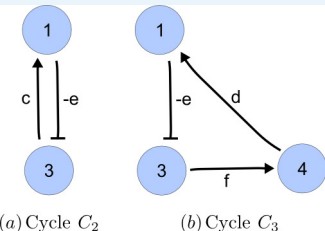

(a) Cycle $C_2$ (b) Cycle $C_3$

**Appendix 2—figure 2.** Cycles of length $m > 1$ in $G_R[\mathbf{A}]$.

A cycle of length $m$ is a subset of $m$ distinct nodes and $m$ distinct edges that join $i_k$ to $i_{k+1}$ for $k = 1, ..., m$ and an edge from $i_m$ back to $i_1$. By this definition, loops are also cycles of length one. The weight of a cycle $w(c)$ is the product of weights of the edges that form the cycle. Cycles are classified as positive or negative according to the sign of their weight. The graph $G_R[\mathbf{A}]$ used as an example has, aside from the four loops associated to the diagonal terms in $\mathbf{A}$, a negative cycle of length 2 and a negative cycle of length 3. $C_2$ is negative and its weight is $w(C_2) = -e \cdot c$, whereas $C_3$ has weight $w(C_3) = -e \cdot f \cdot d$ and is also a negative cycle (*Appendix 2—figure 2*).

A subgraph of the reaction graph is a directed graph formed by a subset of edges and whose set of nodes are a subset $\gamma_k = \{i_1...i_k\}$ of those in $G_R$, with $\gamma_k \subset \{1 ... N\}$. The induced subgraph of $\gamma_k$, referred as the I-subgraph $I_{\gamma_k}$, is the subgraph of $G_R[\mathbf{A}]$ formed by the subset of nodes $\gamma_k$ and all the edges that join nodes within this set. The induced subgraph $I_{\gamma_k}$ is identical to the graph $\mathrm{G_R}[\mathbf{A}(\gamma_k)]$ obtained by applying the definition of the reaction graph to the principal submatrix $\mathbf{A}(\gamma_k)$, so that all the definitions and properties of the reaction graph carry over its I-subgraphs. As an example, consider the $3$-by-3 principal submatrix matrix $\mathbf{A}(\gamma_3)$ induced by the sequence $\gamma_3 = \{1, 2, 4\}$:

$$\mathbf{A}(\gamma_3) = \begin{bmatrix} l_1 & -b & d \\ 0 & l_2 & 0 \\ 0 & 0 & l_4 \end{bmatrix} \tag{15}$$

The I-subgraph associated with $\gamma_k$ (*Appendix 2—figure 3*) is obtained by applying the definition of the reaction graph to the matrix $\mathbf{A}(\gamma_3)$ or, equivalently, by erasing from the complete graph $G_R[\mathbf{A}]$ the nodes that do not belong to $\gamma_3$ and the edges that do not start and finish in the nodes of $\gamma_3$. Note that the induced subgraphs of the interaction graph $G_I[\mathbf{F}^{\mathrm{RD}}(\gamma_k)]$ obtained by considering all possible sequences $\gamma_k$ are in a one-to-one correspondence with $\mathbf{F}^{\mathrm{RD}}(\gamma_k)$, the $k$ x $k$ principal submatrices appearing in the expansion of the $k$-th coefficient of the characteristic polynomial in *Equations (10)* and *(11)*. Likewise, all the terms $\mathbf{J}^{\mathrm{R}}(\gamma_k)$ in *Equation (13)* of the coefficient $a_k$ correspond to one and only one I-subgraph of the reaction graph $G_I[\mathbf{J}^{\mathrm{R}}]$. Hence, the graph definitions introduced previously provide a method to associate a graph to each of the terms in the algebraic stability conditions.

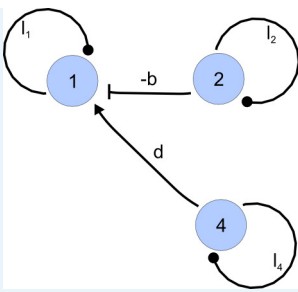

**Appendix 2—figure 3.** I-subgraph induced by $\mathbf{A}(\gamma_3)$.

A spanning subgraph is a subgraph that includes all the nodes in $G_R$, but not necessarily all edges. A linear spanning subgraph $\ell$, also referred to as an L-subgraph, is a spanning subgraph of $G_R$, in which each node has indegree 1 and outdegree 1. This definition implies that an L-subgraph is composed by a set of disjoint cycles and isolated loops, where the cycles are disjoint in the sense that each node belongs to one and only one cycle. The three different L-subgraphs contained $G_R[\mathbf{A}]$ are depicted in **Appendix 2—figure 4**.

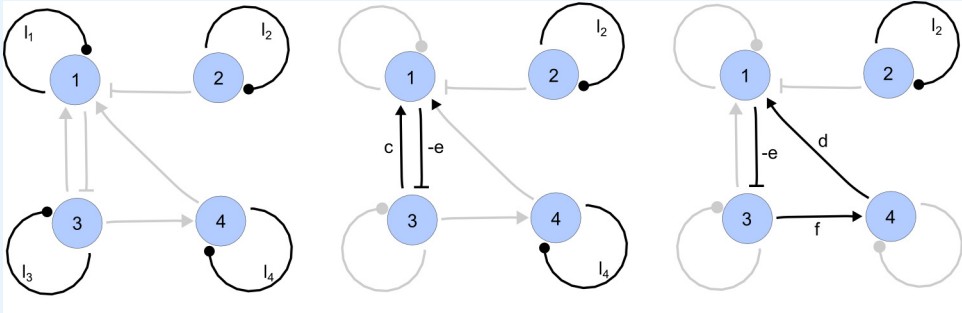

$(a)$ L-subgraph $\ell_1$      $(b)$ L-subgraph $\ell_2$      $(c)$ L-subgraph $\ell_3$

**Appendix 2—figure 4.** Linear spanning subgraphs of $G_R[\mathbf{A}]$.

The number of cycles in a L-subgraph is denoted by $s(\ell)$. The weight of a L-subgraph is simply defined as the product of weights of the cycles contained in it:

$$w(\ell) = \prod_{c \subset \ell} w(c) \tag{16}$$

For the linear spanning subgraphs depicted in **Appendix 2—figure 4**, the number of cycles and weights are:

$$s(\ell_1) = 4 \quad ; \quad w(\ell_1) = (l_1) \cdot (l_2) \cdot (l_3) \cdot (l_4)$$

$$s(\ell_2) = 2 \quad ; \quad w(\ell_2) = (l_2) \cdot (l_4) \cdot (-e \cdot c) \tag{17}$$

$$s(\ell_3) = 2 \quad ; \quad w(\ell_3) = (l_2) \cdot (-e \cdot f \cdot d)$$

The notion of linear spanning subgraphs can be naturally extended to the I-subgraphs of $G_R[\mathbf{A}]$. The L-subgraphs contained in $I_{\gamma_k}$ are all the different subgraphs of order $k$ formed by a set of disjoint cycles spanning the $k$ nodes of the induced subgraph.

The L-subgraphs contained in the reaction graph of a matrix are the factors that determine the stability of the system. It has been shown that the graph methodology provides a way to

assign an I-subgraph to each of the terms appearing in the algebraic equations that determine the stability of the system. Particularly, these equations are expressed in terms of principal minors $\det[\mathbf{J}^{\mathbf{R}}(\gamma_k)]$ of the reaction matrix. Next, it will be shown that the value of each of these minors is determined *only* by the weights of the L-subgraphs contained in the associated I-subgraph $I_{\gamma_k}$. The intimate relationship between the dynamics of a reaction-diffusion system and the cyclical structure of the reaction network is explained by this fact.

The expression of the determinant of an $N$ x $N$ matrix $\mathbf{A}$ as a linear combination of the weights of the L-subgraphs in $G_R[\mathbf{A}]$ is known as the Coates formula:

$$\det[\mathbf{A}] = (-1)^N \sum_{\ell \subseteq G_R} (-1)^{s(\ell)} w(\ell) \tag{18}$$

where the sum goes through all the L- subgraphs in $G_R[\mathbf{A}]$. A sketch of the proof of the Coates formula is given following the work of **Chen, 1997** (p. 143), and a more formal proof can be found in **Brualdi and Cvetkovic, 2008** (p. 94). The first part of the proof shows that there is a one-to-one correspondence between the non-vanishing terms in the determinant of a matrix and the linear spanning subgraphs in its associated graph. The second part of the proof shows that the sign of the contribution of the non-vanishing terms is also dictated by the structure the linear spanning subgraphs. The classical definition of the determinant of a $N$ x $N$ matrix is:

$$\det(\mathbf{A}) = \sum_p \varepsilon_{i_1 \ldots i_N} a_{1i_1} \cdot \ldots \cdot a_{Ni_N} \tag{19}$$

where the sum is over all the $N!$ permutations $p = \{1, ..., N\} \rightarrow \{i_1, ..., i_N\}$. The signature of the permutation is given by $\varepsilon_{i_1, \ldots, i_N}$ and is equal to $+1$ if $p$ is an even permutation and equal to $-1$ if it is odd. All non-vanishing terms in **Equation (19)** are a product $a_{1i_1} \cdot \ldots \cdot a_{Ni_N}$ of $N$ coefficients. Each index appears twice, one as a row index and one as a column index, so that each row and column contribute to the product with exactly one coefficient. Hence, the subgraph in $G_I[\mathbf{A}]$ defined by the entries $a_{1i_1} \cdot \ldots \cdot a_{Ni_N}$ has $N$ edges with exactly one edge coming into every node and one edge coming out of every node. Therefore, the subgraph associated to every term in **Equation (19)** is by definition a linear spanning subgraph in $G_I[\mathbf{A}]$.

Conversely, every linear spanning subgraph $\ell$ in $G_I[\mathbf{A}]$ has $N$ nodes with indegree and outdegree equal to one. The $N$ edges in $\ell$ are associated to $N$ coefficients in $\mathbf{A}$, the edge directed to node $j$ being the only one in the $j$-th row, and the edge coming out of node $j$ being the only one in the $j$-th column. Arranging the indexes by increasing row order, the weight of $\ell$ becomes $w(\ell) = a_{1i_1}, ..., a_{Ni_N}$, showing the correspondence between each linear spanning subgraph in $G_I[\mathbf{A}]$ with one and only one of the permutations $p$ in the definition of the determinant. In this way, a one-to-one correspondence has been established between the L-subgraphs in $G_I[\mathbf{A}]$ and the non-vanishing permutations terms in the $det[\mathbf{A}]$. Explicit calculation of the determinant of the example matrix illustrates the first part of the proof, as $\det[\mathbf{A}]$ is proven to be a linear combination of the weights of the L-subgraphs $\ell_1$, $\ell_2$, $\ell_3$ represented in **Appendix 2—figure 4**:

$$\det(\mathbf{A}) = w(\ell_1) - w(\ell_2) + w(\ell_3) \tag{20}$$

The one-to-one correspondence provides a convenient way to label a particular L-subgraph by the associated permutation. The permutation $p = \{i_1, ..., i_N\}$ defines univocally the L-subgraph $\ell(p)$ as the subgraph of $G_I[\mathbf{A}]$ obtained by selecting the edge from node $i_1$ to node 1, from node $i_2$ to node 2 and generally from node $i_k$ to node $k$ for $k = 1, ..., N$. In this way, the sign of the contribution of an L-subgraph to the determinant can be derived considering the signature of its associated permutation. The L-subgraphs $\ell_1$ and $\ell_3$ of the example correspond to the even permutations $p_1 = \{1, 2, 3, 4\}$ and $p_3 = \{4, 2, 1, 3\}$. Thus, the corresponding terms in the determinant must have positive sign, as it is confirmed examining the explicit expression in

*Equation (20)*. Conversely, $\ell_2$ is associated to the odd permutation $p_2 = \{3, 2, 1, 4\}$, and consequently the sign of the corresponding term is negative. In the same way that permutations define univocally a L-subgraph, a cyclic permutation of $k$ integers defines univocally a cycle passing through $k$ nodes in $G_I[\mathbf{A}]$.

The second part of the proof shows how the signature of a permutation $\varepsilon_p$ is related to the structure of the associated L-subgraph; more precisely, the number of cycles $s(\ell)$ contained in it. The parity of a permutation is the number of transpositions in which it can be decomposed. The decomposition is generally not unique, but the parity is invariant, so that permutations are classified as even or odd according to this number. The signature of a transposition is defined as $-1$, and by extension the signature of a permutation is given by the product of the signatures of its factors. Hence, the signature of even permutations is +1, whereas the sign of odd permutations is -1. The theory of symmetric groups of finite degree establishes that for any permutation $p$ there is a unique decomposition of $p$ as a product of $s(p)$ cyclic permutations (*Clarke, 1974*):

$$p = cp_1 \times cp_2 \times ... \times cp_s \tag{21}$$

In turn, any cyclic permutation of $i$ objects can be written as the product of $i - 1$ transpositions. Hence, any permutation of $N$ objects can be factorized as $s(p)$ cyclic permutations of $i, j, k, ...$, objects, with $i + j + k + ... = N$. The signature of the permutation, given by the product of the signatures of the cyclic factors is then $\varepsilon_p = (-1)^{i-1}(-1)^{j-1}(-1)^{k-1}... = (-1)^{N-S(p)}$. Rearranging, the following identity is obtained:

$$\varepsilon_p = (-1)^N (-1)^{s(p)} \tag{22}$$

It has been shown that a permutation $p$ corresponds to an L-subgraph $\ell$, and that the cyclic permutations in $p$ correspond to the cycles in $\ell$. Thus, replacing the permutation $p$ for $\ell$ and the number of cyclic permutations in $p$ for the number of cycles in $s(\ell)$ completes the proof of the Coates formula.

The expression of $\det[\mathbf{A}]$ in *Equation (20)* can now be derived strictly from the graphical structure of the associated graph. Indeed, the sign of the contributions of $\ell_1$ and $\ell_3$ are positive because they contain an even number of cycles (four loops in the former case, one loop and one cycle of length 3 in the latter case), whereas $\ell_2$ contains an odd number of cycles (two loops and a cycle of length 2) and accordingly, its contribution is negative.

The Coates formula leads naturally to the definition of the weight of an induced subgraph as the signed sum of the weights of the L-subgraphs contained in it. According to this definition, the weight of the I-subgraph $I_{\gamma_k}$ is equal to the determinant of the principal submatrix $A(\gamma_k)$:

$$w(I_{\gamma_k}) \equiv \det[\mathbf{A}(\gamma_k)] = (-1)^k \sum_{\ell \subseteq I_{\gamma_k}} (-1)^{s(\ell)} w(\ell) \tag{23}$$

This definition is the last element required to reformulate the stability conditions for a reaction-diffusion system from a graph-theoretical point of view. The coefficient of order $k$ in the characteristic polynomial was expressed in *Equations (10–11)* as a sum over all the principal minors of order $k$ in $\mathbf{F}^{\mathbf{RD}}$. A method to associate a graph $G_I[\mathbf{F}^{\mathbf{RD}}]$ to $\mathbf{F}^{\mathbf{RD}}$ has been established. Particularly, each $k \times k$ principal submatrix $\mathbf{F}^{\mathbf{RD}}(\gamma_k)$ corresponds to an induced subgraph $I_{\gamma_k}$ of order $k$ in the interaction graph. Furthermore, the associated principal minor $[\det \mathbf{F}^{\mathbf{RD}}(\gamma_k)]$ is given by the weight $w(I_{\gamma_k})$ of the associated I-subgraph in $G_I$. Substitution of this identity restates the algebraic expression of $a_k$ given in *Equation (12)* as sum of weights of the induced subgraphs as:

$$a_k = (-1)^k \sum_{I_{\gamma_k} \subseteq G_I} w(I_{\gamma_k}) \tag{24}$$

where the summation goes over all the I-subgraphs of order $k$ in the interaction graph. Expanding the weight of the I-subgraphs in terms of the L-subgraphs according to *Equation (23)* leads to

$$a_k = \sum_{I_{\gamma_k} \subset G_I} \sum_{\ell \subseteq I_{\gamma_k}} (-1)^{s(\ell)} w(\ell) \tag{25}$$

Likewise, substitution of the weights of the I-subgraphs in the reaction graph in *Equation (13)* leads to the graphical counterpart of the uncoupled expressions of $a_k$:

$$a_k = \sum_{I_{\gamma_k} \subset G_R} \left\{ \sum_{\ell \subseteq I_{\gamma_k}} (-1)^{s(\ell)} w(\ell) + \sum_{m=1}^{k-1} q^{2(k-m)} \sum_{I_{\gamma_m} \subset I_{\gamma_k}} \det[\mathbf{D}(\bar{\gamma}_m)] \sum_{\ell' \subseteq \gamma_m} (-1)^{s(\ell')} w(\ell') \right.$$
$$\left. + q^{2k} \det[\mathbf{D}(\gamma_k)] \right\} \tag{26}$$

Examination of these results provides an important insight on the relationship between the feedback structure of the reaction scheme and the stability of the associated reaction-diffusion system. More precisely, the graph-based expressions reveal that every cycle in the network has a defined role in the dynamics of the system. This allows to break down the complete network into smaller functional motifs, thus providing a powerful tool to analyze general networks independently of their complexity or specific parameter values. In the following, we show how this analysis can be applied to the three examples shown in *Figure 2*.

## Cycle analysis of the networks shown in *Figure 2*

In the following, we show how the graph-theoretical formalism can be used to analyze the three networks presented in *Figure 2*. For each network (*Appendix 2—figures 5*, *6*, *7*), we present the coefficients of the characteristic polynomial, stability conditions, and instability conditions in terms of cycles. Finally, we highlight the feedback that provides instability and the trade-off between stabilizing and destabilizing feedbacks that underlies the minimum diffusion ratio $d$.

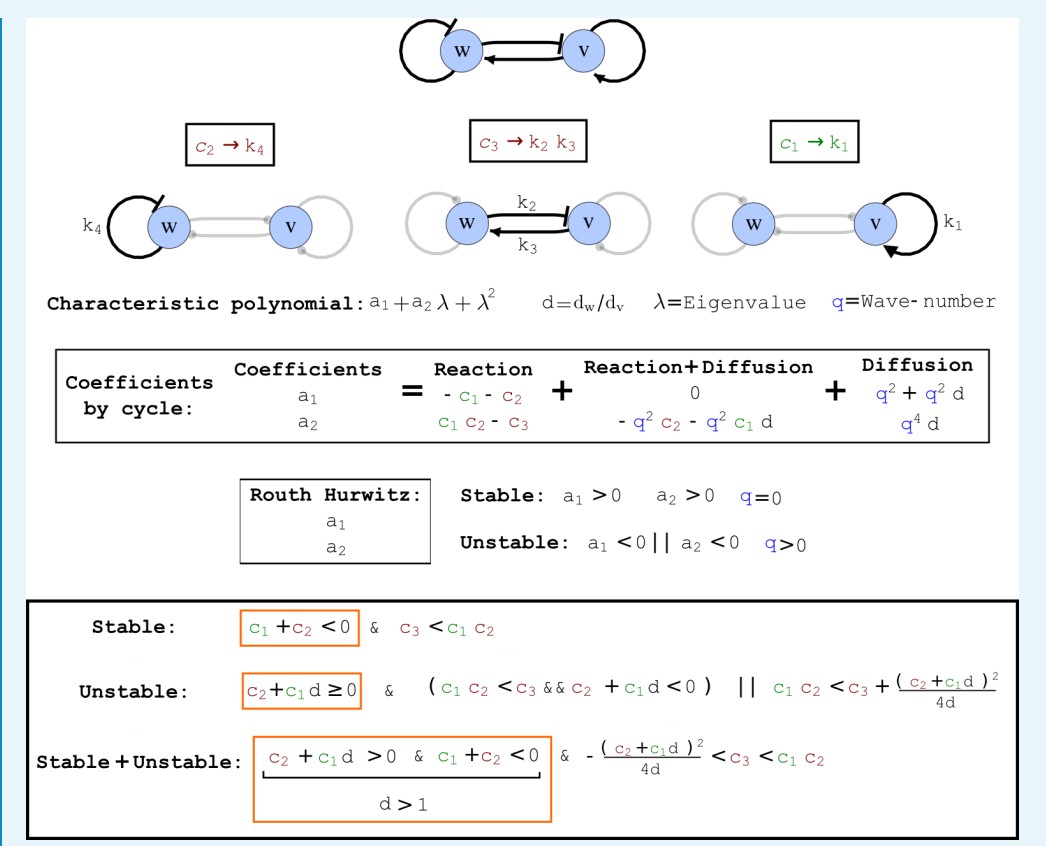

**Appendix 2—figure 5.** Output from RDNets showing the full set of conditions required for pattern formation in the Type I network shown in *Figure 2*, left panel. The trade-off between stabilizing and destabilizing feedbacks that underlies the minimum diffusion ratio *d* is highlighted with orange boxes.

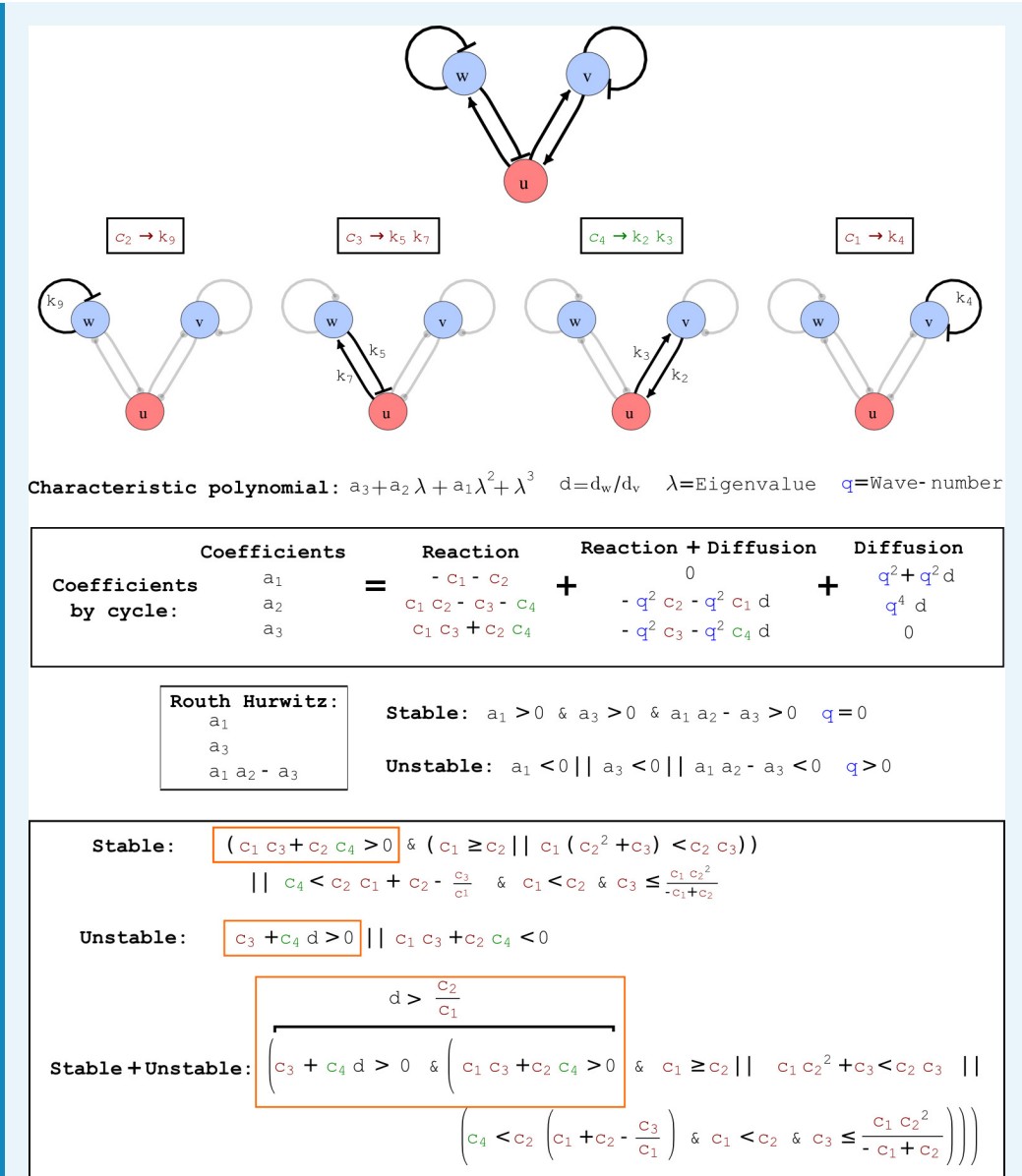

**Appendix 2—figure 6.** Output from RDNets showing the full set of conditions required for pattern formation in the Type II network shown in *Figure 2*, middle panel. The trade-off between stabilizing and destabilizing feedbacks that underlies the minimum diffusion ratio *d* is highlighted with orange boxes.

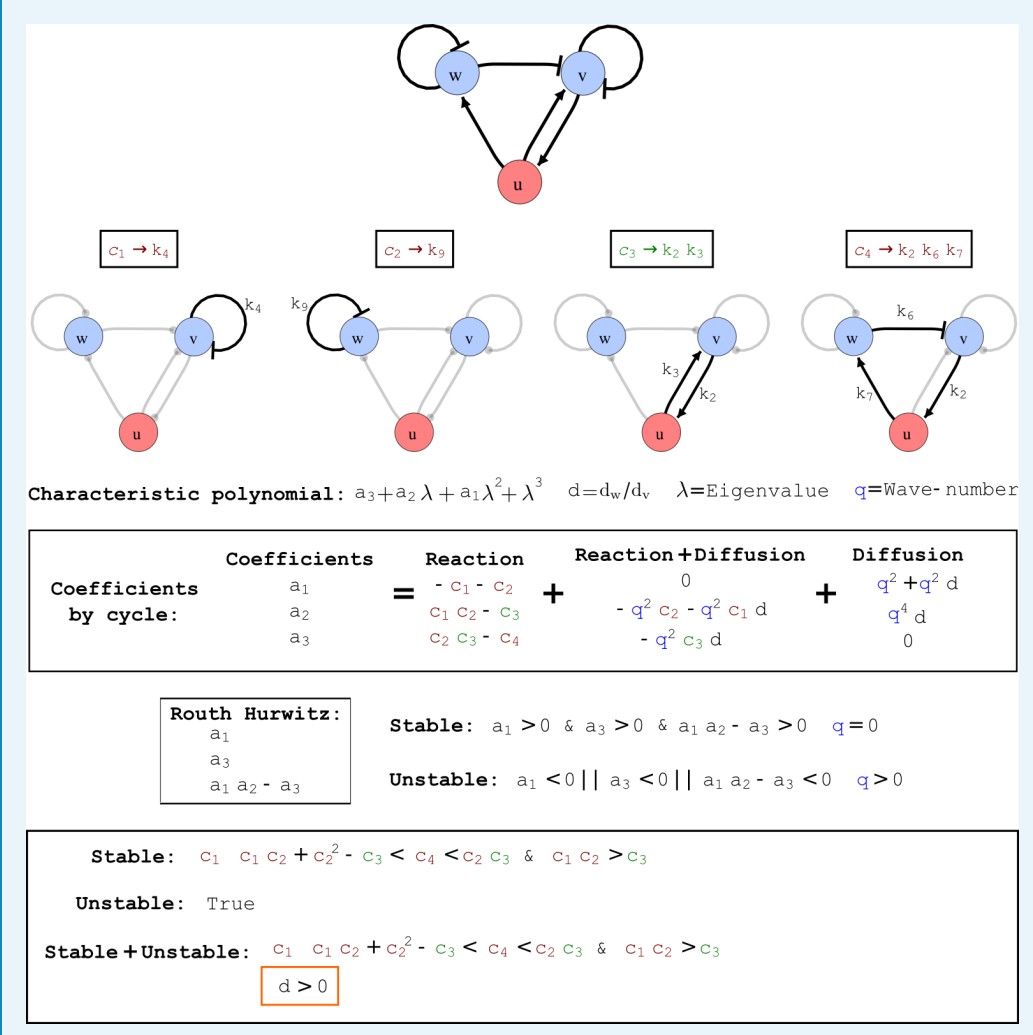

**Appendix 2—figure 7.** Output from RDNets showing the full set of conditions required for pattern formation in the Type III network shown in **Figure 2**, right panel. The trade-off between stabilizing and destabilizing feedbacks that underlies the minimum diffusion ratio $d$ is highlighted with orange boxes.

## Non-diffusible destabilizing cycles and noise amplification

Numerical simulations of the networks identified by our analysis reveal that when a destabilizing cycle comprises only non-diffusible reactants, the reaction-diffusion system does not form a periodic pattern but rather amplifies the noise in the initial conditions. This behavior depends on the specific dispersion relation associated with these systems that shows an asymptotic behavior as the wave-number $q$ increases and leads to an amplification of any of the fluctuations in the initial conditions. Therefore, although these types of reaction-diffusion systems fulfill the Turing instability conditions, they do not amplify a preferential wavelength. This type of behavior has been previously described in reaction-diffusion systems composed of two diffusible reactants and one immobile reactant that activates itself (**White and Gilligan, 1998**; **Klika et al., 2012**). Similar behaviors can be observed for indirect non-diffusible auto-activation implemented for example by two immobile nodes that mutually activate or repress each other. In RDNets, these networks are filtered out by default, but they can be re-included in the analysis by selecting the option 'Noise Amplifying Nets'.

# Appendix 3: Mechanism of self-organizing pattern formation

In this section, we use numerical simulations to investigate the mechanism that underlies pattern formation in Type II and Type III networks and relate our findings to the classical interpretations of reaction-diffusion patterning proposed by Turing and Meinhardt and Gierer. We use two types of numerical simulations throughout the discussion: i) simulations on continuous one-dimensional domains, and ii) simulations on a pair of cells inspired by the original simulation proposed by Turing. The simulations on continuous one-dimensional domains were performed using linear models with cubic saturation terms derived as described in Appendix 6. The simulations on a pair of cells were performed using linear models without saturation terms as originally done by Turing (**Turing, 1952**).

## Previous proposals of self-organizing pattern formation

### The role of differential diffusivity in models based on local auto-activation and lateral inhibition

Self-organizing pattern formation has previously been described as the combination of two processes: local auto-activation and lateral inhibition (LALI) (**Oster, 1988**; **Meinhardt and Gierer, 2000**; **Maini, 2004**; **Newman and Bhat, 2009**; **Green and Sharpe, 2015**) implemented by a poorly diffusive self-enhancing activator and a long-range inhibitor whose main role is to limit the expansion of activation peaks. Lateral inhibition can be implemented in two alternative ways (**Koch and Meinhardt, 1994**): 1) in the activator-inhibitor model, lateral inhibition is implemented by a rapidly diffusing inhibitor that is promoted by the activator and that limits the expansion of activation peaks; 2) in the substrate-depletion model, it is implemented by the consumption of a rapidly diffusing substrate that is required for the self-enhancement of the activator. The schematic representation of the activator-inhibitor model shown in *Appendix 3—figure 1* illustrates how local auto-activation and lateral inhibition are assumed to underlie pattern formation.

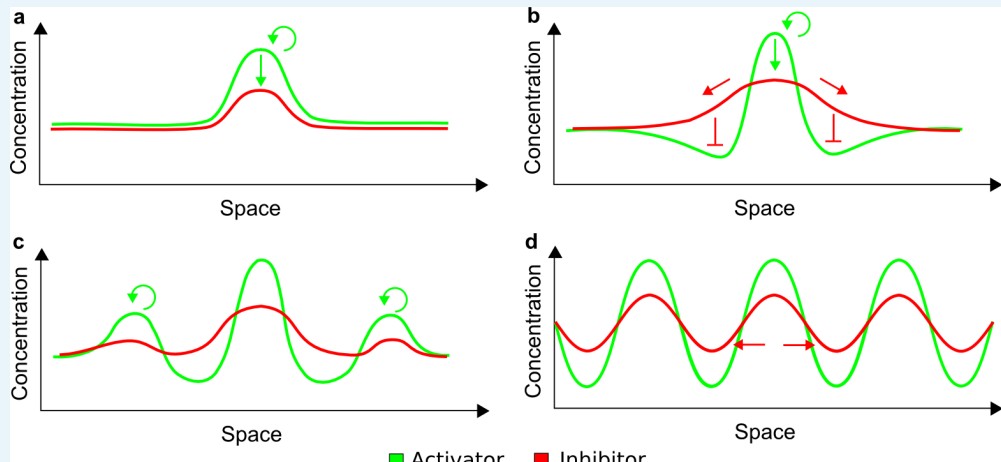

**Appendix 3—figure 1.** Schematic representation of the activator-inhibitor model based on LALI. (**a**) Small random fluctuations in the homogeneous distribution of the activator and the inhibitor give a little advantage to the activator to promote itself and to grow in concentration (green arrows). Since the activator promotes the inhibitor, a higher concentration of the inhibitor is also formed in the same region. (**b**) The inhibitor diffuses more rapidly than the activator and thereby inhibits the formation of other activator peaks in surrounding regions. It also promotes further growth of the activator due to the local decrease of the inhibitor. (**c**) Other activation peaks are formed by the same mechanism. (**d**) The overall process leads to

the formation of periodic patterns, where the different spatial profiles of activator and inhibitor peaks are assumed to reflect the higher diffusivity of the inhibitor (red arrows).

Two implicit assumptions underlie the classical interpretation of the activator-inhibitor model discussed above: i) local auto-activation and lateral inhibition happen in chronological order: first the activator promotes itself, and then the inhibitor diffuses into the neighboring regions to limit the propagation of the activator; ii) the differential diffusivity between activator and inhibitor is a necessary condition to stabilize neighboring regions and to prevent 'an overall auto-catalytic explosion' (*Meinhardt and Gierer, 2000*).

Our study reveals that in the presence of immobile reactants, reaction-diffusion systems can form periodic patterns even when all diffusible reactants have the same diffusivity. In the following, we show that the LALI concept cannot be easily applied to these systems. Our analysis challenges the classical interpretation of activator-inhibitor models and demonstrates that the role of the differential diffusivity is not to stabilize neighboring regions by preventing the formation of activator peaks, but rather to destabilize the system upon spatial perturbations as originally proposed by Turing (*Turing, 1952*). Finally, we show how reaction-diffusion networks with two diffusing reactants and one immobile reactant can become unstable to spatial perturbations even without differential diffusivity of the mobile reactants.

## Relationship of Type II networks to LALI models

The type II network presented in *Figure 2* can form a self-organizing pattern even when all mobile reactants have equal diffusivities. Our analysis based on a graph-theoretical formalism demonstrates that this can be achieved by reaction terms that increase the stability of the homogeneous steady state. In particular, the combination of the conditions for homogeneous steady state stability and instability to spatial perturbations shows that the minimum diffusion ratio $d$ required to form a pattern is defined by the ratio between two stabilizing feedbacks $c_1$ and $c_2$:

$$d > \frac{|c_2|}{|c_1|}, \quad d = \frac{d_w}{d_v} \tag{27}$$

If the stabilizing feedback $c_1$ is stronger than $c_2$, the system can form self-organizing patterns even with equal diffusivity ($d = 1$). These two stabilizing feedbacks correspond to the negative self-regulatory loops of the diffusible nodes v and w. Such negative self-regulatory loops have often been interpreted as decay terms. From the LALI perspective, this suggests the possibility that although the diffusion coefficients of v and w can be equal, their range - i.e. the ratio between diffusion and decay - must be different, such that the decay of the destabilizing node v is greater than the decay of the stabilizing node w:

$$\frac{d_w}{|c_2|} > \frac{d_v}{|c_1|} \tag{28}$$

Within the LALI framework, the network topology suggests that u and v, which mutually activate each other, could behave like the short-range auto-activator, while w could implement the long-range inhibitor. Although v and w have the same diffusivities, w may still behave like a classical long-range inhibitor due to its longer half-life to limit the expansion of the activator. This idea also appears to be consistent with the simulated periodic patterns of v and w that are qualitatively similar to the peaks of an activator and an inhibitor, respectively (*Figure 2c* and *Appendix 6—Table 4*). We note, however, that opposite periodic patterns of v and w can be obtained with the same diffusivities and half-lives but using different kinetic parameters to implement the cycle $c_3$ that connects u and w (*Appendix 3—figure 2*). The LALI concept is therefore not an appropriate framework to describe pattern formation for these networks.

Indeed, the LALI concept requires the subjective definition of a subset of nodes that behaves as the activator and a subset of nodes that behave as the inhibitor with the final aim of identifying different effective ranges (*Miura, 2007*; *Korvasova et al., 2015*). Such effective ranges need to be defined ad hoc for each network and in cases with equally diffusing reactants appear to be just reformulations of the reaction kinetics that do not contribute to the identification of the general principles that underlie pattern formation.

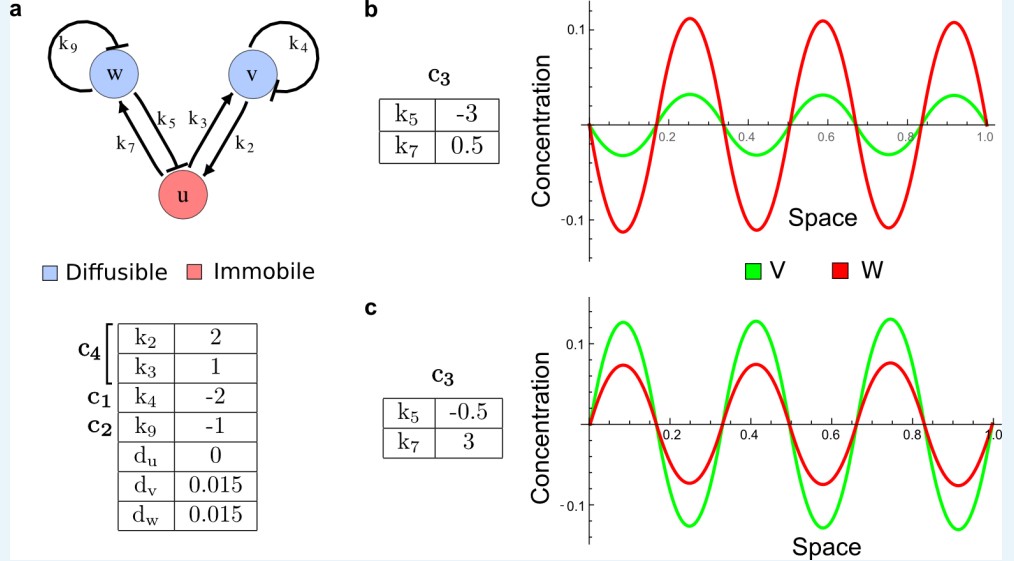

**Appendix 3—figure 2.** Different ranges of the pattern of v and w in the network of **Figure 2c**. (**a**) The Type II network shown in **Figure 2c** (center) with reaction rates that implement the cycles $c_1$, $c_2$, $c_4$ and diffusion rates. (**b–c**) Different reaction rates for the cycle $c_3$ determine different ranges for the peaks of v and w despite their identical diffusion coefficients and half-lives.

Other Type II networks are also difficult to relate to classical LALI activator-inhibitor models. For example, the network shown in **Appendix 3—figure 3**, which has no self-regulatory negative loop on v, allows for a minimum diffusion ratio $d$ that depends on a trade-off between the destabilizing cycle $c_4$ and the stabilizing cycles $c_1$, $c_2$ and $c_3$:

$$\text{Stability}: \quad |c_4| < \left| \frac{c_1 c_3}{c_2} \right| \tag{29}$$

$$\text{Instability}: \quad |d\,c_4| > |c_1 c_2| \tag{30}$$

$$\text{Diffusion constraint}: \quad d > \left| \frac{c_2^2}{c_3} \right| \tag{31}$$

$$d = \frac{d_w}{d_v}$$

In this case, the two stabilizing terms that define the minimum diffusion ratio $d$ are $c_2^2$ and $c_3$, which correspond to the squared strength of the self-regulatory negative loop of w and to the strength of the negative feedback between v and u, respectively. These terms cannot be related in a straightforward manner to the range of the two diffusible nodes. Within the LALI framework, the network topology suggests that u and v could behave like the short-range auto-activator, while w could behave like the long-range inhibitor. However, numerical simulations show that the system can form periodic patterns that are opposite to those of

classical activator-inhibitor models (*Appendix 3—figure 3*). The LALI framework therefore has little explanatory power to describe the pattern formation processes for Type II networks.

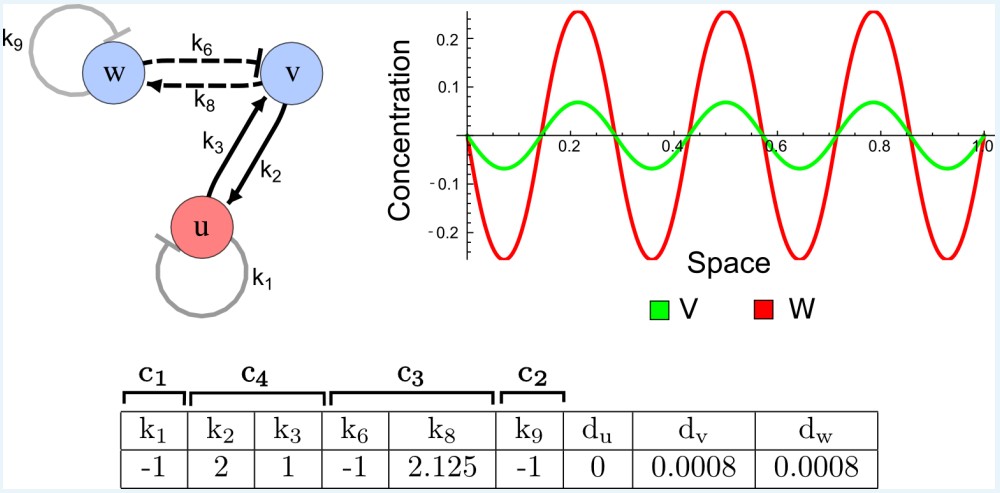

| | c₁ | c₄ | | c₃ | | c₂ | | | |
|---|---|---|---|---|---|---|---|---|---|

| $k_1$ | $k_2$ | $k_3$ | $k_6$ | $k_8$ | $k_9$ | $d_u$ | $d_v$ | $d_w$ |
|---|---|---|---|---|---|---|---|---|
| -1 | 2 | 1 | -1 | 2.125 | -1 | 0 | 0.0008 | 0.0008 |

**Appendix 3—figure 3.** Opposite pattern between the activator-inhibitor model and a Type II network. On the left: In the three-node network, the non-diffusible node u and the diffusible node v fulfill the role of the activator by mutually promoting each other (solid black lines) and by promoting their own inhibitor w (dashed lines). On the right: In contrast to the patterns of a classical two-node activator-inhibitor system, numerical simulations of the three-node network show activator peaks of v that appear more extended than inhibitor peaks of w.

## Relationship of Type III networks to LALI models

The LALI concept of local auto-activation and long-range lateral inhibition is also difficult to relate to the mechanism that underlies Type III networks. These networks satisfy the instability to spatial perturbations due to their topology and have pattern-forming conditions that only depend on the requirements for homogeneous steady state stability. For example, the Type III network shown in *Figure 2* can form a pattern for any diffusion ratio $d$:

$$
\begin{aligned}
\text{Stability:} \quad & c_1(c_1 c_2 + c_4^2 - c_3) < c_4 < c_2 c_3 \quad \& \quad c_1 c_2 > c_3 \\
\text{Instability:} \quad & d\, c_4 > 0 \\
\text{Diffusion constraint:} \quad & d > 0
\end{aligned}
$$

These stabilizing terms cannot be related in a straightforward manner to a short activation range and a long inhibition range. In addition, numerical simulations show that depending on the diffusion ratio $d$, which can vary freely, periodic patterns qualitatively opposite to the patterns expected in an activator-inhibitor system can be formed (*Figure 2c* right and *Appendix 3—figure 4a*). Moreover, as in the case of Type II networks, the qualitative aspect of the periodic patterns not only changes due to the diffusion ratio $d$ but also depending on the reaction kinetics, e.g. in the different Type III topology shown in *Appendix 3—figure 4b*.

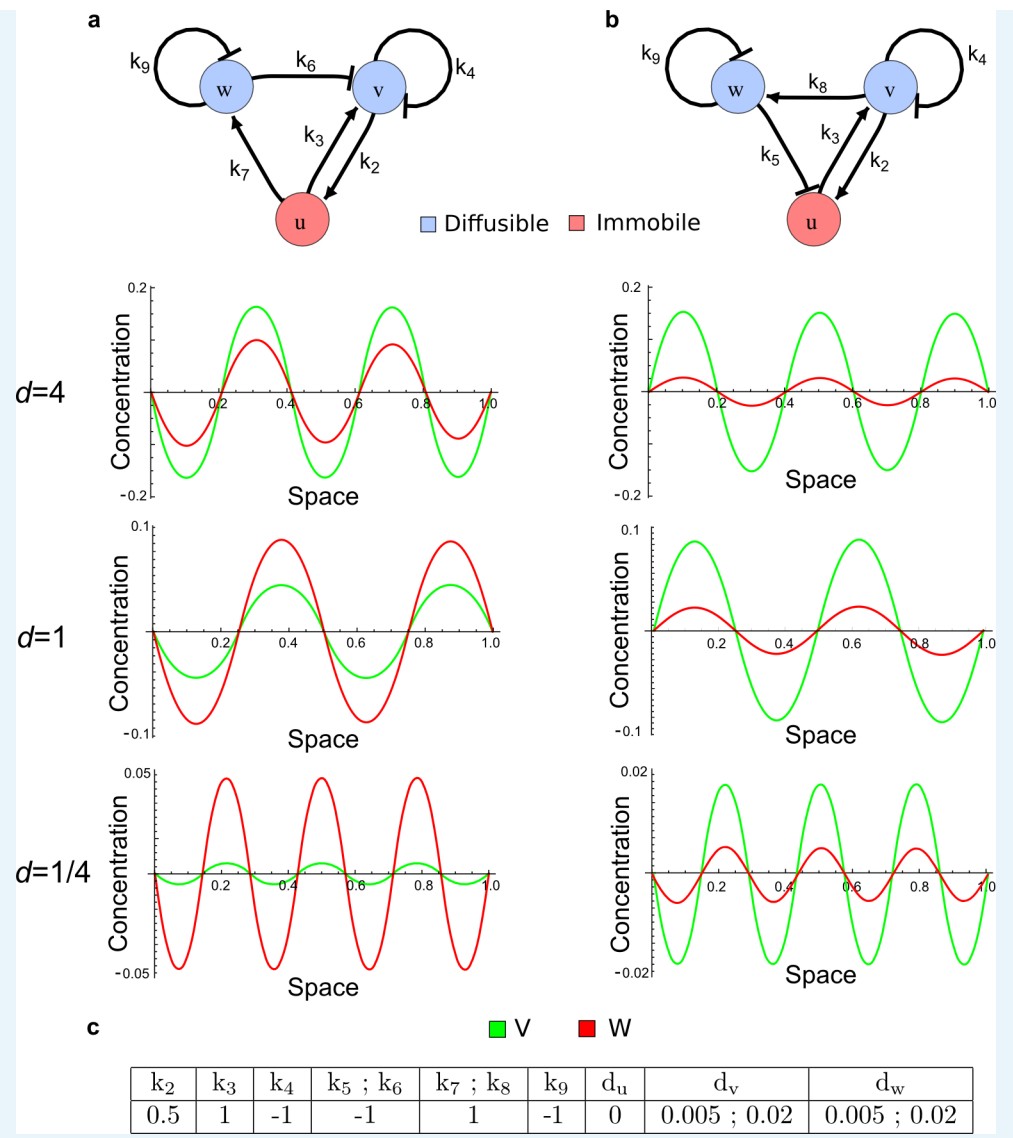

| $k_2$ | $k_3$ | $k_4$ | $k_5$ ; $k_6$ | $k_7$ ; $k_8$ | $k_9$ | $d_u$ | $d_v$ | $d_w$ |
|---|---|---|---|---|---|---|---|---|
| 0.5 | 1 | -1 | -1 | 1 | -1 | 0 | 0.005 ; 0.02 | 0.005 ; 0.02 |

**Appendix 3—figure 4.** Opposite ranges for the peaks of v and w in Type III networks. (**a**) Numerical simulations of the network shown in *Figure 2c* (right) with different diffusion ratios *d*: With *d* = 4, the periodic patterns of v and w are similar to the patterns of an activator and an inhibitor, respectively; but with *d* = 1 and *d* = 1/4, opposite patterns are observed. (**b**) A similar Type III network shows periodic patterns of v and w that are similar to the pattern of classical two-component activator-inhibitor systems independently of the diffusion ratio *d*. (**c**) Parameters used for the simulations in **a** and **b**. Identical parameters were used for the rates $k_5;k_6$ and $k_7;k_8$. The diffusion coefficients were set to $d_v$ = 0.005 and to $d_w$ = 0.02 for the case with *d* = 4, to $d_v$ = 0.02 and $d_w$ = 0.02 for the case with *d* = 1, and to $d_v$ = 0.02 and $d_w$ = 0.005 for the case with *d* =1/4.

These observations suggest that the final aspect of self-organizing periodic patterns is not necessarily related with the effective ranges of activators and inhibitors and does not reflect a mechanism that prevents the expansion of activator peaks. We propose instead that the periodic patterns are simply associated with the final growth at which each reactant reaches a dynamic equilibrium, which is determined both by reaction and diffusion terms.

In summary, applying the LALI concept based on differential diffusivity to investigate more complex networks provides little insights into the mechanism that underlies pattern formation. As an alternative, the graph-theoretical formalism presented in this study can be systematically

applied to all networks and helps to break down Turing systems into different parts by identifying destabilizing and stabilizing feedbacks. Our analysis highlights that the main role of diffusion is to destabilize the system to spatial perturbations by helping the destabilizing feedbacks to be stronger than the stabilizing feedbacks. In the next section, we show that these findings are consistent with the original reaction-diffusion example described by Turing, where the differential diffusivity is a necessary condition to destabilize the system. This contrasts with interpretations of reaction-diffusion patterning based on LALI models with local auto-activation and lateral inhibition, where the differential diffusivity requirement has been described as a necessary condition to stabilize activator peaks that would be otherwise unstable (*Appendix 3—figure 1*).

## The role of differential diffusion in Turing's model

### The original reaction-diffusion example proposed by Turing

In his seminal paper, Turing presented a simple example to demonstrate how diffusion can destabilize a reaction-diffusion system by amplifying small fluctuations (*Turing, 1952*). In this example, he considered two diffusible morphogens $X$ and $Y$ that react according to the equations in *Appendix 3—figure 5*. This system is in equilibrium when $X = 1$ and $Y = 1$. Turing described a numerical simulation consisting of a pair of cells that initially have roughly the same amount of X and Y ($X = 1.06, Y = 1.02$ in the first cell and $X = 0.94, Y = 0.98$ in the second cell). The simulation was performed by separately calculating the concentration changes resulting from either reaction or diffusion terms (*Appendix 3—figure 5*).

$$\frac{\partial \mathrm{X}}{\partial t} = \overbrace{5\mathrm{X} - 6\mathrm{Y} + 1}^{\blacksquare\ \text{Reaction}} + \overbrace{d_X \nabla^2 \mathrm{X}}^{\square\blacksquare\ \text{Diffusion}}$$

$$\frac{\partial \mathrm{Y}}{\partial t} = \underbrace{6\mathrm{X} - 7\mathrm{Y} + 1}_{\blacksquare\ \text{Reaction}} + \underbrace{d_Y \nabla^2 \mathrm{Y}}_{\square\blacksquare\ \text{Diffusion}}$$

**Appendix 3—figure 5.** A simple system of two reaction-diffusion equations proposed by Turing. X corresponds to the activator and Y to the inhibitor in the activator-inhibitor model.

When no diffusion is considered ($d_X = d_Y = 0$), this system returns to the equilibrium because of the feedbacks that implement a self-regulatory stable system (*Appendix 3—figure 6a–b*). To reach equilibrium, however, the system transiently increases or decreases the absolute concentrations of X and Y depending on the initial perturbations. The transient increase and decrease of X and Y depends on the auto-activation of X and the auto-inhibition of Y, respectively. These two self-regulatory loops are both key feedbacks that can destabilize the system in the presence of diffusion. In previous interpretations based on LALI models (see above), the auto-activation of the activator (X) was described as an important ingredient to destabilize the system, but the importance of the auto-inhibition of the inhibitor Y was generally overlooked (*Koch and Meinhardt, 1994*; *Marcon and Sharpe, 2012*; *Economou and Green, 2014*). The simulation in *Appendix 3—figure 6b* shows that when the relative difference between the activator and the inhibitor is large enough ($\Delta_c$), the auto-activation of X can bring the concentrations above the equilibrium steady state while the auto-inhibition of Y can bring the concentrations below the the equilibrium steady state. Due to the intrinsic stability of the network, however, the relative difference between X and Y concentrations ($\Delta_c$) is reduced over time, and deviations are only temporary.

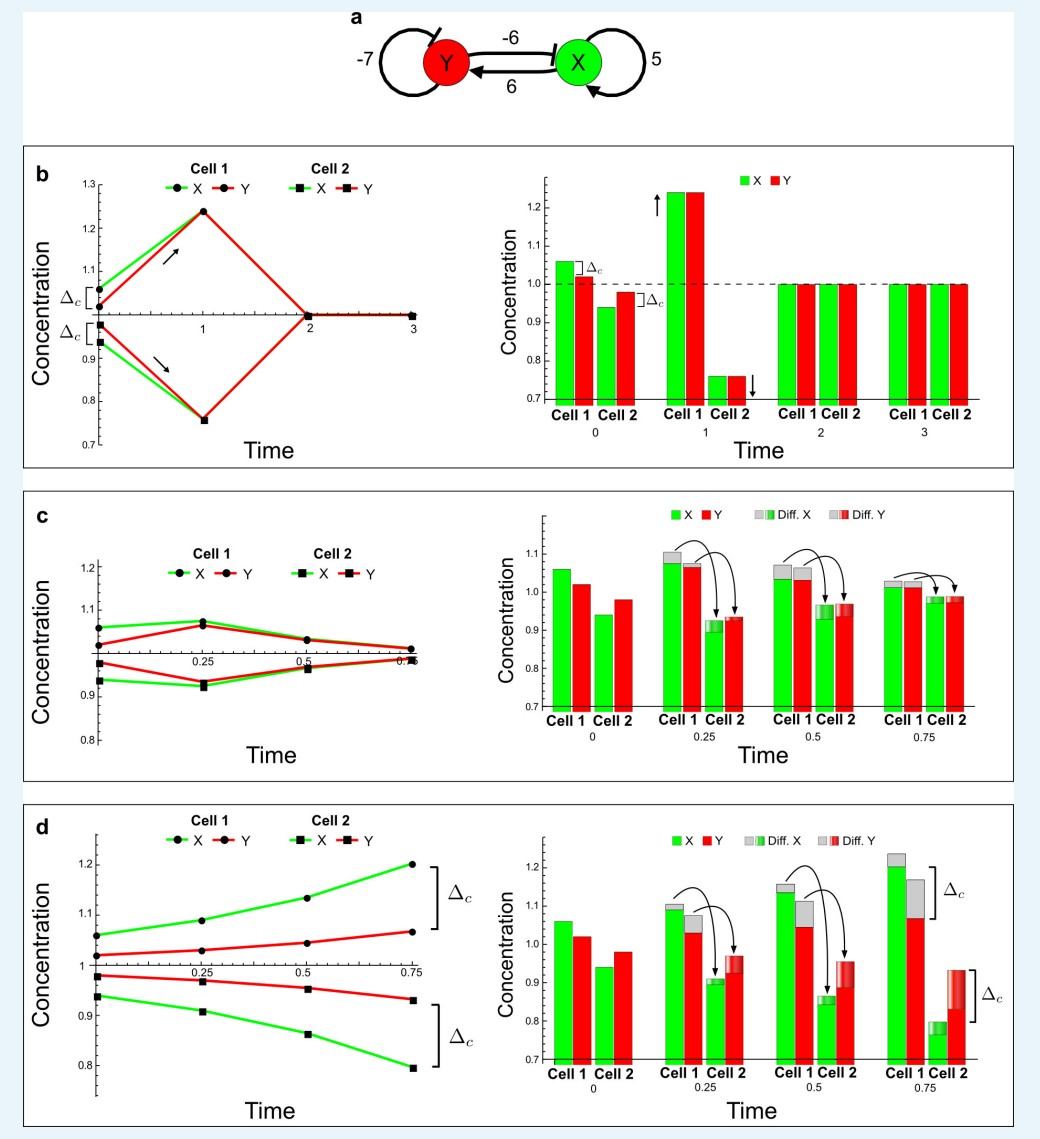

**Appendix 3—figure 6.** The original example proposed by Turing to show how diffusion can destabilize a reaction-diffusion system by amplifying small fluctuations. (**a**) The network diagram of the simple example proposed by Turing: X corresponds to the activator and Y to the inhibitor in the LALI activator-inhibitor model. (**b–d**) Numerical simulations on a pair of cells (cell 1 and cell 2) with initial conditions X = 1.06, Y = 1.02 in the first cell and X = 0.94, Y = 0.98 in the second cell as originally proposed by Turing. On the left, a graph shows the concentration change of X (green line) and Y (red line) over time in both cells. On the right, the histograms show the same concentration changes over time but additionally highlight the change due to diffusion: gray regions correspond to the amount of X and Y that diffuses out from the first cell and diffuses into the second cell as shown by graded green and red regions for X and Y, respectively. The diffusion process is represented by black arrows. (**b**) No diffusion: When X and Y do not diffuse, the system goes back to the equilibrium state X = 1, Y = 1 (dashed line). However, the small difference $\Delta_c$ between X and Y in the initial conditions stimulates the system to transiently deviate above and below the equilibrium state (black arrows) in the first and second cell, respectively (see the concentration at time 1). The intrinsic stability of the system eventually guarantees that $\Delta_c$ is reduced over time (see equations in *Appendix 3—figure 5*), and equal amounts of X and Y in the same cell bring the system back to equilibrium. (**c**) Equal diffusion: When X and Y diffuse equally, the system quickly returns to equilibrium. Diffusion acts as an equilibrating force by redistributing more activator than

inhibitor due to the higher concentration difference of activator between cell 1 and cell 2 (see the larger flow of the activator with respect to the inhibitor at time 0.25 and 0.5). (**d**) Differential diffusivity: If the inhibitor Y diffuses faster than X, the larger flow of Y maintains the relative difference $\Delta_c$ between activator and inhibitor in both cells and the systems keeps deviating from equilibrium. In cell 1, the greater dilution of the inhibitor with respect to the activator allows the activator and the inhibitor to grow further. In cell 2, the larger amount of inhibitor allows the activator and the inhibitor to decrease further since the inhibitor inhibits itself as well as the activator.

Turing observed that diffusion, which normally act as an equilibrating force, could increase $\Delta_c$ under certain conditions and further deviate the system from equilibrium. If both X and Y diffuse equally ($d_X = d_Y = 1$), the system quickly returns to its steady state, and diffusion works as an equilibrating force to redistribute X and Y towards homogeneity by moving more activator than inhibitor from the first to the second cell (*Appendix 3—figure 6c*). However, if Y diffuses more than X ($d_X = 0.5$ and $d_Y = 4.5$) the larger flow of Y from the first to the second cell helps to maintain a larger relative difference $\Delta_c$ in the first and second cell, respectively. Importantly, the same flow of Y drives the deviation from the equilibrium state in both cells simultaneously - the first cell deviates above equilibrium while the second cell deviates below. This challenges the classic LALI interpretation of the activator-inhibitor model shown in *Appendix 3—figure 1*: First, there is no chronological order between the formation of an activator peak and lateral inhibition in the surrounding areas, since they happen simultaneously. Second, it shows that these two processes are a direct consequence of the differential diffusivity, whose main role is not to prevent the expansion of activator peaks, but rather to maintain a larger relative difference $\Delta_c$ between activator and inhibitor to promote a simultaneous deviation above and below the equilibrium state.

In summary, the example presented by Turing shows that the role of differential diffusivity is to destabilize the reaction-diffusion system to simultaneously deviate above and below equilibrium. This suggests that the classical interpretation of the activator-inhibitor model within the LALI framework (*Appendix 3—figure 1*) is inaccurate: In particular, the long-range inhibitor does not limit the expansion of forming activator peaks but rather promotes the simultaneous formation of activation and inhibition peaks. Indeed, irrespective of the strength of initial perturbations, if X and Y have equal diffusivities (or if they do not diffuse) the system will always return to its equilibrium state and will never lead to 'an overall auto-catalytic explosion' (*Meinhardt and Gierer, 2000*). An indiscriminate expansion of activator peaks can be obtained only when X diffuses and Y is immobile or in systems implemented by just one diffusible self-enhancing activator. However, these systems represent different scenarios that do not provide an explanation for the role of differential diffusivity in classical reaction-diffusion systems.

## A re-interpretation of the substrate-depletion model within the Turing framework

In addition to the activator-inhibitor system, the substrate-depletion model is another classical two-component reaction-diffusion system, which forms out-of-phase periodic patterns of activator and substrate as opposed to the in-phase patterns of activator-inhibitor systems (*Appendix 3—figure 7*). Applying the LALI concepts of local auto-activation and lateral inhibition to explain pattern formation in the substrate-depletion model is non-trivial (*Appendix 3—figure 7d–g*), since it is unclear how the higher diffusivity of the substrate could limit the expansion of a forming activator peak. A different explanation that is not directly based on differential diffusivity was proposed by assuming that activator peaks stop expanding when the substrate is consumed below a certain threshold (*Koch and Meinhardt, 1994*).

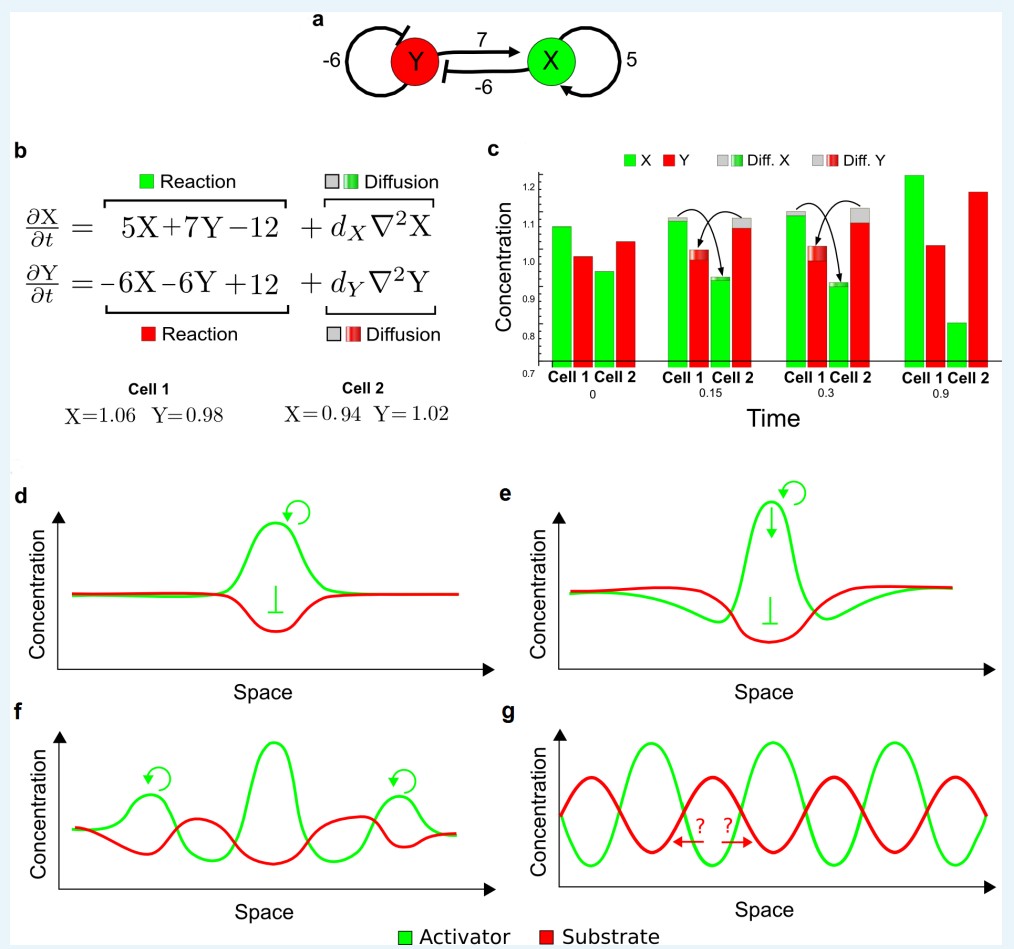

**Appendix 3—figure 7.** Schematic representation of the substrate-depletion model. (**a**) The network diagram of a substrate-depletion model: The activator X (green) inhibits the substrate Y (red) and promotes itself. (**b**) Equations of the substrate-depletion model and initial conditions used for the simulation in **c**. A constitutive removal of the activator (−12) and a constitutive production of the substrate (+12) were chosen to have an equilibrium state in X = 1, Y = 1 as in the original example proposed by Turing. (**c**) Simulation on a pair of cells shows that the greater diffusivity of the substrate Y plays a similar role as in the activator-inhibitor model by simultaneously destabilizing the system in opposite directions. In the first cell, the arrival of more substrate allows the activator to grow further, whereas in the second cell the dilution of the substrate leads to a decrease in activator and consequently to an overall increase of the substrate. Thus, the model simultaneously deviates from equilibrium when the activator X is high and the substrate Y is low and vice versa. (**d–g**) Schematic representation of an attempt to interpret the substrate-depletion model within the LALI framework based on local auto-activation and lateral inhibition. (**d**) A local advantage of the activator allows the formation of an activation peak (green) with correspondent depletion of the substrate (red). (**e**) The peak of activation stops expanding not due to the differential diffusivity but because the substrate falls below a certain threshold. Indeed, it appears that the higher diffusion of the substrate would promote further growth of the activator rather than limit its expansion (red arrows). The role of differential diffusivity is unclear. (**f**) Other peaks are formed in surrounding regions by the same mechanism. (**g**) Opposite periodic patterns of the activator and substrate are eventually formed.

In contrast, the original interpretation by Turing can easily explain the mechanism that underlies pattern formation in the substrate-depletion model. In this system, the interactions

between X and Y are opposite to the interactions in the activator-inhibitor model; therefore, the system deviates from equilibrium when the activator X is high and the substrate Y is low in the same cell and vice versa. Nevertheless, differential diffusivity plays the same role as in the activator-inhibitor model by causing a larger flow of the substrate Y from one cell to the other, which simultaneously destabilizes the system in two opposite directions with respect to the equilibrium state (*Appendix 3—figure 7a–c*).

## Simultaneous destabilization in two opposite directions by differential diffusivity

The simultaneous destabilization in opposite directions by differential diffusivity described in the section above is also supported by numerical simulations on more than two cells started with small random deviations from the equilibrium steady state as initial conditions. These simulations do not show the chronological series of events depicted by classical interpretations of the LALI framework based on local auto-activation and lateral inhibition (*Appendix 3—figure 1*) but rather show the simultaneous formation of activation and inhibition peaks across the whole spatial domain (*Appendix 3—figure 8*).

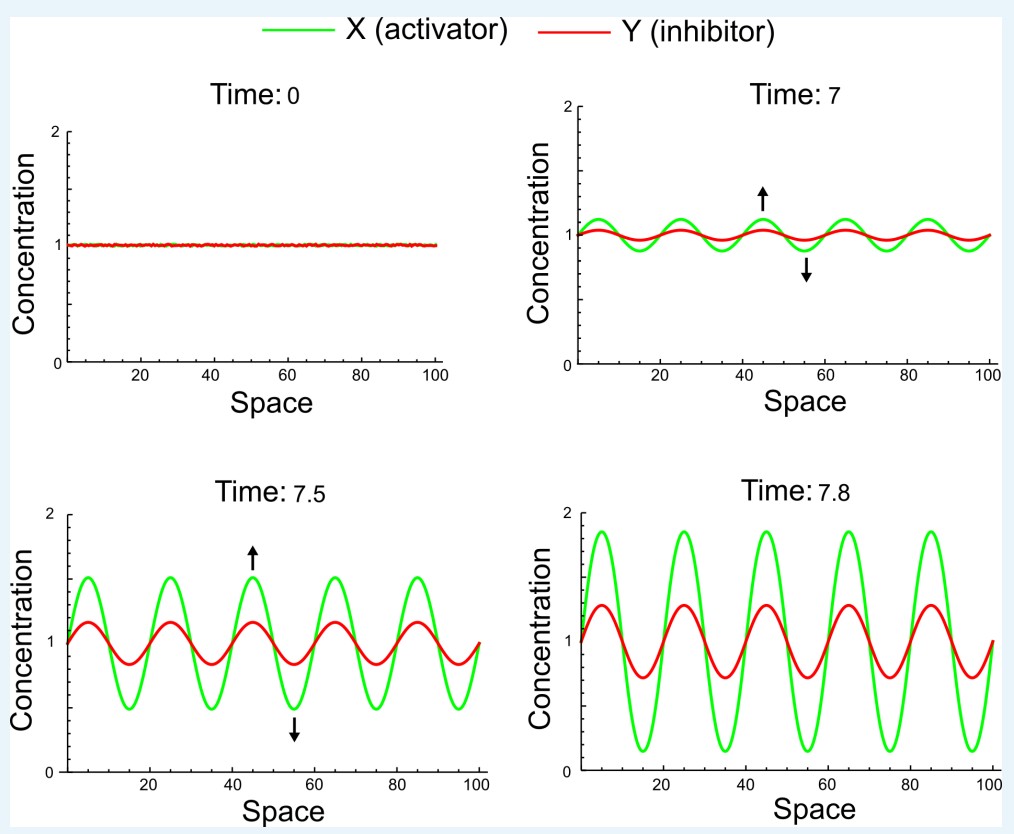

**Appendix 3—figure 8.** Simulation of the original reaction-diffusion network proposed by Turing started with random perturbations around the homogeneous steady state and uniformly distributed in the interval (-0.001, 0.001). Black arrows highlight the deviations of the activator (green) and the inhibitor (red). These deviations are simultaneously promoted above and below the steady state across the entire spatial domain by the differential diffusivity. The final aspect of the periodic peaks does not reflect a difference between activation and inhibition ranges but rather reflects a different speed of growth for each periodic pattern, which is determined both by reaction and diffusion terms.

The only situation when patterning dynamics in agreement with the LALI interpretation can be observed is when large localized perturbations of the activator are used as initial conditions (*Appendix 3—figure 9a*). However, these simulations do not only reflect the dynamics of the Turing network but also the dilution and propagation of a localized perturbation, which stimulates the formation of a dissipative soliton (*Purwins et al., 2005*). This is not the best scenario to investigate the mechanisms that drive pattern formation because the localized perturbation hides the underlying mechanisms that break the symmetry of an initially homogeneous state. The original simulation proposed by Turing (*Appendix 3—figure 6*) and the one-dimensional simulation shown in *Appendix 3—figure 8* suggest that breaking symmetry is not the result of LALI but instead the result of a simultaneous deviation from the equilibrium state in opposite directions due to the differential diffusivity. In agreement with this interpretation, if the magnitude of the localized perturbation is reduced, the simultaneous appearance of inhibitor and activator peaks can be recovered (*Appendix 3—figure 9b*). The LALI model therefore does not accurately describe the underlying Turing mechanism that can drive the self-organization of periodic patterns in the absence of initial asymmetries.

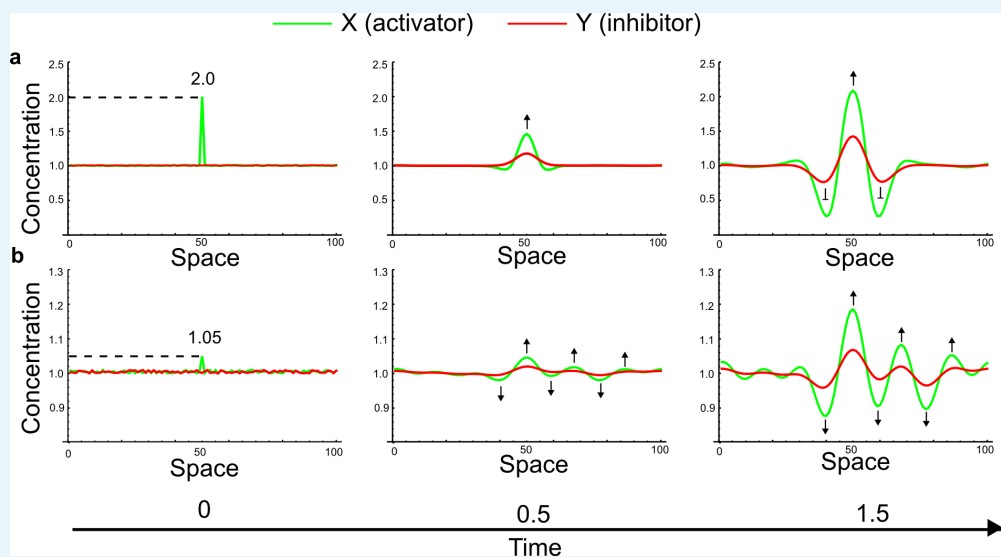

**Appendix 3—figure 9.** Simulations of the original example proposed by Turing (*Appendix 3—figure 5*) started with small random fluctuations around the homogeneous steady state and a large localized concentration of activator at the center of the domain. (**a**) When the localized perturbation has a high magnitude, after an initial dilution of the perturbation patterning dynamics in agreement with the classical interpretation based on the LALI mechanism can be observed: A peak of activator forms first, and a higher amount of inhibitor appears to diffuse into the surrounding areas to inhibit the spreading of the activator. (**b**) If the magnitude of the localized perturbation is reduced, a simultaneous formation of activator and inhibitor peaks is observed (black arrows). This suggests that LALI is only a phenomenological description of the effect of the large localized perturbation of the activator rather than a bona fide model of the dynamics of real patterning systems.

# Mechanism of pattern formation with equally diffusing signals

In the following, we discuss how reaction-diffusion models that contain immobile factors can become unstable to spatial perturbations even when all diffusible reactants have the same diffusivity.

## Instability with equally diffusing signals due to immobile reactants

The original example presented by Turing can be modified into a Type III network by introducing an additional reactant Z that participates in a mutual activation with the reactant X and is repressed by Y:

$$
\begin{aligned}
\frac{\partial X}{\partial t} &= 5Z - 6Y + 1 + d_X \nabla^2 X \\
\frac{\partial Y}{\partial t} &= 6X - 7Y + 1 + d_Y \nabla^2 Y \\
\frac{\partial Z}{\partial t} &= 5X - 6Y + 1 + d_Z \nabla^2 Z
\end{aligned}
\tag{32}
$$

In this network, the auto-catalysis of X is no longer direct as the one in **Appendix 3—figure 6a**, but it is implemented by the mutual activation with Z (**Appendix 3—figure 12a**). Similar to the original example of Turing, the system is in equilibrium when $X = 1, Y = 1$ and $Z = 1$. The LALI model does not provide a satisfactory explanation for the pattern formation mechanism in this system, since the inhibitor does not spread faster into the lateral domains of the activator peak due to equal diffusivities (**Appendix 3—figure 10** and **Appendix 3—figure 11**). In the following, we therefore analyze in detail how this modified system is destabilized by amplifying small fluctuations.

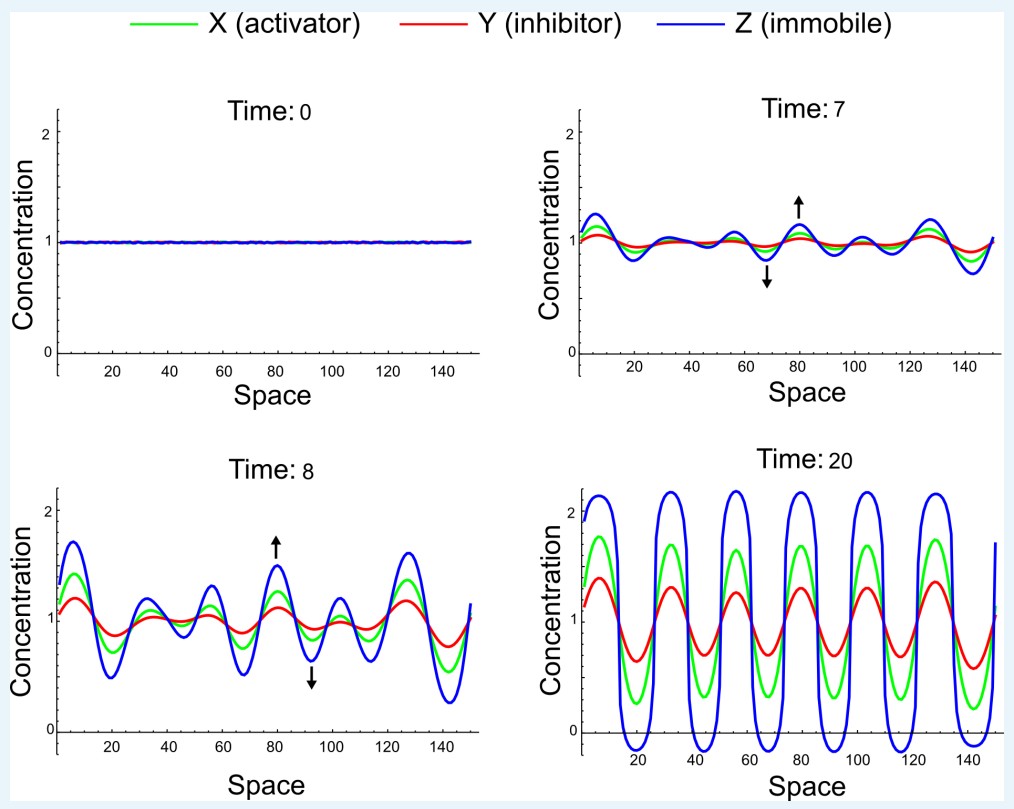

**Appendix 3—figure 10.** A simulation of the three-component reaction-diffusion system (**Equation 32**) with $d_X = d_Y = 1$, $d_Z = 0$ initialized with random perturbations around the homogeneous steady state uniformly distributed in the interval (-0.001, 0.001). X, Y, and Z simultaneously deviate above and below the steady state across the entire spatial domain (black arrows). The final aspect of the periodic peaks does not reflect a difference between the ranges of the activator X and the inhibitor Y, but rather reflects a different speed of growth for each periodic pattern, which is determined both by reaction and diffusion terms.

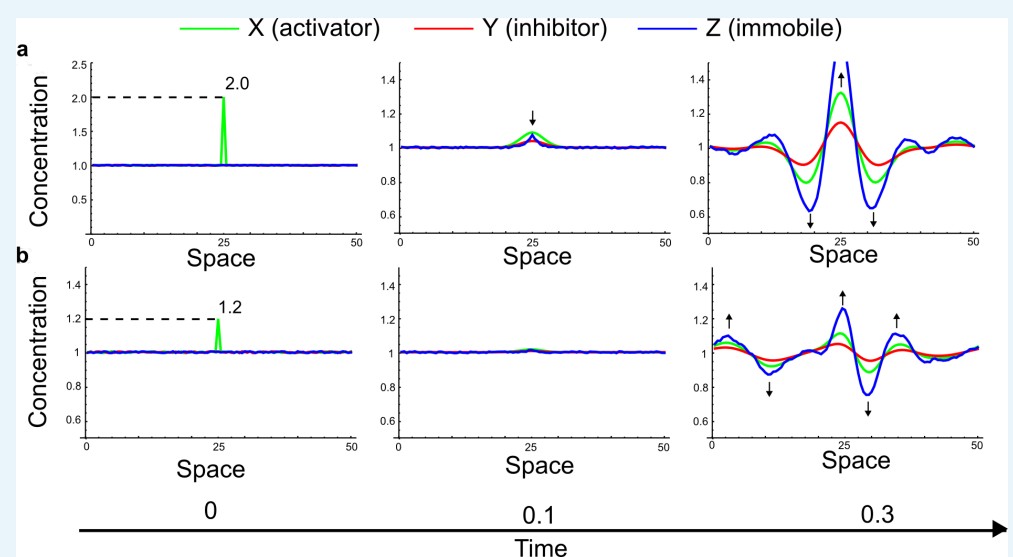

**Appendix 3—figure 11.** Simulations of the three-node extension to the original example proposed by Turing (*Equation 32*) started with small random fluctuations around the homogeneous steady state and a large localized concentration of activator at the center of the domain. (**a**) Even when the localized perturbation has a high magnitude similar to the simulations in *Appendix 3—figure 9*, the patterning dynamics are inconsistent with the LALI mechanism since the inhibition does not spread faster into the surrounding areas than the activator. (**b**) If the magnitude of the localized perturbation is reduced, a simultaneous formation of activator and inhibitor peaks is observed similar to the simulations in *Appendix 3—Figure 9* (black arrows).

We performed numerical simulations of a pair of cells starting from a slight perturbation of the equilibrium state: $X = 1.04, Y = 1.02, Z = 1.06$ in the first cell and $X = 0.96, Y = 0.98, Z = 0.94$ in the second cell. When no diffusion is considered ($d_X = d_Y = d_Z = 0$), the system returns to the equilibrium owing to its intrinsic stability (*Appendix 3—figure 12b*). As in the original example of Turing (see above), the equilibrium is reached by transiently increasing or decreasing the absolute concentrations of X, Y, and Z depending on the initial perturbations, but a reduction of the relative difference between X and Y concentrations ($\Delta_c$) brings the system back to equilibrium (*Appendix 3—figure 12b*). Similarly, if all reactants diffuse equally ($d_X = d_Y = d_Z = 1$), the system quickly returns to its equilibrium since diffusion helps to redistribute X, Y, and Z towards homogeneity (*Appendix 3—figure 6c*).

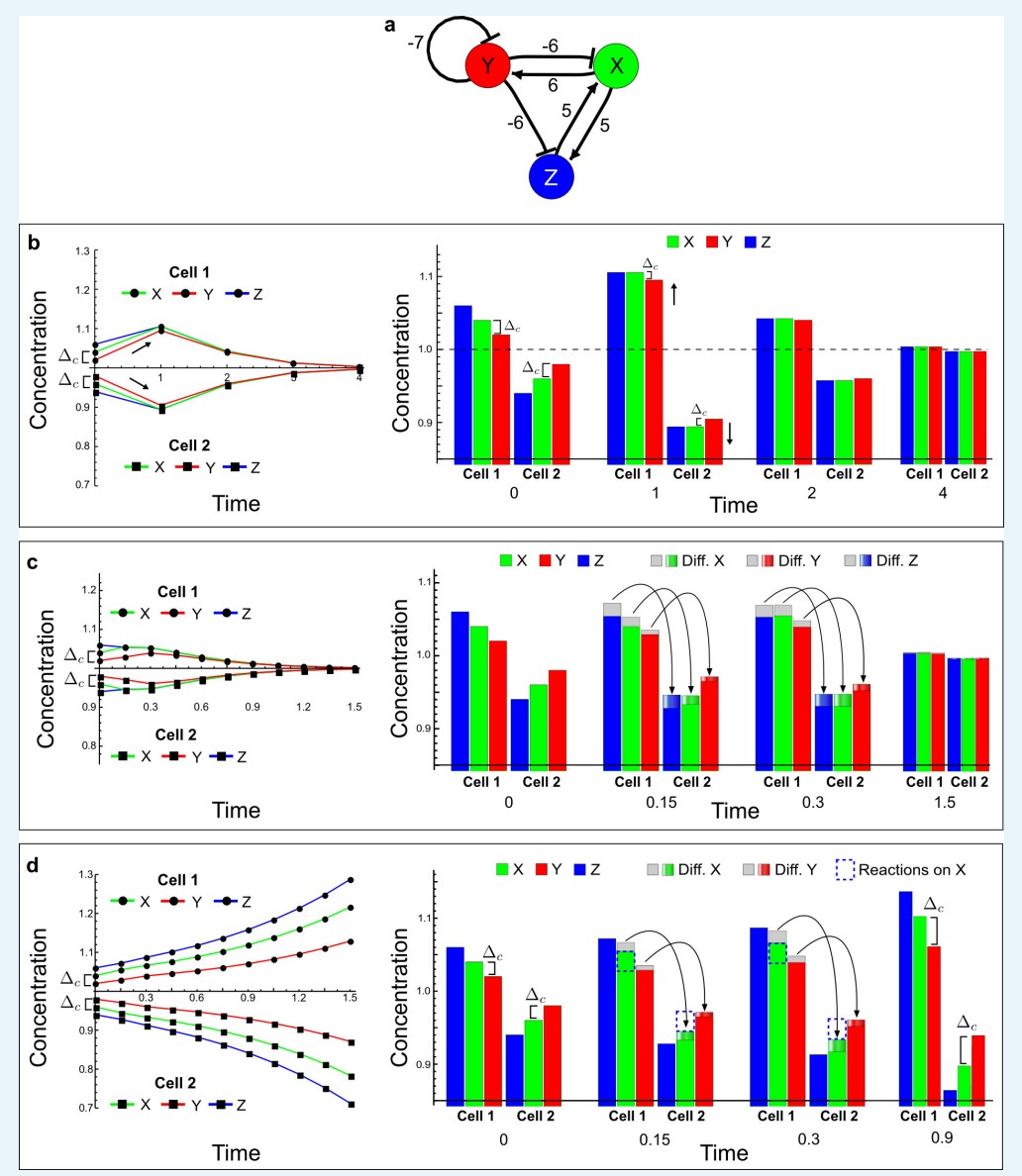

**Appendix 3—figure 12.** A Type III network that extends the simple example proposed by Turing. (**a**) The self-enhancement of the activator X is implemented by a mutual activation with Z, and both X and Z are inhibited by the inhibitor Y. (**b–d**) Numerical simulations of a pair of cells (cell 1 and cell 2) with initial conditions X = 1.04, Y = 1.02, and Z = 1.06 in the first cell, and X = 0.96, Y = 0.98, and Z = 0.94 in the second cell. On the left, the graphs show the concentration changes of X (green line), Y (red line), and Z (blue line) over time in both cells. On the right, the histograms show the same concentration changes over time but additionally highlight the change due to diffusion: Gray regions correspond to the amount of X, Y, and Z that diffuse out from the first cell, and the amount that diffuses into the second cell is indicated by graded green, red, and blue regions for X, Y, and Z, respectively. The diffusion process is represented by black arrows. (**b**) No diffusion: When X, Y, and Z do not diffuse, the system returns to the equilibrium state X = 1, Y = 1 and Z = 1 (dashed line). However, the small differences between X, Y, and Z in the initial conditions stimulate the system to transiently deviate above and below the equilibrium state (black arrows) in the first and second cell, respectively (see the concentration at time 1). The intrinsic stability of the system guarantees that $\Delta_c$ is reduced over time, and the system returns to equilibrium. (**c**) Equal diffusion: When X, Y, and Z diffuse equally, the system also quickly returns to equilibrium. Diffusion acts as an equilibrating force by redistributing more X and Z, and less Y, due to the

higher concentration difference of X and Z between cell 1 and cell 2 (see the larger flow of X (green) and Z (blue) with respect to the flow of Y (red) at time 0.15 and 0.3). (**d**) Equal diffusion of X and Y in the presence of non-diffusible Z: Diffusion still acts as an equilibrating force by redistributing more X than Y from the first to the second cell. However, the fact that Z is not subjected to diffusion in combination with the diffusion of Y allows the reaction term of X (dashed blue line and first equation in *Equation (32)* to compensate for the larger flow of X. For example, at time 0.15 Z and Y in the first cell promote an increase of X (dashed blue line) that is greater than the amount of X that diffuses out (gray region above the green bar), while Z and Y in the second cell promote a decrease of X (dashed blue line) that is larger than the amount of X that diffuses in (graded green). At the next time point, the dilution of Y allows Z to further diverge from the equilibrium state (see third equation in *Equation [32]*) and together with Z promotes a stronger compensation (dashed blue line) for the even larger flow of X. In this way, the system increases the difference between X and Y ($\Delta_c$) despite their equal diffusivities and keeps deviating them from equilibrium in opposite directions.

However, if X and Y diffuse equally ($d_X = d_Y = 1$) and Z is immobile ($d_Z = 0$), the relative difference between X and Y ($\Delta_c$) is progressively increased, and both cells keep on deviating from equilibrium (*Appendix 3—figure 6d*). In the original example presented by Turing (see above), the deviation from equilibrium was guaranteed by differential diffusivity that implemented a larger flow of Y from the first to the second cell. In contrast, when X and Y have the same diffusivity, the flow of X instead is larger due to the higher concentration gradient of X between the two cells. This leads to a progressive decrease in $\Delta_c$ and therefore to a return to equilibrium (*Appendix 3—figure 6c*). In the simulation of the three-node network shown in *Appendix 3—figure 12d*, X and Y have the same diffusivity and consequently a larger flow of X is also observed. However, in this case the immobile reactant Z together with the mobile reactant Y can compensate for the larger flow of X. This is possible because Z is not redistributed by diffusion and together with a small flow of Y can promote and inhibit X in the first and second cell respectively (see first equation in *Equation (32)* and dashed blue line in *Appendix 3—figure 12d*). In addition, the small flow of Y form the first to the second cell allows Z to further deviate from equilibrium because of its positive feedback with X (see third equation in *Equation [32]*), which re-enforces the effect of Z over time and maintains the divergence of the entire system from equilibrium. Immobile reactants therefore fulfill a role as 'capacitors' that can integrate the effect of diffusing reactants to destabilize the reaction-diffusion system and to drive self-organizing pattern formation.

Inspecting the example shown in *Appendix 3—figure 12* from a LALI perspective, it could be speculated that the effect of the immobile reactant in Type III networks is just equivalent to a reduction of the effective range of the auto-activation to satisfy the differential diffusivity independently of the activator diffusion rate. This simplification, however, neglects the important role of the specific reaction terms associated with the immobile reactant. Indeed, in the example shown in *Appendix 3—figure 12d*, the immobile reactant Z does not just reduce the range of activator peaks; rather its dynamics - and in particular its ability to diverge fast from equilibrium - are an integral part of the mechanism that allows the peaks to form in the first place. These dynamics do not only depend on the mutual activation between X and Z but also on the the inhibition of Z by Y, whose diffusion stimulates Z to diverge from equilibrium. Interpreting these dynamics as an effective change in spatial range of the activator appears to provide little insights into the general mechanism that underlie pattern formation. Indeed, in other Type III networks, e.g. more complex networks with more than one immobile reactant, the identification of similar effective ranges requires the definition of alternative ad hoc approximations, which ultimately correspond just to a reformulation of the reaction kinetics of the network.

In conclusion, Type III networks can form a pattern even when the diffusing signals have the same range due to the capacitor effect of immobile reactants and associated reaction terms. Remarkably, this can happen even when the activator diffuses more than the inhibitor (*Appendix 3—figure 13*). This in an intrinsic property of Type III topologies, where the

immobile reactant acts as a buffer to amplify any small advantages or disadvantages over the inhibitor diverging quickly from the equilibrium state. In agreement with this finding, if the fast divergence of the immobile reactant is limited by a negative self-regulation, most networks are of Type II, meaning that their ability to compensate for the range of the mobile reactants depends on the relative speed at which each reactant grows. For example, in the Type II network shown in *Appendix 3—figure 3*, this can happen only when the growth of the activator is slowed down sufficiently by the inhibitor through the negative cycle $c_3$.

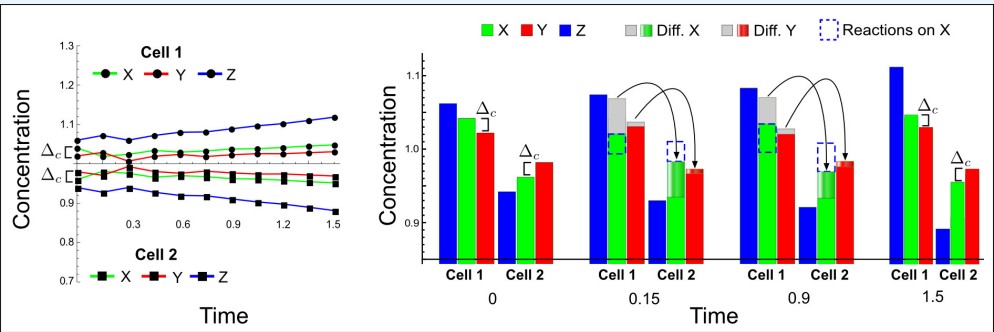

**Appendix 3—figure 13.** Numerical simulations of the Type III network shown in *Appendix 3—figure 12a* with an immobile reactant Z and X diffusing four time faster than Y ($d_X = 4$, $d_Y = 1$, $d_Z = 0$). In this case, there is an even larger flow of X from the first to the second cell. Initially, the immobile reactant Z is not able to compensate for this larger flow (see time point 0.15, where the dashed line that represents the reaction term of X is smaller than the diffusion of X [gray and graded green regions]), which allows the activator to decrease below the inhibitor level in the first cell and to increase above the inhibitor level in the second cell. In turn, the deviation from equilibrium of the immobile reactant Z is also slightly reduced (see the blue line at time point 0.3 in the graph on the left). However, since Z is not diluted by diffusion, it can recover and grow further over time because of the diffusion of Y. Z and Y together are able to eventually compensate for the flow of X (see the larger contribution of the reaction term of X at time point 0.9 in the histograms (dashed blue line) with respect to the diffusion of X [gray and graded green regions]). This allows to quickly restore the initial relative difference between X and Y and to keep deviating from equilibrium.

## Summary: The role of immobile reactants in driving self-organizing patterns

The examples presented in this Appendix highlight that in classical two-reactant Turing models the differential diffusivity destabilizes the equilibrium state by maintaining an imbalance between reactants, which drives a further deviation from equilibrium. Importantly, the reaction terms of the Turing system guarantee that the deviation happens simultaneously above and below the equilibrium state. In the activator-inhibitor model, for example, the differential diffusivity not only gives an advantage to the self-enhancement of the activator but simultaneously to the auto-inhibition of the inhibitor. Therefore, in accordance with a recent proposal (*Klika et al., 2012*), we suggest that the negative self-regulation of the inhibitor has a more important role than previously assumed.

In agreement with these observations, one-dimensional simulations like the one shown in *Appendix 3—figure 8* reveal that the periodic patterns of Turing systems are formed with a simultaneous appearance of activation and inhibition peaks and that the patterning dynamics do not follow the sequence of events described by the LALI mechanism based on local auto-activation and lateral-inhibition. The periodic patterns therefore do not reflect a longer range of the inhibitor to limit the auto-activation. Instead, we propose that the periodic patterns of both the activator and the inhibitor reflect only one range, usually referred to as the

wavelength, which is determined by the differential diffusion but also by reaction terms. According to this view, the periodic patterns formed in the activator-inhibitor model reflect different amplitudes of activator and inhibitor levels - rather than a difference in the ranges - that depend both on differential diffusivity and reaction terms.

We therefore propose that the role of immobile reactants in Type II and Type III networks is not to implement an effective difference in the ranges of local auto-activation and long-range inhibition, but rather to help the system to diverge from equilibrium, which is normally achieved by differential diffusivity in classical two-component Turing systems. We find that immobile factors can help to destabilize the system, since they are not subjected to the equilibrating effect of diffusion and therefore fulfill a role as 'capacitors' that can integrate the effect of diffusing reactants to destabilize the reaction-diffusion system by quickly amplifying perturbations.

# Appendix 4: Synthetic reaction-diffusion circuit design

The networks presented in *Figure 5—figure supplement 1* are all alternative implementations of synthetic reaction-diffusion systems obtained by addition of negative feedbacks to an existing synthetic circuit that implements a positive feedback. In contrast to classical activator-inhibitor models, these networks show that many realistic reaction-diffusion systems do not require differential diffusivity. In addition, given the explicit representation of cell-autonomous factors, these networks also suggest at which level of the signaling pathways the new feedbacks should be introduced. On the one hand, these predictions help to bridge the gap between theoretical models and real systems, and on the other hand they present engineers with new challenges for the implementation of specific synthetic network designs. The high-throughput results of RDNets can be used to choose the network design that better fits the available synthetic toolkits.

In the following, we provide an example of an alternative implementation of the reaction-diffusion system presented in *Figure 5* with a network that has a more complex synthetic design but that requires almost no parameter optimization. In particular, we analyze one of the highly robust Type III networks identified by RDNets (highlighted in *Figure 5—figure supplement 1*). This network requires the addition of three negative feedback loops: one corresponding to the decay of the ligand involved in the positive feedback ($c_1$), one implementing a negative feedback between the ligand and its signaling ($c_3$), and another feedback between an additional ligand and its signaling ($c_2$, *Appendix 4—figure 1b*). Strikingly, this Type III network has very simple pattern forming requirements (*Appendix 4—figure 1c*): To guarantee stability, $c_2$ has to be greater than the other negative feedback, and the strength of the positive feedback $c_4$ has to be small. The network does not have special requirements for the instability, which is intrinsically guaranteed by the Type III network topology. A comparison between the original network that implemented the positive feedback (*Appendix 4—figure 1a*) and this candidate topology (*Appendix 4—figure 1b*) reveals that two of the new feedbacks could be implemented simply by increasing the turn-over rate of IP ($c_1$) and by implementing a negative feedback between IP-signaling and its own expression or activity ($c_3$). The third feedback ($c_2$), however, implies a more complex synthetic design: in this case, another ligand that signals through an independent receptor should be introduced. Moreover, the signaling of the new ligand should implement a negative feedback on its own expression or activity that is mediated by components of the signaling pathway of IP (z). This last requirement is more challenging to implement since it requires to design a signaling pathway component of IP that can simultaneously promote IP and inhibit the new ligand. The sharing of this component is an intrinsic requirement of the model and guarantees the coupling between the two signaling pathways. The simple addition of a downstream target of IP that independently inhibits the new ligand does not represent an equivalent implementation. A possible common element that could fulfill this design is a factor that mediates the secretion of IP but also the produces an inhibitor of the new ligand. This factor, which in the model corresponds to z, would have to be promoted by both signaling pathways.

Marcon *et al*. eLife 2016;5:e14022. DOI: 10.7554/eLife.14022

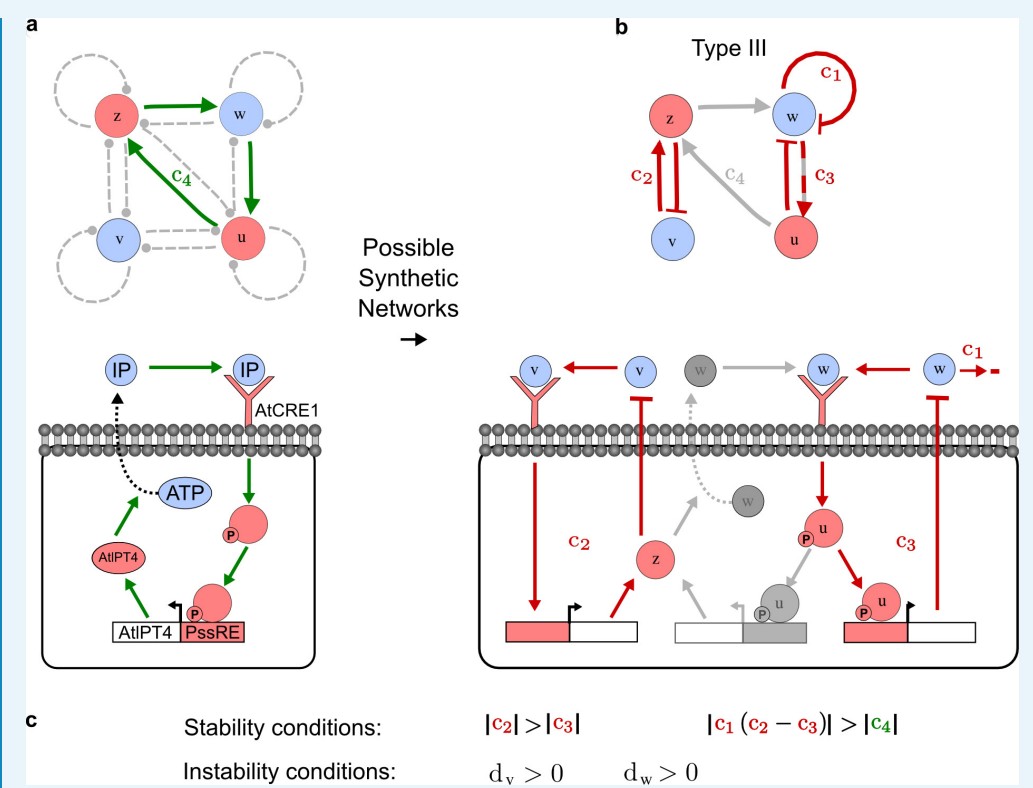

**Appendix 4—figure 1.** An alternative synthetic network for the case in *Figure 5*. (a) A previous synthetic circuit was developed to implement a positive feedback loop between the cytokinin IP and its signaling in yeast. (b) A robust Type III network identified by RDNets as a possible extension with three new negative feedbacks $c_1$, $c_2$, $c_3$ to form a reaction-diffusion pattern. (c) Pattern forming conditions for the network shown in **b**. Note the extremely simple requirements on cycle strength.

In conclusion, although the practical implementation of this system appears more challenging, it is also extremely appealing - once the network topology has been developed, the system would robustly guarantee the formation of a spatial pattern with the only requirement that the destabilizing feedback is not too strong.

# Appendix 5: Derivation and simulations of models with sigmoidal kinetics

RDNets can automatically provide reaction and diffusion parameters that satisfy the pattern forming requirements of network topologies. These parameters are used to develop and simulate partial differential equations whose partial derivatives have a linear part equal to the correspondent element in the Jacobian matrix analyzed by RDNets. In other words, the linear part of the partial derivative of any reaction regulation term (linear or non-linear) can be easily related with the reaction rates analyzed by RDNets. RDNets provides in-built functionalities to simulate linear models with simple cubic saturation terms as the ones derived in *Raspopovic et al. (2014)* and in *Miura and Maini (2004)*. However, the predictions of our analysis are not limited to linear models but are also particularly relevant for partial differential equations with non-linear terms such as sigmoidal kinetics that have often been used in gene network modeling (*Mjolsness et al., 1991*). In the following, we present a strategy to derive models with sigmoid regulation functions that can be built on top of the parameter rates identified by RDNets.

Since the automated linear-stability analysis performed by RDNets is a systematic exploration of Jacobian matrices $J$ and diffusion matrices $D$, we define an approach to derive non-linear dynamical models from the Jacobian $J$. This corresponds to the reverse of what has been done in previous studies, where the Jacobian matrix $J$ and the pattern forming conditions were derived by analyzing dynamical non-linear models (*Diambra et al., 2015*; *Koch and Meinhardt, 1994*).

A general sigmoid regulation function $s(x, k)$ that describes a change in concentration promoted by an input $x$ is defined as

$$s(x,k) = \frac{1}{1+e^{-kx}} \tag{33}$$

where $k$ is a parameter that defines the non-linearity or steepness of the sigmoid (*Appendix 5—figure 1*) and $x$ is the input concentration of a reactant.

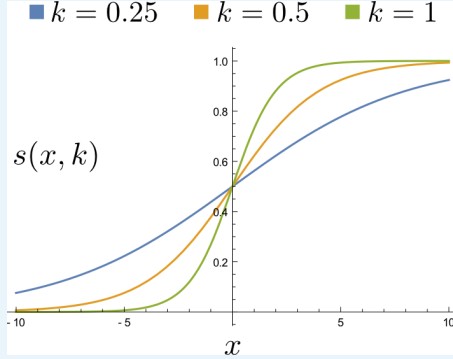

**Appendix 5—figure 1.** A sigmoid regulation function with different $k$ values.

The function $s(x, k)$ is equal to $0.5$ when $x$ is equal to zero:

$$s(0, k) = 0.5$$

We rewrite $s(x, k)$, such that a differential equation with these sigmoid regulation terms would have steady state at $x = 0$:

$$s(x,k) = \frac{1}{1+e^{-kx}} - 0.5$$

Next, we observed that the partial derivative of $s(x,k)$ evaluated at steady state is equal to

$$s'(x,k)_x^{x=0} = \left( \frac{ke^{-kx}}{(e^{-kx}+1)^2} \right)^{x=0} = \frac{k}{4}$$

We therefore modify $s(x,k)$, such that the resulting partial differential equation at steady state has derivative $k$:

$$s(x,k) = 4 \left( \frac{1}{1+e^{-kx}} - 0.5 \right) \quad s'(x,k)_x^{x=0} = k \tag{34}$$

Since the elements of the Jacobian represent the partial derivatives evaluated at steady state, the sigmoid regulation functions in **Equation (34)** have steepness constants that directly relate with the reaction rates of the Jacobian but also have an intrinsic saturation as in real biological systems. A reaction-diffusion system, where all the regulation functions have sigmoid kinetics, will form a pattern as predicted by the linear stability analysis. As an example, we write the equation of the synthetic network presented in **Figure 5** with sigmoid regulation terms and perform a 1D simulation (**Appendix 5—figure 2** and **Equation [35]**).

$$
\begin{aligned}
\frac{\partial u}{\partial t} &= \alpha_2 s(\mathrm{v}, \mathrm{k}_2) + \alpha_5 s(\mathrm{w}, \mathrm{k}_5) \\
\frac{\partial v}{\partial t} &= -\alpha_3 s(\mathrm{u}, \mathrm{k}_3) + d_v \nabla^2 v \\
\frac{\partial w}{\partial t} &= \alpha_{12} s(\mathrm{z}, \mathrm{k}_{12}) - \mathrm{k}_9 w + d_w \nabla^2 w \\
\frac{\partial z}{\partial t} &= \alpha_{13} s(\mathrm{u}, \mathrm{k}_{13}) - \mathrm{k}_{16} z
\end{aligned}
\tag{35}
$$

where $(\alpha_2, \alpha_5, \alpha_4, \alpha_{12}, \alpha_{13})$ are additional terms that can be used to change the saturation of the sigmoids. In this example they are set to one.

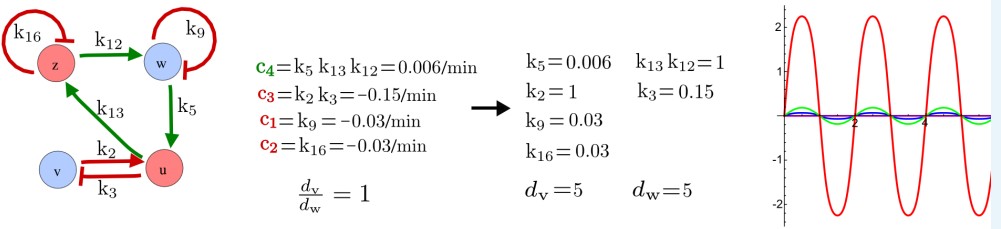

**Appendix 5—figure 2.** Simulation of the synthetic circuit in **Figure 5**. Left: Parameters used in **Equation 35** to simulate the synthetic circuits shown in **Figure 5**. Right: One-dimensional simulation using the sigmoid regulation terms.

Note that in **Equation (35)** the terms with $k_9$ and $k_{16}$ are not sigmoid regulation terms, since they represent first order decays. Similar to linear models, numerical simulations confirm that this system can form a stable spatial pattern. We also observe that changing the saturation of different regulatory sigmoid has interesting effects on the final aspect of the pattern. These numerical simulations are left for future theoretical analysis and are beyond the focus of this study that deals

with realistic reaction-diffusion topologies and the analytical conditions that lead to pattern formation.

Marcon *et al.* eLife 2016;5:e14022. DOI: 10.7554/eLife.14022

# Appendix 6: Model definitions and parameters used for the simulations

The simulations in the main text were performed by deriving systems of partial differential equations with linear reaction terms and negative cubic saturation terms from the networks, similar to previous approaches (**Raspopovic et al., 2014**; **Miura and Maini, 2004**). Systems of partial differential equations can be derived from the Jacobian matrix $\mathbf{J}$ and the diffusion matrix $\mathbf{D}$ as shown in the following example (**Appendix 6—figure 1**):

$$\begin{pmatrix} u_t \\ v_t \\ w_t \end{pmatrix} = \left[ \overset{\mathbf{J}}{\begin{pmatrix} 0 & k_2 & 0 \\ k_3 & k_4 & k_6 \\ k_7 & 0 & k_9 \end{pmatrix}} + \nabla^2 \overset{\mathbf{D}}{\begin{pmatrix} 0 & 0 & 0 \\ 0 & d_v & 0 \\ 0 & 0 & d_w \end{pmatrix}} \right] \begin{pmatrix} u \\ v \\ w \end{pmatrix} - \overset{\mathbf{S}}{\begin{pmatrix} u^3 \\ v^3 \\ 0 \end{pmatrix}}$$

$$\Downarrow$$

$$\begin{aligned} u_t &= k_2 v - u^3 \\ v_t &= k_3 u + k_4 v + k_6 w - v^3 + d_v \nabla^2 v \\ w_t &= k_7 u + k_9 w + d_w \nabla^2 w \end{aligned}$$

where the matrix $\mathbf{S}$ contains cubic saturation terms only for the nodes that are part of the destabilizing positive feedback (green arrows) and is zero otherwise.

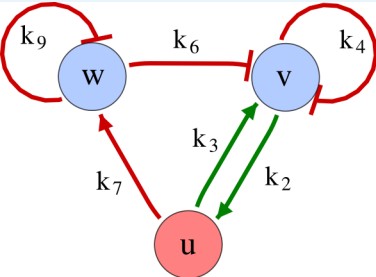

**Appendix 6—figure 1.** Example of a three-node network.

Systems derived in this manner have a homogeneous steady state at zero, in the example above $(u_0, v_0, w_0) = (0, 0, 0)$, and thus form self-organizing periodic patterns with concentration peaks of positive and negative values. Negative values do not represent negative concentrations (which are impossible), but represent a relative negative deviation from the homogeneous steady state, which was set to 0 for convenience.

The simulations presented in this study were executed using random initial conditions around the homogeneous steady state uniformly distributed in the interval (-0.001, 0.001).

## Parameters for the simulations in *Figure 1c*

The simulations in **Figure 1c** were performed on a unit-length spatial domain $0 \le x \le 1$ with the network described above. The amplitude of the concentration profiles was rescaled to facilitate visualization. The parameters are given in **Appendix 6–Tables 1** and **2** for the simulations on the left and on the right, respectively, of **Figure 1c**.

**Appendix 6—Table 1.** Parameters for the simulation in *Figure 1c*, left.

| $k_2$ | $k_3$ | $k_4$ | $k_6$ | $k_7$ | $k_9$ | $d_u$ | $d_v$ | $d_w$ |
|---|---|---|---|---|---|---|---|---|
| 1.5 E-4 | 0.33 E-4 | -1E-4 | -0.67 E-4 | 1 | -1 | 0 | 0.06 | 0.06 |

**Appendix 6—Table 2.** Parameters for the simulation in *Figure 1c*, right.

| $k_2$ | $k_3$ | $k_4$ | $k_6$ | $k_7$ | $k_9$ | $d_u$ | $d_v$ | $d_w$ |
|---|---|---|---|---|---|---|---|---|
| 1.5 E-4 | 0.33 E-4 | -1E-4 | 0.67 E-4 | -1 | -1 | 0 | 0.06 | 0.06 |

# Parameters for the simulations in *Figure 2c*

One-dimensional simulations were performed on a unit-length spatial domain: $0 \leq x \leq 1$. Two-dimensional simulations were performed on a squared domain with side length $l = 5$: $0 \leq x \leq 5$ and $0 \leq y \leq 5$. The networks and parameters of the simulations are shown in *Appendix 6–Tables 3*, *4*, and *5* and in *Appendix 6—figures 2*, *3*, and *4*.

**Appendix 6—Table 3.** Parameters for the simulation in *Figure 2c*, left.

| $k_1$ | $k_2$ | $k_3$ | $k_4$ | $d_v$ | $d_w$ |
|---|---|---|---|---|---|
| 0.5 | -1 | 0.53125 | -1 | 0.0125 | 0.05 |

**Appendix 6—Table 4.** Parameters for the simulation in *Figure 2c*, center.

| $k_2$ | $k_3$ | $k_4$ | $k_5$ | $k_7$ | $k_9$ | $d_u$ | $d_v$ | $d_w$ |
|---|---|---|---|---|---|---|---|---|
| 0.125 | 1 | -0.5 | -1 | 0.09375 | -0.25 | 0 | 0.02 | 0.02 |

**Appendix 6—Table 5.** Parameters for the simulation in *Figure 2c*, right.

| $k_2$ | $k_3$ | $k_4$ | $k_6$ | $k_7$ | $k_9$ | $d_u$ | $d_v$ | $d_w$ |
|---|---|---|---|---|---|---|---|---|
| 0.104 | 1 | -1 | -0.5 | 1 | -0.25 | 0 | 0.1 | 0.01 |

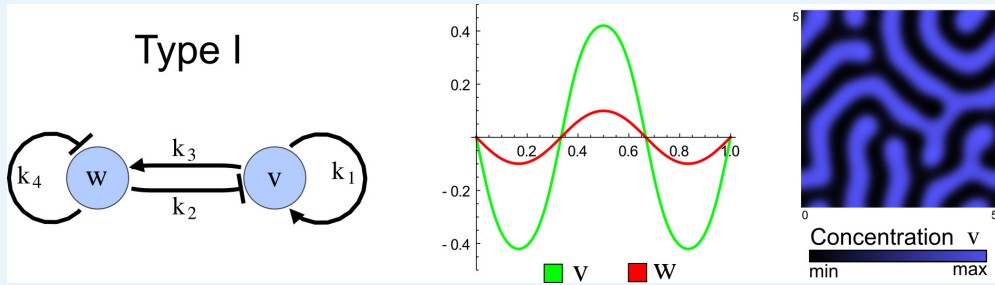

**Appendix 6—figure 2.** Network for the simulation in *Figure 2c*, left.

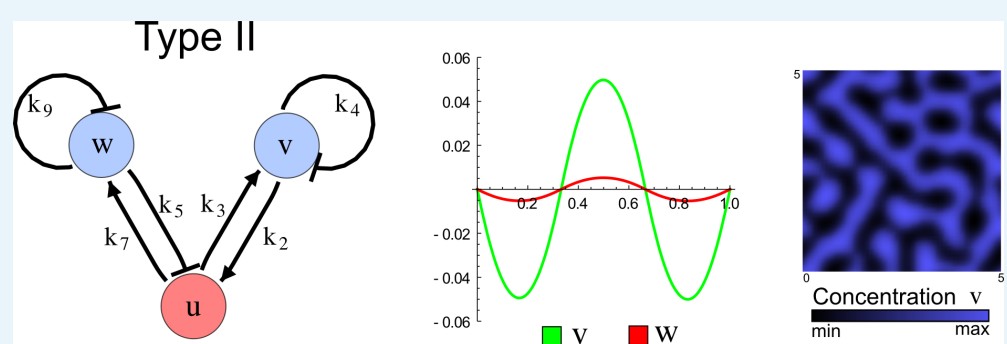

**Appendix 6—figure 3.** Network for the simulation in *Figure 2c*, center.

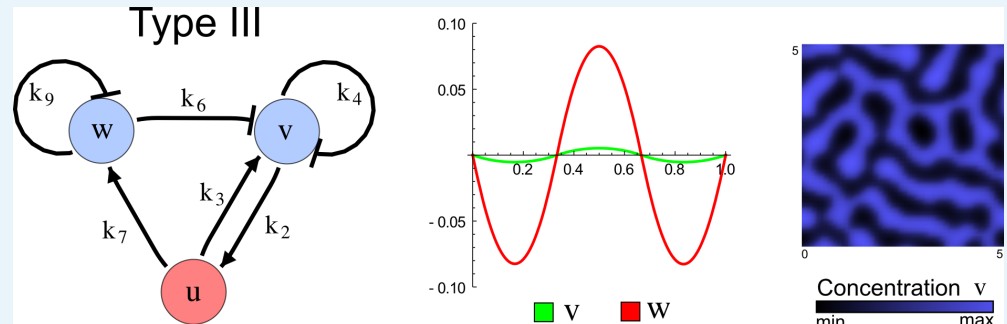

**Appendix 6—figure 4.** Network for the simulation in *Figure 2c*, right.

## Parameters for the simulation in *Figure 4b*

The one-dimensional simulation was performed on a domain with length 5: $0 \leq x \leq 5$. The network and parameters used for the simulation are shown in *Appendix 6—figure 5* and *Appendix 6–Table 6*. A system of partial differential equations that forms a pattern was obtained by defining cubic saturation terms for $\beta$, Sm, S, and W that are all part of the destabilizing positive feedback highlighted in green.

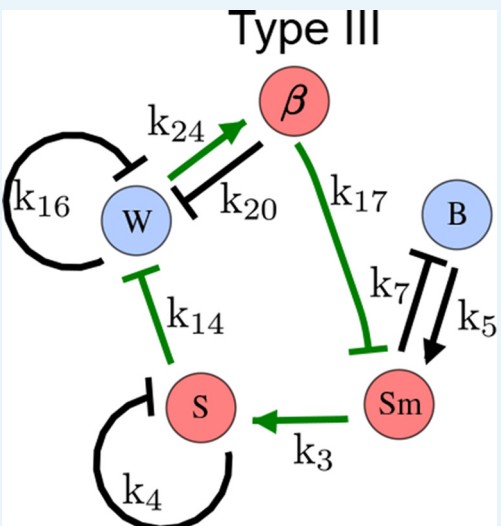

**Appendix 6—figure 5.** Network for the simulation in *Figure 4b*.

**Appendix 6—Table 6.** Parameters for the simulation in *Figure 4b*.

| $k_3$ | $k_4$ | $k_5$ | $k_7$ | $k_{14}$ | $k_{16}$ | $k_{17}$ | $k_{20}$ | $k_{24}$ | $d_{Sm}$ | $d_S$ | $d_B$ | $d_W$ | $d_\beta$ |
|---|---|---|---|---|---|---|---|---|---|---|---|---|---|
| 0.5 | -1 | 1 | -1 | -1 | -1 | -1 | -1 | 1 | 0 | 0 | 0.05 | 0.05 | 0 |

