## [Decision Letter]

Thank you for submitting your work entitled "High-throughput mathematical analysis identifies signaling networks for robust reaction-diffusion patterning" for consideration by *eLife*. Your article has been reviewed by two peer reviewers, and the evaluation has been overseen by Naama Barkai as the Reviewing Editor and Senior Editor.

The reviewers have discussed the reviews with one another and the Reviewing Editor has drafted this decision to help you prepare a revised submission.

The following individuals involved in review of your submission have agreed to reveal their identity: Hans Othmar.

Summary:

Determining the existence/non-existence of Turing instabilities is difficult for anything but the simplest network topologies. Consequently, a tool/resource/method to perform this calculation for more complex networks would be a significant and novel contribution to the field.

This paper outlines such a method. The method is clearly described (including a well-written supplemental), and rigorous. The use of examples to illustrate the approach is useful and convincing (particularly the synthetic biology example in which the method seems very useful).

The main finding of the paper is that Turing instabilities can readily form for a large range of parameters in realistic biological circuits including with different diffusivities of the components than commonly believed- this is a novel conclusion and is different from expectations from simpler networks.

Essential revisions:

1) The previous results and references mentioned by reviewer #2 should be incorporated into the analysis. This would probably require quite a substantial effort, but it probably necessary;

2). Several mis-interpretations are discussed by Reviewer #1. (in particular presence of LALI in type II/III networks but also the other points). Please correct and explain.

Reviewer #1:

1) I do not agree that the type II/type III networks do not show local activation, long range inhibition (LALI)

The authors find new networks that generate Turing instabilities without differential diffusivities (but see #2 below for a minor point). In several places, they state as a key result that these networks are "fundamentally different from the concepts of short-range activation and long-rage inhibition". We disagree. We think that LALI is present, but differential diffusivity is no longer required for LALI.

Type II network:

Viewed from the perspective of the variable u, then u inhibits itself over a long range via w, and activates itself over a short range via v.

Mathematically, the condition for instability (rewritten) is:

D_w / D_v > c3 / c4

(with c3, c4 defined to be positive)

Reinterpreting the parameters c3,c4 by the more traditional molecular half-lives, c3 = 1/ tW, c4 = 1/ tV

Then the instability condition is:

D_w t_w > D_v t_v

Now, if we write the lengthscale of a linearly degraded and diffusing molecule as

L_w = sqrt(D_w t_w)

Then we get:

L_w > L_vi.e. LALI.

Type III network

Viewed from the perspective of the variable u, u activates itself over a short range by v, and over a long range by the combined action of w and v. The effective range of w and v together is expected to be larger than v alone (if w diffuses); thus we expect LALI for any (positive) diffusivities.

To make this a little more concrete, assuming that v and w reach pseudosteady state relative to u (i.e. move to some region of parameter space).

Then the fourier transform of u, u(q), should obey (the linear terms only):

\partial u(q) / \partial t =

=u (q).[A / (D_v q^2+c_v) – B / ((D_v q^2+c_v) (D_w q^2+c_w))]

If we look at the second term, expanding to lowest order in q^2, we get a term like:

-u(q). B / (const + (D_v + D_w)q^2)

In this case the A-term is the local activation by v; the B-term is the long range inhibition via w and v – which by definition has an effective diffusivity larger than D_v.

Now these arguments won't work for all regions of parameter space – but we expect the same qualitative logic to hold true – LALI (albeit with no direct differential diffusivity).

2) Turing instabilities require differential diffusivity

This has been proved in the general case, see the Satnoianu 2005 reference. I think the point to make is that by having an immobile component and a diffusing component automatically satisfies differential diffusivity. This is more of a mathematical, rather than a biological point.

Satnoianu 2005:

Satnoianu, R. A., and P. van den Driessche. "Some remarks on matrix stability with application to Turing instability." Linear algebra and its applications 398 (2005): 69-74.

3) The graph theory is interesting, but its main point is obscure in the writing

I found the graph theory section very interesting. The main conclusion as I see it: it provides an elegant/pictorial way to calculate the a_k (Routh-Hurwitz) coefficients. These can then be used to derive the instability conditions.

However, from the writing it wasn't clear that this was the main result from the graph theory – when reading the main text and the supplemental, I thought that the graph theory was used to derive instability conditions directly without recourse to the Routh parameters. I would recommend more plainly/directly saying what the graph theory does.

Relatedly, the requirement for a cycle with positive weight ("the instability cycle") is perhaps related to the condition for Turing instability outlined in Satnoianu 2000 (referred to in the main text). In this case, you might be able to make a graph-theory-only condition: necessary conditions for instability are the existence of a positive weight cycle.

4) Clarify some 'robustness' terms in the main text

The term 'robust' was used in various places, some of which it was clear, and some of which it wasn't ("more robust" should be something like "more robust to parameter changes").

5) There are examples of a 3 node network with only 1 diffusing component that has a Turing instability. See example 3.1 in Satnoianu 2000. Only one diffusive species is required!

(What is really nice, however, is that I could get other examples using RDNets.com)

Reviewer #2:

This paper describes a computational approach and associated software aimed at automating the process of finding kinetic networks that support Turing instabilities. While this is a worthwhile effort, there are several issues concerning this paper.

First and foremost is that the authors are apparently unaware of a result proved long ago that gives necessary conditions on the kinetic network alone for the absence of Turing instabilities. Thus the violation of any of these conditions can give rise to a Turing instability. The result appears in a paper by Othmer in 1980, entitled 'Synchronized and differentiated modes of cellular dynamics', which appeared in Dynamics of Synergetic Systems, H. Haken ed. The theorem goes as follows (σ(A) is the spectrum of A.

Let D be diagonal with Dj {greater than or equal to} 0. In order that σ(K − μD) ⊂ LHP for all such D and all μ ∈ [0, ∞),it is necessary that

• σ(K) ⊂ LHP

• σ(K[i1, i2, · · ·, ip]) ⊂ LHP for all pth

-order submatrices of K, where 1 {less than or equal to} p {less than or equal to} n − 1.

The result could alter the search for Turing instabilities described in Appendix 1 quite dramatically, since one could first categorize the networks that have a sub-network that is unstable when severed from the full network at the steady state of the full system, and only then determine the pattern of diffusion coefficients that produce different types of instabilities. This eliminates the need to examine all RH determinants over the entire range of wave numbers. Notice that the theorem does not require that the fully-isolated sub-network be unstable on its own, though this is allowable, but rather that the Jacobian of the terms affecting the sub-network have one or more eigenvalues in the RHP.

A second remark is that much of Appendix 1 and most of Appendix 2 contains material readily available in the literature, and could be eliminated. It would be better to replace this material with a precise formulation of the problem the authors are addressing. For example, they implicitly assume that all diffusible species must satisfy homogeneous Neumann boundary conditions, but this means that the results don't apply to common situations in pattern formation, such as the production of a morphogen at the boundary of the domain.

---

## [Author Response]

*Summary:*

*Determining the existence/non-existence of Turing instabilities is difficult for anything but the simplest network topologies. Consequently, a tool/resource/method to perform this calculation for more complex networks would be a significant and novel contribution to the field.*

*This paper outlines such a method. The method is clearly described (including a well-written supplemental), and rigorous. The use of examples to illustrate the approach is useful and convincing (particularly the synthetic biology example in which the method seems very useful).*

*The main finding of the paper is that Turing instabilities can readily form for a large range of parameters in realistic biological circuits including with different diffusivities of the components than commonly believed- this is a novel conclusion and is different from expectations from simpler networks.*

We thank the editor for the positive assessment of our work and for highlighting the scientific impact of our findings. To emphasize the novel conclusions regarding the components’ diffusivities pointed out by the editor, we have changed the title to “High-throughput mathematical analysis identifies Turing networks for patterning with equally diffusing signals” and improved the Abstract in a similar direction.

*Essential revisions:*

*1) The previous results and references mentioned by reviewer #2 should be incorporated into the analysis. This would probably require quite a substantial effort, but it probably necessary;*

As outlined in more detail below, the approach that we used to search for Turing patterning systems already took the theoretical results mentioned by reviewer #2 into account. We have now clarified the details of our approach in Appendix 1.

*2). Several mis-interpretations are discussed by Reviewer #1. (in particular presence of LALI in type II/III networks but also the other points). Please correct and explain.*

We thank reviewer #1 for the insightful comments. Indeed, the relation of our findings with “local activation, long range inhibition” (LALI) models was not sufficiently detailed in the previous version of our manuscript. We agree that a thorough discussion of the underlying pattern forming mechanisms is of central interest to a broad readership. We have therefore significantly extended our analysis and now provide an additional section comprising 18 pages of simulations and discussion (Appendix 3) to address the role of LALI in Type II/III networks.

*Reviewer #1:*

*1) I do not agree that the type II/type III networks do not show local activation, long range inhibition (LALI)*

*The authors find new networks that generate Turing instabilities without differential diffusivities (but see #2 below for a minor point). In several places, they state as a key result that these networks are "fundamentally different from the concepts of short-range activation and long-rage inhibition". We disagree. We think that LALI is present, but differential diffusivity is no longer required for LALI.*

*Type II network:*

*Viewed from the perspective of the variable u, then u inhibits itself over a long range via w, and activates itself over a short range via v.*

*Mathematically, the condition for instability (rewritten) is:*

*D_w / D_v > c3 / c4*

*(with c3, c4 defined to be positive)*

*Reinterpreting the parameters c3,c4 by the more traditional molecular half-lives, c3 = 1/ tW, c4 = 1/ tV*

*Then the instability condition is:*

*D_w t_w > D_v t_v*

*Now, if we write the lengthscale of a linearly degraded and diffusing molecule as*

*L_w = sqrt(D_w t_w)*

*Then we get:*

*L_w > L_vi.e. LALI.*

*Type III network*

*Viewed from the perspective of the variable u, u activates itself over a short range by v, and over a long range by the combined action of w and v. The effective range of w and v together is expected to be larger than v alone (if w diffuses); thus we expect LALI for any (positive) diffusivities.*

*To make this a little more concrete, assuming that v and w reach pseudosteady state relative to u (i.e. move to some region of parameter space).*

*Then the fourier transform of u, u(q), should obey (the linear terms only):*

*\partial u(q) / \partial t =*

*=u (q).[A / (D_v q^2+c_v) – B / ((D_v q^2+c_v) (D_w q^2+c_w))]*

*If we look at the second term, expanding to lowest order in q^2, we get a term like:*

*-u(q). B / (const + (D_v + D_w)q^2)*

*In this case the A-term is the local activation by v; the B-term is the long range inhibition via w and v – which by definition has an effective diffusivity larger than D_v.*

*Now these arguments won't work for all regions of parameter space – but we expect the same qualitative logic to hold true – LALI (albeit with no direct differential diffusivity).*

We are grateful to the reviewer for these excellent remarks and the valuable input. We agree with the reviewer that it is important to relate our findings to the LALI mechanism originally proposed by Meinhardt and Gierer. We therefore wrote a new section (Appendix 3) to investigate with additional analyses and numerical simulations whether the dynamics of Type II/III networks fit with the classical description of pattern formation by LALI. This analysis indicates that the mechanism underlying pattern formation is different from the concept of shortrange activation and long-range inhibition based on differential diffusivity. We have therefore corrected the relevant sentences in our manuscript and now state that our analysis identified systems that are different from models of “short-range activation and long-range inhibition based on differential diffusivity”.

1) We value the reviewer’s elegant interpretation of the Type II network shown in Figure 2. As the reviewer points out, the final aspect of the pattern together with the larger value of D_w_/c_2_ compared to D_v_/c_1_ could be interpreted as a longer range of the inhibitor w compared to the activator v. However, we find that by just altering the magnitude of the rates associated with cycle c3 (without modifying the overall strength of c_3_ or D_w_/c_2_ and D_v_/c_1_), the final aspect of the periodic patterns can change, such that v – surprisingly – appears to have a longer range than w (Figure 18). Moreover, the definition of ranges in terms of ratios between diffusion and decay cannot be generalized to other Type II networks that lack negative self-regulatory feedbacks of the diffusible molecules (e.g. Figure 19) or where the minimum diffusion ratio d is defined by more complicated reaction terms (e.g. equation 31 in Appendix 3).

2) Similar to the case discussed above, the final aspect of the patterns in Type III networks can also change depending on the reaction kinetics and diffusion ratios, showing that the relationship between the range of v and w within the same network can vary (Figure 20). Moreover, for other Type III networks, alternative ad hoc approximations of the effective ranges would need to be defined in order to interpret the systems within the LALI framework. In cases with equally diffusing reactants, such effective ranges appear to be just reformulations of the reaction kinetics that do not contribute to the identification of the general principles that underlie pattern formation.

3) To develop a general understanding of the pattern formation process in Type II/III networks, we have extended our work with simulations on a pair of cells inspired by Turing’s original analysis of pattern formation dynamics. Turing’s original simulation highlights that the role of differential diffusivity is not to limit the expansion of the activator but to destabilize the equilibrium state by maintaining an imbalance between reactants that drives a continuous deviation from equilibrium. Importantly, the reaction terms of a Turing system guarantee that the deviation happens simultaneously above and below the equilibrium state. In agreement with these observations, one-dimensional simulations like the one shown in Appendix 6—figure 8 reveal that the periodic patterns of Turing systems are formed with a simultaneous appearance of activation and inhibition peaks. The periodic patterns therefore do not reflect a longer range of the inhibitor to limit the auto-activation.

We propose instead that the role of immobile reactants in Type II and Type III networks is not to implement an effective difference in the ranges of local auto-activation and long-range inhibition, but rather to help the system to diverge from equilibrium, which in classic two-component Turing systems is achieved by differential diffusivity. We find that immobile factors help to destabilize the system since they are not subjected to the equilibrating effect of diffusion and can therefore fulfill a role as “capacitors” that integrate the effect of diffusing reactants to destabilize the reaction-diffusion system by quickly amplifying perturbations (Figure 12).

*2) Turing instabilities require differential diffusivity*

*This has been proved in the general case, see the Satnoianu 2005 reference. I think the point to make is that by having an immobile component and a diffusing component automatically satisfies differential diffusivity. This is more of a mathematical, rather than a biological point.*

*Satnoianu 2005:*

*Satnoianu, R. A., and P. van den Driessche. "Some remarks on matrix stability with application to Turing instability." Linear algebra and its applications 398 (2005): 69-74.*

We have made the appropriate changes in the title and the Abstract of our manuscript to highlight that our analysis refers to the lack of differential diffusivity of the mobile signalling molecules.

*3) The graph theory is interesting, but its main point is obscure in the writing*

*I found the graph theory section very interesting. The main conclusion as I see it: it provides an elegant/pictorial way to calculate the a_k (Routh-Hurwitz) coefficients. These can then be used to derive the instability conditions.*

*However, from the writing it wasn't clear that this was the main result from the graph theory – when reading the main text and the supplemental, I thought that the graph theory was used to derive instability conditions directly without recourse to the Routh parameters. I would recommend more plainly/directly saying what the graph theory does.*

Our graph-theoretical formalism provides a mechanistic understanding of the requirements for stability and diffusion-driven instability in a network. Importantly and in contrast with purely algebraic or numeric approaches, this is independent of the network complexity or the number of nodes because the graph formalism allows to break down the network into cycles that have a clear and defined role in the dynamics of pattern formation. This provides an intuitive understanding of the requirements for the generation of Turing patterns in terms of feedback, cycles, and network topologies.

We now explain in more detail at the beginning of Appendix 2 that the graph-theoretical formalism allows to re-write the coefficients of the characteristic polynomial and therefore to derive the Routh-Hurwitz determinants in terms of cycles.

*Relatedly, the requirement for a cycle with positive weight ("the instability cycle") is perhaps related to the condition for Turing instability outlined in Satnoianu 2000 (referred to in the main text). In this case, you might be able to make a graph-theory-only condition: necessary conditions for instability are the existence of a positive weight cycle.*

The reviewer is correct. The condition in Satnoianu 2000 is related to the requirement for a positive cycle. However, in our work we did not intend to recast in graph-theoretical form results from reaction network theory that have already been proven in another way. This is the reason why we did not pursue this line of inquiry, but the development of such graph-theoretical counterparts has been reviewed elsewhere (e.g. Radde 2010).

Radde N, Bar NS, M Banaji (2010). Graphical methods for analysing feedback in biological networks–A survey. International Journal of Systems Science. 41: 35-46.

*4) Clarify some 'robustness' terms in the main text*

*The term 'robust' was used in various places, some of which it was clear, and some of which it wasn't (e.g. line 212, "more robust" should be something like "more robust to parameter changes").*

We have now clarified the term “robust” according to the reviewer’s suggestion where appropriate.

*5) There are examples of a 3 node network with only 1 diffusing component that has a Turing instability. See example 3.1 in Satnoianu 2000. Only one diffusive species is required!*

*(What is really nice, however, is that I could get other examples using RDNets.com)*

We thank the reviewer for pointing out the value of our software. There are indeed networks that satisfy Turing instability conditions with only one diffusing component. We observe, however, that in such networks the de-stabilizing cycle encompasses only non-diffusible nodes, which causes the network to amplify any of the fluctuations in the initial conditions without a preferred wavelength. Similar behaviors have also been observed when there is a positive self-regulatory feedback on the non-diffusible node (Klika et al. 2012, White and Gilligan 1998). RDNets offers the possibility to filter out such “noise amplifiers”.

Klika V, Baker RE, Headon D, EA Gaffney (2012). The influence of receptor-mediated interactions on reaction-diffusion mechanisms of cellular self-organisation. Bull Math Biol 74: 935-957.

White KAJ, CA Gilligan (1998). Spatial heterogeneity in three species, plant–parasite–hyperparasite, systems. Philos Trans R Soc Lond B Biol Sci 353: 543-557.

*Reviewer #2:*

*This paper describes a computational approach and associated software aimed at automating the process of finding kinetic networks that support Turing instabilities. While this is a worthwhile effort, there are several issues concerning this paper.*

*First and foremost is that the authors are apparently unaware of a result proved long ago that gives necessary conditions on the kinetic network alone for the absence of Turing instabilities. Thus the violation of any of these conditions can give rise to a Turing instability. The result appears in a paper by Othmer in 1980, entitled 'Synchronized and differentiated modes of cellular dynamics', which appeared in Dynamics of Synergetic Systems, H. Haken ed. The theorem goes as follows (σ(A) is the spectrum of A.*

*Let D be diagonal with Dj {greater than or equal to} 0. In order that σ(K − μD) ⊂ LHP for all such D and all μ ∈ [0, ∞),it is necessary that*

*• σ(K) ⊂ LHP*

*• σ(K[i1, i2, · · ·, ip]) ⊂ LHP for all pth*

*-order submatrices of K, where 1 {less than or equal to} p {less than or equal to} n − 1.*

*The result could alter the search for Turing instabilities described in Appendix 1 quite dramatically, since one could first categorize the networks that have a sub-network that is unstable when severed from the full network at the steady state of the full system, and only then determine the pattern of diffusion coefficients that produce different types of instabilities. This eliminates the need to examine all RH determinants over the entire range of wave numbers. Notice that the theorem does not require that the fully-isolated sub-network be unstable on its own, though this is allowable, but rather that the Jacobian of the terms affecting the sub-network have one or more eigenvalues in the RHP.*

We thank the reviewer for pointing out that the existence of an unstable subnetwork is sufficient for the existence of Turing patterns. In our analysis, we had already taken the implications of this theorem into account. We used the equivalent result from Cross 1978 (as explained in Othmer 1980) and another theorem proven by Kellog (theorem 4, page 174 in Kellog 1972; reviewed in Hershkowitz 1992) to characterize the conditions that determine stable Turing patterns. We now clarify this point in Appendix 1 (see “Step 5. Selecting unstable networks with diffusion”).

Cross GW (1978). Three types of matrix stability. Linear algebra and its applications. 20: 253- 263.

Kellogg RB (1972). On complex eigenvalues of M and P matrices. Numerische Mathematik. 19: 170-175.

Hershkowitz D (1992). Recent directions in matrix stability. Linear Algebra and its Applications. 171: 161-186

*A second remark is that much of Appendix 1 and most of Appendix 2 contains material readily available in the literature, and could be eliminated. It would be better to replace this material with a precise formulation of the problem the authors are addressing. For example, they implicitly assume that all diffusible species must satisfy homogeneous Neumann boundary conditions, but this means that the results don't apply to common situations in pattern formation, such as the production of a morphogen at the boundary of the domain.*

Considering the broad readership of *eLife*, Appendix 1 and Appendix 2 are vital for an understanding of our findings by non-experts and additionally serve to define the notation of our analysis. Following the suggestion of the reviewer, we have now improved Appendix 1 by adding a precise formulation of the problem we are addressing, including the definition of boundary conditions. We restrict our analysis to zero flux boundary conditions because we are interested in deriving the analytical conditions required to form self-organizing spatial patterns in the absence of pre-existing asymmetries or external inputs.